# Exploring Edge Probability Graph Models Beyond Edge Independency: Concepts, Analyses, and Algorithms

## Abstract

Desirable random graph models (RGMs) should *(i)* reproduce *common patterns in real-world graphs (e.g., high clustering)*, *(ii)* generate *variable* (i.e., not overly similar) graphs, and *(iii)* remain *tractable* to compute and control graph statistics. A common class of RGMs (e.g., Erdős-Rényi and stochastic Kronecker) outputs edge probabilities, and we need to realize (i.e., sample from) the edge probabilities to generate graphs. Typically, each edge's existence is assumed to be determined independently for simplicity and tractability. However, with edge independency, RGMs theoretically cannot produce high subgraph densities and high output variability simultaneously. In this work, we explore realization beyond edge independence that can better reproduce common patterns while maintaining high tractability and variability. Theoretically, we propose an edge-dependent realization framework called *binding* that provably preserves output variability, and derive *closed-form* tractability results on subgraph (e.g., triangle) densities in generated graphs. Practically, we propose algorithms for graph generation with binding and parameter fitting of binding. Our empirical results demonstrate that binding exhibits high tractability and well reproduce patterns such as high clustering, significantly improving upon existing RGMs assuming edge independency.

## 1 Introduction

Random graph models (RGMs) help us understand, analyze, and predict real-world systems (Drobyshevskiy & Turdakov, 2019), with various practical applications, e.g., graph algorithm testing (Murphy et al., 2010), statistical testing (Ghoshdastidar et al., 2017), and graph anonymization (Backstrom et al., 2007). Desirable RGMs should generate graphs with *common patterns* in real-world graphs, such as high clustering,[1] power-law degrees, and small diameters (Chakrabarti & Faloutsos, 2006). At the same time, the generated graphs should be *variable*, i.e., not highly-similar or even near-identical, and the RGMs should be *tractable*, i.e., we can compute and control graph statistics.[2] Many RGMs output individual edge probabilities and generate graphs accordingly, e.g., the Erdős-Rényi model (Erdős & Rényi, 1959), the Chung-Lu model (Chung & Lu, 2002), the stochastic block model (Holland et al., 1983), and the stochastic Kronecker model (Leskovec et al., 2010). To generate graphs from edge probabilities, we need realization (i.e., sampling), where edge independency (i.e., the edge existences are determined mutually independently) is widely assumed for the sake of simplicity and tractability. Although edge-independent RGMs have high tractability and they may reproduce some common patterns (e.g., power-law degrees and small diameters), they empirically fail to preserve some other patterns, especially high clustering (Moreno et al., 2018; Seshadri et al., 2013). Moreover, edge-independent RGMs theoretically cannot generate graphs with high triangle density and high output variability at the same time (Chanpuriya et al., 2021).

Naturally, we ask: Can we apply realization without assuming edge independency so that we can improve upon such RGMs to generate graphs with common patterns and high variability, while still ensuring high tractability? To address this question, we propose and explore the concept of *edge probability graph models* (EPGMs), i.e., RGMs that are still based on edge probabilities but do not assume edge independency, from theoretical and practical perspectives. Our key message is a positive answer to the question. Specifically, our novel contributions are four-fold:

---

[1] *High clustering* means high subgraph densities, as used by, e.g., Newman (2003) and Pfeiffer et al. (2012).

[2] In this work, *tractability* refers to the feasibility of deriving graph statistics, rather than the ability to handle large-scale graphs (which we refer to as *scalability*).

1. **Concepts (Section 4):** We formally define EPGMs with related concepts, and theoretically show some basic properties of EPGMs, e.g., even with edge dependency introduced, the *variability is maintained* in the generated graphs as the corresponding edge-independent model.

2. **Analyses (Section 5):** We propose *pattern-reproducing*, *tractable*, and *flexible* realization schemes called *binding* to construct EPGMs with different levels of edge dependency, and derive tractability results on the *closed-form* subgraph (e.g., triangle) densities.

3. **Algorithms (Section 5):** We propose practical algorithms for graph generation with binding, and for efficient parameter fitting to control the graph statistics generated by EPGMs with binding.

4. **Experiments (Section 6):** We use our binding and fitting algorithms to generate graphs. Via experiments on real-world graphs, we show the power of edge dependency to reproduce common graph patterns and validate the correctness of our theoretical analyses and practical algorithms.

**Reproducibility.** The code and datasets are available in the online appendix (Anonymous, 2024).

## 2 PRELIMINARIES

**Graphs.** A node-labelled *graph* $G = (V, E)$ is defined by a *node set* $V = V(G)$ and an *edge set* $E = E(G) \subseteq \binom{V}{2} \coloneqq \{V' \subseteq V \colon |V'| = 2\}$.[3] For a node $v \in V$, the set of *neighbors* of $v$ is $N(v; G) = \{u \in V \colon (u, v) \in E(G)\}$. The *degree* $d(v; G)$ of $v$ is the number of its neighbors, i.e., $d(v; G) = |N(v; G)|$. Given $V' \subseteq V$, the *induced subgraph* of $G$ on $V'$ is $G[V'] = (V', E \cap \binom{V'}{2})$.

**Random graph models (RGMs).** Fix a node set $V = [n] = \{1, 2, \ldots, n\}$ with $n \in \mathbb{N}$. Let $\mathcal{G}(V) = \{G = (V, E) \colon E \subseteq \binom{V}{2}\}$ denote the set of all $2^{\binom{n}{2}}$ possible node-labelled graphs on $V$. A *random graph model* (RGM) is defined as a probability distribution $f : \mathcal{G}(V) \to [0, 1]$ with $\sum_{G \in \mathcal{G}(V)} f(G) = 1$. For each graph $G \in \mathcal{G}(V)$, $f(G)$ is the probability of $G$ being generated by the RGM $f$. For each node pair $(u, v)$ with $u, v \in V$, the (marginal) *edge probability* of $(u, v)$ under the RGM $f$ is $\Pr_f[(u, v)] \coloneqq \sum_{G \in \mathcal{G}(V)} f(G) \mathbf{1}[(u, v) \in E(G)]$, where $\mathbf{1}[\cdot]$ is the indicator function.

**Edge independent graph models (EIGMs).** Given edge probabilities, edge independency is widely assumed in many existing RGMs, resulting in the concept of edge independent graph models (EIGMs; also known as *inhomogeneous Erdős-Rényi graphs* (Klopp et al., 2017)).

**Definition 2.1** (EIGMs (Chanpuriya et al., 2021)). Given edge probabilities $p \colon \binom{V}{2} \to [0, 1]$, the *edge independent graph model* (EIGM) w.r.t. $p$ is the RGM $f_p^{\text{EI}}$ defined by $f_p^{\text{EI}}(G) = \prod_{(u,v) \in E(G)} p(u, v) \prod_{(u',v') \notin E(G)} (1 - p(u', v')), \forall G \in \mathcal{G}(V)$.

## 3 RELATED WORK AND BACKGROUND

### 3.1 LIMITATIONS OF EIGMS

This work is motivated by the theoretical findings of Chanpuriya et al. (2021) on the limitations of EIGMs and the power of edge (in)dependency. They defined the concept of *overlap* to measure the variability of RGMs, where a high overlap value implies low variability. Roughly speaking, the overlap of an RGM is the expected proportion of edges co-existing in two generated graphs.

**Definition 3.1** (Overlap (Chanpuriya et al., 2021)). Given an RGM $f : \mathcal{G}(V) \to [0, 1]$, the *overlap* of $f$ is defined as $\text{Ov}(f) = \frac{\mathbb{E}_f\left[|E(G') \cap E(G'')|\right]}{\mathbb{E}_f|E(G)|}$, where $G$, $G'$, and $G''$ are three mutually independent random graphs generated by $f$.

*Remark* 3.2. High variability (i.e., low overlap) is important for RGMs (De Cao & Kipf, 2018), as generating overly similar graphs undermines RGMs' effectiveness in their common applications, e.g., graph algorithm testing, statistical testing, and graph anonymization (see Section 1).

Chanpuriya et al. (2021) showed that EIGMs are unable to generate graphs with high triangle density (i.e., with many triangles) unless EIGMs memorize a whole input graph (i.e., have high overlap).

**Theorem 3.3** (Limited triangles by EIGMs (Chanpuriya et al., 2021)). *For any* $p \colon \binom{V}{2} \to [0, 1]$, $\mathbb{E}_{f_p^{EI}}[\triangle(G)] \leq \frac{\sqrt{2}}{3} \left(\text{Ov}(f_p^{EI}) \sum_{(u,v) \in \binom{V}{2}} p(u, v)\right)^{3/2}$, *where* $\triangle(G)$ *is the number of triangles in* $G$.

---

[3] In this work, we consider undirected unweighted graphs without self-loops following common settings for random graph models. See Appendix D.1 for discussions on more general graphs.

Chanpuriya et al. (2024) recently extended their theoretical results, showing triangle-density upper bounds w.r.t. overlap in different types of edge-dependent RGMs.[4] However, they did not provide practical graph generation algorithms[5] or detailed tractability results, while tractability results and practical graph generation are part of our focus in this work.

Some methods shift edge probabilities by accept-reject (Mussmann et al., 2015) or mixing different EIGMs (Kolda et al., 2014b; Lancichinetti et al., 2008), in order to improve upon existing EIGMs. Such methods are essentially still EIGMs, and by Theorem 3.3, they inevitably have high overlap (i.e., low variability). See Appendix E.6 for more discussion and evaluation on such methods.

### 3.2 Edge dependency in RGMs

Despite the popularity of EIGMs, edge dependency also widely exists in various RGMs, e.g., preferential attachment models (Barabási & Albert, 1999), small-world graphs (Watts & Strogatz, 1998), copying network models (Kleinberg et al., 1999), random geometric graphs (Penrose, 2003), and exponential random graph models (Lusher et al., 2013). Some other models use additional mechanisms on top of existing models to introduce edge dependency by, e.g., directly forming triangles (Pfeiffer et al., 2012; Wegner & Olhede, 2021). Exchangeable network models (ENMs) (Lovász & Szegedy, 2006; Diaconis & Janson, 2007) also involve edge dependency, where isomorphic graphs are generated with the same probability (i.e., all nodes are treated probabilistically in symmetry). However, ENMs cannot generate graphs with sparsity and power-law degrees, which are common patterns in real-world graphs (Crane & Dempsey, 2016). Recent efforts have introduced *asymmetry among nodes* to enhance expressiveness (Crane & Dempsey, 2018; Wu et al., 2025). In the same spirit but from a different perspective, we aim to improve expressiveness by introducing *dependence among edges* upon EIGMs. Since we build RGMs based on edge-probability models, the nodes are asymmetric (i.e., non-exchangeable), except for the Erdős-Rényi model with uniform edge probabilities. Notably, the closed-form tractability results on subgraph densities derived by us (Theorems 5.8 and B.5) are usually unavailable for existing RGMs with edge dependency (Drobyshevskiy & Turdakov, 2019). Usually, only *asymptotic* results, as the number of nodes approaches infinity, are available for such models (Ostroumova Prokhorenkova, 2017; Gu et al., 2013; Bhat et al., 2016). In this work, we propose novel edge-dependent RGMs with the following desirable properties:

- *Reproducing common patterns* observed in real-world graphs across different domains, e.g., high clustering, power-law degrees, and small diameters (Chakrabarti & Faloutsos, 2006).
- *Having high variability*, generating graphs with low overlap (see Definition 3.1).
- *Having high tractability*, with the feasibility to obtain closed-form results of graph statistics.

## 4 Edge probability graph models: Concepts and basic properties

Given edge probabilities, EIGMs generate graphs assuming edge independency. In contrast, we explore a broader class of edge probability graph models (EPGMs) going beyond edge independency.

**Definition 4.1** (EPGMs). Given edge probabilities $p\colon \binom{V}{2} \to [0,1]$, the set $\mathcal{F}(p)$ of *edge probability graph models* (EPGMs) w.r.t. $p$ consists of all the RGMs satisfying the given marginal edge probabilities, i.e., $\mathcal{F}(p) \coloneqq \{f\colon \Pr_f[(u,v)] = p(u,v), \forall u,v \in V\}$.

The concept of EPGMs decomposes each RGM into two factors: **(F1)** the marginal probability of each edge and **(F2)** how the edge probabilities are realized (i.e., sampled), where (F2) has been overlooked by EIGMs and this decomposition introduces a novel way of imposing edge dependency. Below, we show some basic properties of EPGMs and discuss their meanings and implications.

**Property 4.2.** *EIGMs are special cases of EPGMs w.r.t. the same edge probabilities.*

*Proof.* See Appendix B for all the formal statements and proofs not covered in the main text. □

**Property 4.3.** *Each RGM can be represented as an EPGM (w.r.t its marginal edge probabilities).*

While Property 4.3 is an immediate result following the definition of EPGMs, it shows the *generality* of the concept of EPGMs, yet also implies the impossibility of exploring all possible EPGMs, which motivates us to find *good subsets* of EPGMs. Specifically, Property 4.3 tells us that each RGM can be represented as an EPGM w.r.t. *some* edge probabilities. What can we obtain for *given* edge

---

[4]In EPGMs, the overlap is constant yet we can have different triangle densities. See Property 4.7.

[5]Their graph generation algorithm is not practical since it relies on maximal clique enumeration, which is time-consuming (Eblen et al., 2012). See Appendix D.2 for more discussions.

probabilities? For this, in Property 4.4, we obtain the upper bounds of expected subgraph densities in the graphs generated by EPGMs with given edge probabilities.

**Property 4.4** (Upper bound of edge-group probabilities in EPGMs). *For any $p\colon \binom{V}{2} \to [0,1]$ and any $P \subseteq \binom{V}{2}$, $\mathrm{Pr}_f[P \subseteq E(G)] \leq \min_{(u,v)\in P} p(u,v), \forall f \in \mathcal{F}(p)$.*

*Remark* 4.5. Later, we shall show that the upper bound in Property 4.4 is tight, i.e., we can find EPGMs achieving the upper bound (see Lemma B.2).

Property 4.4 can be applied to obtain the upper bounds of the expected number of specific subgraphs, e.g., cliques and cycles. Below is an example on the number of triangles (i.e., $\triangle(G)$).

**Corollary 4.6.** *For any $p$, $\mathbb{E}_f[\triangle(G)] \leq \sum_{\{u,v,w\}\in\binom{V}{3}} \min(p(u,v), p(u,w), p(v,w)), \forall f \in \mathcal{F}(p)$.*

**Property 4.7** (EPGMs have constant expected degrees and overlap). *For any $p\colon \binom{V}{2} \to [0,1]$, the expected node degrees and overlap (see Definition 3.1) of all the EPGMs w.r.t. $p$ are constant.*

Property 4.7 implies that, for given edge probabilities, compared to EIGMs, considering more general EPGMs neither changes expected degrees nor impairs the variability of the generated graphs. Many EIGMs (e.g., Chung-Lu and Kronecker) can generate graphs with desirable degrees, and this property ensures that EPGMs can inherit such strengths (see Figure 1 for empirical evidence). As discussed in Remark 3.2, high variability is important and desirable for RGMs.

In this work, we explore EPGMs from both theoretical and practical perspectives, aiming to answer two research questions inspired by the basic properties of EPGMs above:

- **(RQ1; Theory)** What good subsets of EPGMs are pattern-reproducing, flexible, and tractable?
- **(RQ2; Practice)** How to generate graphs using such EPGMs and fit the parameters of EPGMs?

## 5 BINDING: PATTERN-REPRODUCING, FLEXIBLE, AND TRACTABLE EPGMS

We aim to construct EPGMs that reproduce *common patterns* (specifically, high clustering) and are *flexible* (i.e., different levels of dependency), in a *tractable* (i.e., controllable graph statistics) way.

### 5.1 BINDING: A GENERAL FRAMEWORK FOR EPGMS WITH HIGH CLUSTERING

As discussed in Section 1, desirable RGMs should generate graphs with common patterns, e.g., high clustering, power-law degrees, and small diameters (Chakrabarti & Faloutsos, 2006). We focus on the bottleneck of EIGMs and aim to construct EPGMs with high clustering (i.e., subgraph densities).[6] To this end, we study and propose *binding*, a general mathematical framework that introduces positive dependency among edges, where multiple edge existences are determined together.

---

**Algorithm 1:** General Binding

**Input** : (1) $p\colon \binom{V}{2} \to [0,1]$: edge probabilities;
  (2) $\mathcal{P}$ s.t. $\binom{V}{2} = \bigcup_{P\in\mathcal{P}} P$ and
  $P \cap P' = \emptyset, \forall P \neq P' \in \mathcal{P}$: pair partition
**Output:** $G$: generated graph
1 $E \leftarrow \emptyset$
2 **for** $P \in \mathcal{P}$ **do**
3 $\quad\lfloor\ E \leftarrow E \cup \mathtt{binding}(p, P)$
4 **return** $G = (V, E)$
5 **Procedure** $\mathtt{binding}(\hat{p}, \hat{P})$
6 $\quad$ sample a random variable $s \sim \mathcal{U}(0, 1)$
7 $\quad \hat{E} \leftarrow \emptyset$
8 $\quad$ **for** $(u, v) \in \hat{P}$ **do**
9 $\quad\quad$ **if** $s \leq \hat{p}(u, v)$ **then**
10 $\quad\quad\quad\lfloor\ \hat{E} \leftarrow \hat{E} \cup \{(u, v)\}$
11 $\quad$ **return** $\hat{E}$

---

Binding is the probabilistic process in Algorithm 1, where edge dependence is imposed in each group of pairs. Specifically, in each group, if a node pair is sampled as an edge, all the pairs with higher edge probabilities must be sampled too. Note that, Algorithm 1 describes a general framework, while our practical algorithms (Algorithms 2 and 3) do not need to choose an explicit partition $\mathcal{P}$ beforehand.

**Definition 5.1** (Binding). Given edge probabilities $p$ and a partition $\mathcal{P}$, *binding* gives the RGM $f^{\mathrm{BD}}_{p;\mathcal{P}}$ as follows. For each $P_i \in \mathcal{P}$, write $P_i = \{(u_{i1}, v_{i1}), \ldots, (u_{i|P_i|}, v_{i|P_i|})\}$ such that $p(u_{i1}, v_{i1}) \geq \cdots \geq p(u_{i|P_i|}, v_{i|P_i|})$, and let $P_{i;k} := \{(u_{i1}, v_{i1}), \ldots, (u_{ik}, v_{ik})\}$ for each $k \in [|P_i|]$. Then, for each $k \in [|P_i|]$ and the graph $G$ with edges $\bigcup_i P_{i;k}$, $f^{\mathrm{BD}}_{p;\mathcal{P}}(G) = \prod_i (p(u_{ik}, v_{ik}) - p(u_{i,k+1}, v_{i,k+1}))$, where we take $p(u_{i,|P_i|+1}, v_{i,|P_i|+1}) = 0$. For any other graph $G$, $f^{\mathrm{BD}}_{p;\mathcal{P}}(G) = 0$.

There are two basic properties of binding: *(i)* binding is *correct*, i.e., generates EPGMs, and *(ii)* binding improves subgraph densities upon EIGMs.

---

[6]Notably, we shall also empirically show that binding maintains (or even improves) the generation quality w.r.t. several different graph metrics, including but not limited to degrees and diameters (see Section 6.3).

**Proposition 5.2.** *Algorithm 1 with input $p$ (and any $\mathcal{P}$) produces an EPGM w.r.t. $p$.*

**Proposition 5.3.** *Binding produces higher or equal subgraph densities, compared to the corresponding EIGMs.*

*Remark* 5.4. There are EPGMs with lower subgraph densities, which are against our motivation to improve upon EIGMs w.r.t. subgraph densities and are out of this work's scope. That said, they may be useful in scenarios where dense subgraphs are unwanted, e.g., disease control.

With binding, we can construct EPGMs with different levels of edge dependency by different ways of binding the node pairs. Let us first study two extreme cases.

**Minimal binding.** EIGMs are the case with minimal binding, i.e., without binding, where the partition contains only sets of a single pair, i.e., $\mathcal{P} = \{\{(u, v)\} \colon u, v \in V\}$.

**Maximal binding.** Maximal binding corresponds to the case with $\mathcal{P} = \{\binom{V}{2}\}$, i.e., all the pairs are bound together. It achieves the upper bound of subgraph densities, i.e., the maximal edge-group probabilities in Property 4.4 (see Lemma B.2), as mentioned in Remark 4.5.

## 5.2 Local binding: Flexible and tractable spectrum between two extremes

Building upon the general framework introduced in Section 5.1, we propose practical binding algorithms. Intuitively, the more pairs we bind together, the higher subgraph densities we have. Between minimal binding (i.e., EIGMs) and maximal binding that achieves the upper bound of subgraph densities, we can have a *flexible* spectrum. However, the number of possible partitions of node pairs $\binom{V}{2}$ grows exponentially w.r.t. $|V|$. Hence, we propose to introduce edge dependency without explicit partitions. Specifically, we propose *local binding*, where we repeatedly sample node groups,[7] and bind pairs between each sampled node group together. Pairs between the same node group are structurally related, compared to pairs sharing no common nodes.

**Real-world motivation.** In social networks, each group "bound together" can represent a group interaction, e.g., an offline social event (meeting, conference, party) or an online social event (group chat, Internet forum, online game). In such social events, people gather together, and the communications/relations between them likely co-occur. At the same time, not all people in such events would necessarily communicate with each other, e.g., some people are more familiar with each other. This is the point of considering binding with various edge probabilities (instead of just inserting cliques). In general, group interactions widely exist in graphs in different domains, e.g., social networks (Felmlee & Faris, 2013), biological networks (Naoumkina et al., 2010), and web graphs (Dourisboure et al., 2009). See Appendix D.4 for more discussions.

---

**Algorithm 2:** Local binding

**Input** : (1) $p \colon \binom{V}{2} \to [0, 1]$: edge probabilities;
    (2) $g \colon V \to [0, 1]$: node-sampling probabilities;
    (3) $R$: maximum number of rounds for binding
**Output:** $G$: generated graph

1   $\mathcal{P} \leftarrow \emptyset; i_{round} \leftarrow 0; P_{rem} \leftarrow \binom{V}{2}$   ▷ Initialization
2   **for** $i_{round} = 1, 2, \ldots, R$ **do**
3    **if** $P_{rem} = \emptyset$ **then**
4     **break**      ▷ Pairs exhausted
5    $i_{round} \leftarrow i_{round} + 1$
6    sample $V_s \subseteq V$ with $\Pr[v \in V_s] = g(v)$
    independently
7    $P_s \leftarrow \binom{V_s}{2} \cap P_{rem}$
8    **if** $P_s \neq \emptyset$ **then**
9     $\mathcal{P} \leftarrow \mathcal{P} \cup \{P_s\}$
10     $P_{rem} \leftarrow P_{rem} \setminus P_s$

11   $\mathcal{P} \leftarrow \mathcal{P} \cup \{\{(u, v)\} \colon (u, v) \in P_{rem}\}$
12   **return** *the output of Algorithm 1 with inputs $p$ and $\mathcal{P}$*

---

In Algorithm 2, we repeatedly sample a subset of nodes (Line 6) and group the ungrouped pairs between the sampled nodes (Line 9). We maintain $P_{rem}$ to ensure disjoint partitions (Lines 7 and 10). For practical usage, we consider a limited number (i.e., $R$) of rounds for binding (Line 2) otherwise it may take a long time to exhaust all the pairs. Algorithm 2 is also a probabilistic process, and we use $f_{p;g,R}^{\mathrm{LB}}$ to denote the corresponding RGM, i.e., $f_{p;g,R}^{\mathrm{LB}}(G) = \Pr[\text{Algorithm 2 outputs } G \text{ with inputs } p, g, \text{ and } R]$. As a special case of binding, local binding is also correct, i.e., generates EPGMs.

**Proposition 5.5.** *Algorithm 2 with input $p$ (and any $g$ and $R$) produces an EPGM w.r.t. $p$.*

*Remark* 5.6. We introduce node-sampling probabilities (i.e., $g$) to sample node groups with better tractability, without explicit partitions. With higher node-sampling probabilities, larger node groups

---

[7]We use independent *node* sampling (yet still with *edge* dependency), which is simple, tractable, and works empirically well in our experiments. See Appendix D.3 for more discussions.

are bound together, and the generated graphs are expected to have higher subgraph densities. Specifically, local binding forms a spectrum between the two extreme cases. When $g(v) \equiv 0$, local binding reduces to minimal binding, i.e., EIGMs. When $g(v) \equiv 1$, it reduces to maximal binding.

**Theorem 5.7** (Time complexity of graph generation with local binding). *Given* $p \colon \binom{V}{2} \to [0,1]$, $g \colon V \to [0,1]$, *and* $R \in \mathbb{N}$, $f_{p;g,R}^{LB}$ *generates a graph in* $O(R \left( \sum_{v \in V} g(v) \right)^2 + |V|^2)$ *time with high probability,*[8] *with the worst case* $O(R|V|^2)$.

We derive tractability results of local binding on the closed-form expected number of motifs (i.e., induced subgraphs; see Section 2). For this, we derive the probabilities of all the possible motifs for each node group, then we can compute the expected number of motifs by taking the summation over all different node groups, which can be later used for parameter fitting (see Section 5.4).

**Theorem 5.8** (Tractable motif probabilities with local binding). *For any* $p \colon \binom{V}{2} \to [0,1]$, $g \colon V \to [0,1]$, $R \in \mathbb{N}$, *and* $V' = \{u, v, w\} \in \binom{V}{3}$, *we can compute the closed-form* $\Pr_{f_{p;g,R}^{LB}}[E(G[V']) = E^*], \forall E^* \subseteq \binom{V'}{2}$, *as a function w.r.t.* $p$, $g$, *and* $R$ *(the detailed formulae are in Appendix B.3).*

*Proof sketch.* See Appendix B.3 for the full proof and the detailed formulae. Higher $p$ and $g$ values give higher clustering. The choice of $R$ is mainly for controlling the running time. □

*Remark* 5.9. Having closed-form formulae of motif probabilities allows us to estimate the output and fit the parameters of RGMs (see Section 5.4). Theorem 5.8 can be extended to larger $|V'|$ with practical difficulties from the increasing sub-cases as motif size increases. See Appendix B.3.

**Theorem 5.10** (Time complexity of computing motif probabilities with local binding). *Computing* $\Pr_{f_{p;g,R}^{LB}}[E(G[V']) = E^*]$ *takes* $O(|V|^3)$ *in total for all* $E^* \subseteq \binom{V'}{2}$ *and* $V' \in \binom{V}{3}$.

## 5.3 PARALLEL BINDING: THE PARALLELIZABLE ICING ON THE CAKE

In local binding, the sampling order matters, i.e., later rounds are affected by earlier rounds. Specifically, if one pair is already determined in an early round, even if it is sampled again in later rounds, its (in)existence cannot be changed. This property hinders the parallelization of the binding process and the derivation of tractability analyses. This property also implies that each pair can only be bound together once, entailing less flexibility in the group interactions.

We thus propose a more flexible and naturally *parallelizable* binding algorithm, *parallel binding*. Specifically, we consider the probabilistic process in Algorithm 3, and let $f_{p;g,R}^{PB}$ denote the corresponding RGM defined by $f_{p;g,R}^{PB}(G) = \Pr[\text{Algorithm 3 outputs } G \text{ with inputs } p, g, \text{ and } R]$.

---

**Algorithm 3:** Parallel binding

**Input** : (1) $p \colon \binom{V}{2} \to [0,1]$: edge probabilities;
(2) $g \colon V \to [0,1]$: node-sampling probabilities;
(3) $R$: the number of rounds for binding

**Output:** $G$: generated graph

1   $E \leftarrow \emptyset$          ▷ Initialization

2   $r(u,v) \leftarrow \min(\frac{1-(1-p(u,v))^{1/R}}{g(u)g(v)}, 1), \forall u, v \in V$

3   $p_{rem}(u,v) \leftarrow \max(1 - \frac{1-p(u,v)}{(1-g(u)g(v))^R}, 0), \forall u, v \in V$

4   **for** $i_{round} = 1, 2, \dots, R$ **do**

5      sample $V_s \subseteq V$ with $\Pr[v \in V_s] = g(v)$ independently

6      $E \leftarrow E \cup \texttt{binding}(r, \binom{V_s}{2})$    ▷ See Alg. 1

7   **for** $(u,v) \in \binom{V}{2}$ *s.t.* $p_{rem}(u,v) > 0$ **do**

8      sample a random variable $s \sim \mathcal{U}(0,1)$

9      **if** $s \leq p_{rem}(u,v)$ **then**

10        $E \leftarrow E \cup \{(u,v)\}$

11   **return** $G = (V, E)$

---

The high-level idea is to make each round of binding probabilistically equivalent (see Lines 4 to 6). Specifically, in each round, we insert edges with low probabilities (compared to the ones in $p$) while maintaining the final individual edge probabilities, by the calculation of $r$ and $p_{rem}$ at Lines 2 and 3. We can straightforwardly parallelize the rounds by, e.g., multi-threading.

Although parallel binding is algorithmically different from (local) binding (e.g., no partition is used), it shares many theoretical properties with local binding. Specifically, Proposition 5.5, Remark 5.6, Theorem 5.7, Theorem 5.8, Remark 5.9, and Theorem 5.10 also apply to parallel binding. This implies that we maintain (or even improve; see Remark 5.11) correctness, tractability, flexibility, and efficiency when using parallel binding instead of local binding. See Appendix B.4 for the formal statements and proofs.

---

[8]That is, $\lim_{|V| \to \infty} \Pr[\text{it takes } O(R \left( \sum_{v \in V} g(v) \right)^2 + |V|^2)] = 1$.

*Remark* 5.11. We also derive tractability results of parallel binding on the expected number of (non-)isolated nodes. It is much more challenging to derive such results for local binding due to the properties mentioned above, i.e., later rounds are affected by earlier rounds. Since our main focus is on subgraph densities, see Appendix C for all the analysis regarding (non-)isolated nodes.

## 5.4 EFFICIENT PARAMETER FITTING WITH NODE EQUIVALENCE

Efficient evaluation of the fitting objective is important. A key challenge is that the naive computation takes $O(|V|^3)$ time in total by considering all $O(|V|^3)$ different possible node groups $V'$ (see Theorems 5.8 and B.5)). We aim to improve the speed of computing the tractability results by considering *node equivalence* w.r.t. motif probabilities in various edge-probability models. Equivalent nodes form equivalent node groups, which reduces the number of distinct node groups to calculate. **Erdős-Rényi (ER) model.** The ER model (Erdős & Rényi, 1959) outputs uniform edge probabilities, and all the nodes are equivalent. Hence, we set all the node-sampling probabilities identical, i.e., $g(v) = g_0, \forall v \in V$ for a single parameter $g_0 \in [0, 1]$. As mentioned in Section 3.2, the ER model is the only case with node exchangeability, and the exchangeability is preserved with binding since the nodes are also treated symmetrically for binding.

**Lemma 5.12** (Reduced time complexity with ER)**.** *For ER, the time complexities of computing* 3-*motif probabilities can be reduced from* $O(|V|^3)$ *to* $O(1)$.

**Chung-Lu (CL) model.** The CL model (Chung & Lu, 2002) outputs edge probabilities with expected degrees $D = (d_1, d_2, \ldots, d_n)$, and nodes with the same degree are equivalent. We set node-sampling probabilities as a function of degree with $k_{deg}$ parameters, where $k_{deg} := |\{d_1, d_2, \ldots, d_n\}|$.

**Lemma 5.13** (Reduced time complexity with CL)**.** *For CL, the time complexities of computing* 3-*motif probabilities can be reduced from* $O(|V|^3)$ *to* $O(k_{deg}^3)$.

**Stochastic block (SB) model** The SB model (Holland et al., 1983) outputs edge probabilities with each node assigned to a block (i.e., a group), and nodes partitioned in the same block are equivalent. Hence, we set the node-sampling probabilities as a function of the block index, with the number of parameters equal to the number of blocks.

**Lemma 5.14** (Reduced time complexity with SB)**.** *For SB, the time complexities of computing* 3-*motif probabilities can be reduced from* $O(|V|^3)$ *to* $O(c^3)$, *where* $c$ *is the total number of blocks.*

**Stochastic Kronecker (KR) model** With a (commonly used 2-by-2) seed matrix $\theta \in [0, 1]^{2 \times 2}$ and $k_{KR} \in \mathbb{N}$, the KR model (Leskovec et al., 2010) outputs edge probabilities as the $k_{KR}$-th Kronecker power of $\theta$. In KR, each node $i \in [2^{k_{KR}}]$ is associated with a binary node label of length $k_{KR}$, i.e., the binary representation of $i - 1$. Nodes with the same number of ones in their binary node labels are equivalent.[9] Hence, we set node-sampling probabilities as a function of the number of ones in the binary representation, with $k_{KR} + 1$ parameters.

**Lemma 5.15** (Reduced time complexity with KR)**.** *For KR, the time complexities of computing* 3-*motif probabilities can be reduced from* $O(|V|^3)$ *to* $O(k_{KR}^7)$.

**Note.** See Appendix B.5 for more details about parameter fitting, e.g., formal definitions of the models and the details of node equivalence.

## 6 EXPERIMENTS

In this section, we empirically evaluate EPGMs with our binding schemes and show the superiority of realization schemes beyond edge independency. Specifically, we show the following two points:

- **(P1)** When we use our tractability results to fit the parameters of EPGMs, we improve upon EIGMs and reproduce high triangle densities, and thus produce high clustering, which is a common pattern in real-world graphs; this also validates the correctness of our tractability results and algorithms.
- **(P2)** We can reproduce other common patterns, e.g., power-law degrees and small diameters, especially when the corresponding EIGMs are able to do so; this shows that improving EIGMs w.r.t. clustering by binding does not harm the generation quality w.r.t. other common patterns.

---

[9]The equivalence in KR is slightly weaker than that in the other three models. This is why the reduced time complexity is $O(k_{KR}^7)$ instead of $O(k_{KR}^3)$. See Appendix B.5.4 for more details.

Table 1: The clustering metrics of generated graphs. The number of triangles (△) is normalized. For each dataset and each model, the best result is in bold and the second best is underlined. AR represents average ranking. The statistics are averaged over 100 random trails. See Table 7 in Appendix E.2 for the full results with standard deviations. **Our binding schemes (LOCLBDG and PARABDG) are consistently and clearly beneficial for improving clustering, and generating graphs with close-to-ground-truth clustering metrics.**

| dataset | | Hams | | | Fcbk | | | Polb | | | Spam | | | Cepg | | | Scht | | | AR over dataset | | |
|---|---|---|---|---|---|---|---|---|---|---|---|---|---|---|---|---|---|---|---|---|---|---|
| metric | | △ | GCC | ALCC | △ | GCC | ALCC | △ | GCC | ALCC | △ | GCC | ALCC | △ | GCC | ALCC | △ | GCC | ALCC | △ | GCC | ALCC |
| model | GROUNDT | 1.00 | 0.23 | 0.54 | 1.00 | 0.52 | 0.61 | 1.00 | 0.23 | 0.32 | 1.00 | 0.14 | 0.29 | 1.00 | 0.32 | 0.45 | 1.00 | 0.38 | 0.35 | N/A | N/A | N/A |
| ER | EDGEIND | 0.01 | 0.01 | 0.01 | 0.01 | 0.01 | 0.01 | 0.03 | 0.02 | 0.02 | 0.01 | 0.00 | 0.00 | 0.04 | 0.03 | 0.03 | 0.03 | 0.03 | 0.03 | 3.0 | 2.7 | 2.5 |
| | LOCLBDG | 1.00 | 0.32 | 0.24 | 1.01 | 0.45 | 0.22 | 0.95 | 0.34 | 0.25 | 0.99 | 0.34 | 0.23 | 1.02 | 0.40 | 0.26 | 1.01 | 0.42 | 0.25 | 1.7 | 1.3 | 1.3 |
| | PARABDG | 0.99 | 0.39 | 0.64 | 1.00 | 0.57 | 0.81 | 1.02 | 0.41 | 0.66 | 0.99 | 0.40 | 0.66 | 0.97 | 0.51 | 0.75 | 0.99 | 0.56 | 0.79 | 1.3 | 2.0 | 2.2 |
| CL | EDGEIND | 0.30 | 0.07 | 0.06 | 0.12 | 0.06 | 0.06 | 0.79 | 0.18 | 0.17 | 0.50 | 0.07 | 0.06 | 0.68 | 0.23 | 0.22 | 0.64 | 0.24 | 0.23 | 3.0 | 3.0 | 2.5 |
| | LOCLBDG | 0.99 | 0.17 | 0.26 | 1.03 | 0.26 | 0.30 | 1.00 | 0.21 | 0.34 | 1.03 | 0.12 | 0.26 | 1.00 | 0.29 | 0.43 | 1.04 | 0.32 | 0.47 | 1.7 | 1.8 | 1.5 |
| | PARABDG | 1.00 | 0.18 | 0.47 | 1.01 | 0.34 | 0.63 | 1.01 | 0.22 | 0.47 | 1.01 | 0.13 | 0.44 | 1.00 | 0.31 | 0.58 | 1.14 | 0.29 | 0.61 | 1.3 | 1.2 | 2.0 |
| SB | EDGEIND | 0.26 | 0.08 | 0.04 | 0.15 | 0.14 | 0.08 | 0.48 | 0.14 | 0.16 | 0.53 | 0.09 | 0.04 | 0.66 | 0.26 | 0.20 | 0.64 | 0.27 | 0.13 | 3.0 | 3.0 | 3.0 |
| | LOCLBDG | 1.04 | 0.22 | 0.24 | 0.93 | 0.43 | 0.33 | 0.99 | 0.24 | 0.35 | 0.98 | 0.15 | 0.22 | 0.99 | 0.32 | 0.41 | 1.03 | 0.35 | 0.39 | 1.7 | 1.2 | 1.3 |
| | PARABDG | 0.99 | 0.24 | 0.52 | 1.03 | 0.53 | 0.56 | 1.01 | 0.18 | 0.25 | 0.99 | 0.16 | 0.36 | 1.05 | 0.33 | 0.36 | 0.97 | 0.34 | 0.44 | 1.3 | 1.8 | 1.7 |
| KR | EDGEIND | 0.18 | 0.04 | 0.06 | 0.05 | 0.04 | 0.04 | 0.10 | 0.04 | 0.07 | 0.06 | 0.01 | 0.03 | 0.13 | 0.07 | 0.12 | 0.03 | 0.03 | 0.05 | 3.0 | 3.0 | 3.0 |
| | LOCLBDG | 1.09 | 0.15 | 0.23 | 0.93 | 0.24 | 0.27 | 1.06 | 0.14 | 0.23 | 0.94 | 0.12 | 0.19 | 0.99 | 0.17 | 0.31 | 1.44 | 0.18 | 0.28 | 2.0 | 2.0 | 1.7 |
| | PARABDG | 1.00 | 0.17 | 0.39 | 0.97 | 0.35 | 0.60 | 0.94 | 0.22 | 0.42 | 1.05 | 0.16 | 0.38 | 1.00 | 0.28 | 0.46 | 1.07 | 0.35 | 0.58 | 1.0 | 1.0 | 1.3 |
| AR over models | EDGEIND | 3.0 | 3.0 | 3.0 | 3.0 | 3.0 | 3.0 | 3.0 | 3.0 | 2.5 | 3.0 | 2.5 | 2.8 | 3.0 | 3.0 | 3.0 | 3.0 | 3.0 | 2.3 | 3.0 | 2.9 | 2.8 |
| | LOCLBDG | 1.8 | 1.5 | 2.0 | 2.0 | 2.0 | 2.0 | 1.5 | 1.5 | 1.0 | 2.0 | 1.8 | 1.3 | 1.5 | 1.5 | 1.3 | 1.8 | 1.3 | 1.3 | 1.8 | 1.6 | 1.5 |
| | PARABDG | 1.3 | 1.5 | 1.0 | 1.0 | 1.0 | 1.0 | 1.5 | 1.5 | 2.5 | 1.0 | 1.8 | 2.0 | 1.5 | 1.5 | 1.8 | 1.3 | 1.8 | 2.5 | 1.3 | 1.5 | 1.8 |

## 6.1 EXPERIMENTAL SETTINGS

**Datasets.** We use six real-world datasets: (1) social networks *hamsterster (Hams)* and *facebook (Fcbk)*, (2) web graphs *polblogs (Polb)* and *spam (Spam)*, and (3) biological graphs *CE-PG (Cepg)* and *SC-HT (Scht)*. See Table 6 in Appendix E.1 for the statistics of the datasets.

**Models.** We consider the four edge-probability models analyzed in Section 5.4: the Erdős-Rényi (ER) model, the Chung-Lu (CL) model, the stochastic block (SB) model, and the stochastic Kronecker (KR) model. Given an input graph, we fit each model to the graph and obtain the output edge probabilities (see Appendix B.5 for more details).

**Realization methods.** We compare three realization methods: EIGMs (EDGEIND), and EPGMs with local binding (LOCLBDG) and with parallel binding (PARABDG).

**Fitting.** Since our main focus is to improve clustering, in our main experiments, we use the number of triangles, an important indicator of clustering (Tsourakakis et al., 2009; Kolda et al., 2014a), as the objective of the fitting algorithms. We use gradient descent to optimize parameters. In the main experiments, the edge probabilities are fixed as those output by the edge-probability models, while we also consider joint optimization of edge probabilities and node-sampling probabilities (see Section 6.5). See Appendix E.1 for the detailed experimental settings. Instead of fitting specific graphs, it is also possible to use EPGMs with binding to generate graphs "from scratch" with different levels of clustering by directly setting the parameters. See Appendix E.7 for more discussions and results.

## 6.2 P1: EPGMS REPRODUCE HIGH CLUSTERING (TABLE 1)

EPGMs with binding reproduce high clustering in real-world graphs. In Table 1, for each dataset and each model, we compare three clustering-related metrics, the number of triangles (△), the global clustering coefficient (GCC), and the average local clustering coefficient (ALCC), in the ground-truth (GROUNDT) graph and the graphs generated with each realization method. For each dataset and each model, we compute the ranking of each method according to the absolute error w.r.t. each metric. We also show the average rankings (ARs) over datasets and models. The statistics are averaged over 100 generated graphs. See Appendix E.2 for the full results with standard deviations. The number of triangles, which is the objective of our fitting algorithms, can be almost perfectly preserved by both LOCLBDG and PARABDG, showing the correctness and effectiveness of our algorithms. Notably, as Theorem 3.3 imply, EIGMs often fail to generate graphs with enough triangles. GCC and ALCC are also significantly improved (upon EIGMs) in most cases, while PARABDG has noticeably higher ALCC than LOCLBDG. In some rare cases, PARABDG generates graphs with exceedingly high GCC and/or ALCC and have higher absolute errors compared to EIGMs.

## 6.3 P2: EPGMS REPRODUCE REAL-WORLD DEGREES AND DISTANCES (FIGURE 1)

EPGMs with binding (LOCLBDG and PARABDG) also reproduce other common patterns in real-world graphs. In Figure 1, for each dataset (each column) and each model (each row), we compare the degree distributions and distance distributions in the ground-truth graph and the graphs generated with each realization method. Specifically, for each realization method, we count the number of nodes with degree at least $k$ for each $k \in \mathbb{N}$ and count the number of pairs in the largest connected component with distance at least $d$ for each $d \in \mathbb{N}$ in each generated graph, and take the average number over 100 generated graphs. See Appendix E.3 for the formal definitions and full results.

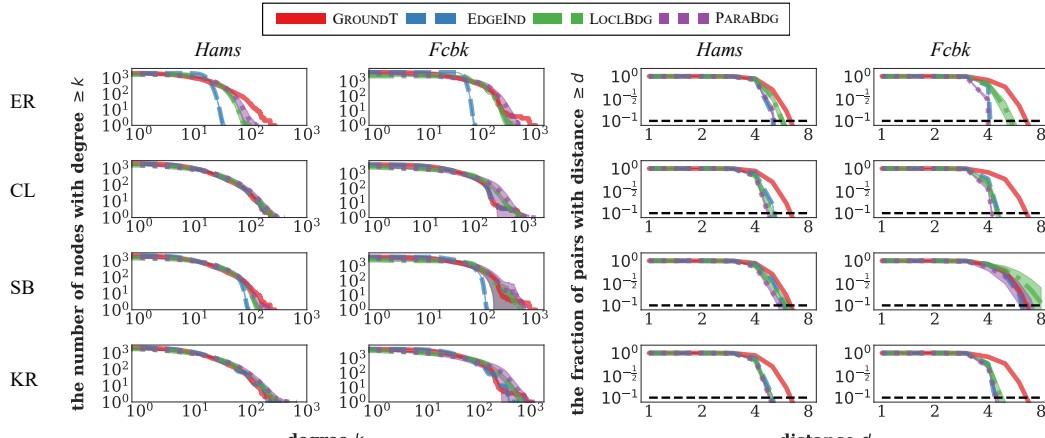

Figure 1: The degree (left) and distance (right) distributions of generated graphs. Each shaded area represents one standard deviation. The plots are in a log-log scale. **Our binding schemes (LOCLBDG and PARABDG) do not negatively affect degree or distance distributions, and provide improvements sometimes (e.g., for the ER model).**

EPGMs with binding generate graphs with common patterns: power-law degrees and small diameters (i.e., small distances). Both schemes (LOCLBDG and PARABDG) perform comparably well while LOCLBDG performs noticeably better with ER and PARABDG performs noticeably better with KR. Importantly, when the edge probabilities output power-law expected degrees (e.g., CL and KR), the degree distributions are well preserved with binding. Edge-independent ER cannot generate power-law degrees (Bollobás & Riordan, 2003), and binding alleviates this problem.

**Other graph metrics.** Notably, with binding, the generated graphs are overall closer to the ground truth w.r.t. some other graph metrics: modularity, core numbers, conductance, average vertex/edge betweenness, and natural connectivity. See Appendix E.3 for more details.

### 6.4 GRAPH GENERATION SPEED (TABLE 2)

In Table 2, we compare the running time of graph generation (averaged over 100 random trials) using EDGEIND,[10] LO-CLBDG, PARABDG, and serialized PARABDG without paralleliza-tion (PARABDG-s) with the stochastic Kronecker (KR) model.

Among the competitors, EDGEIND is the fastest with the simplest algorithmic nature. Between the two binding schemes, PARABDG is noticeably faster than LOCLBDG, and is even faster with paral-lelization. Fitting the same number of triangles, PARABDG usually requires lower node-sampling probabilities and thus deals with fewer

Table 2: The time (in seconds) for graph generation with dif-ferent realization methods.

| | Hams | Fcbk |
|---|---|---|
| EDGEIND | 0.1 | 0.1 |
| LOCLBDG | 4.7 | 49.0 |
| PARABDG | 0.3 | 1.7 |
| PARABDG-S | 3.2 | 12.6 |

pairs in each round, and is thus faster even when serialized. We also conduct scalability experiments by upscaling the input graph. With 32GB RAM, all the proposed algorithms can run with 128,000 nodes. See Appendix E.4 for more detailed discussions and results.

### 6.5 JOINT OPTIMIZATION OF EDGE AND NODE-SAMPLING PROBABILITIES (TABLE 3)

In addition to optimizing node-sampling probabilities for given edge probabilities, we can also jointly optimize both kinds of probabilities. In Table 3, we compare the ground-truth clustering and that gen-erated by EPGMs using three variants of parallel binding: (1) PARABDG with the number of triangles as the objective (the one used in Table 1), (2) PARABDG-W with the numbers of triangles and wedges as the objective (given edge probabilities), (3) PARABDG-JW jointly optimizing both

Table 3: The clustering metrics of the graphs generated by three variants of parallel binding. The number of triangles ($\triangle$) is normalized. For each dataset and each model, the best result is in bold, and the second best is underlined. **Joint optimization further enhances the power of our binding scheme PARABDG to reproduce graph patterns.**

| dataset | Hams | | | Fcbk | | |
|---|---|---|---|---|---|---|
| metric | $\triangle$ | GCC | ALCC | $\triangle$ | GCC | ALCC |
| GROUNDT | 1.000 | 0.229 | 0.540 | 1.000 | 0.519 | 0.606 |
| PARABDG | 0.997 | 0.165 | 0.394 | 0.971 | 0.347 | **0.605** |
| PARABDG-W | 0.964 | 0.176 | 0.260 | 1.021 | 0.408 | 0.458 |
| PARABDG-JW | **0.999** | **0.230** | 0.448 | 1.018 | **0.521** | 0.644 |

kinds of probabilities, with the numbers of triangles and wedges as the objective.

On both *Hams* and *Fcbk*, PARABDG and PARABDG-W can well fit the number of triangles but have noticeable errors w.r.t. the number of wedges (and thus GCC), while PARABDG-JW with joint

---

[10]We use `krongen` in SNAP (Leskovec & Sosič, 2016), which is parallelized and optimized for KR.

optimization accurately fits both triangles and wedges. On the other datasets, the three variants perform similarly well because PARABDG already preserves both triangles and wedges well, and there is not much room for improvement. Notably, with joint optimization, the degree and distance distributions are still well preserved (see Appendix E.5 for more details).

### 6.6 COMPARISON WITH EDGE-DEPENDENT RGMS AND ADVANCED EIGMS (TABLE 4)

We test other edge-dependent RGMs: preferential attachment models (PA; Barabási & Albert (1999)) and random geometric graphs (RGG; Penrose (2003)). We fit them to the numbers of nodes and edges of each input graph. PA fails to generate high clustering. For RGG, we often need dimension $d = 1$ (the smallest dimension gives the highest clustering) to generate enough triangles, while the GCC and ALCC are too high (they are only determined by the dimension $d$). Also, as discussed in Section 3.2, closed-form tractability results on subgraph densities are not

Table 4: The clustering metrics and overlap (lower the better) of the graphs generated by binding and other models. For each dataset and each model, the best result is in bold, and the second best is underlined. **Overall, binding achieves promising performance in generating high-clustering graphs, with high variability.**

| dataset | Hams | | | | Fcbk | | | |
|---|---|---|---|---|---|---|---|---|
| metric | $\triangle$ | GCC | ALCC | overlap | $\triangle$ | GCC | ALCC | overlap |
| GROUNDT | 1.000 | 0.229 | 0.540 | N/A | 1.000 | 0.519 | 0.606 | N/A |
| LOCLBDG-CL | 0.992 | 0.165 | 0.255 | 5.8% | 1.026 | 0.255 | 0.305 | 6.3% |
| PARABDG-CL | **1.000** | **0.185** | 0.471 | 5.9% | **1.006** | 0.336 | 0.626 | 6.2% |
| PA | 0.198 | 0.049 | 0.049 | 4.7% | 0.120 | 0.061 | 0.061 | 6.2% |
| RGG ($d = 1$) | 1.252 | 0.751 | 0.751 | **0.8%** | 0.607 | 0.751 | 0.752 | **1.1%** |
| RGG ($d = 2$) | 1.011 | 0.595 | 0.604 | 0.8% | 0.492 | 0.596 | 0.607 | 1.1% |
| RGG ($d = 3$) | 0.856 | 0.491 | 0.513 | 0.8% | 0.421 | 0.494 | 0.518 | 1.1% |
| BTER | 0.991 | 0.290 | **0.558** | 53.8% | 0.880 | 0.525 | **0.605** | 68.0% |
| LFR ($\mu = 0.0$) | 1.140 | 0.262 | 0.546 | 43.5% | N/A | N/A | N/A | N/A |
| LFR ($\mu = 0.5$) | 0.296 | 0.068 | 0.081 | 13.4% | 0.161 | 0.084 | 0.120 | 17.0% |
| LFR ($\mu = 1.0$) | 0.197 | 0.045 | 0.047 | 7.0% | 0.105 | 0.055 | 0.059 | 6.7% |

unavailable for PA and RGG. See Appendix E.2 for more details. As discussed in Section 3.1, some existing methods shift edge probabilities, and they are essentially EIGMs with an inevitable trade-off between variability and the ability to generate high clustering (see Theorem 3.3). We test the block two-level Erdős-Rényi (BTER) model (Kolda et al., 2014b) that essentially uses a mixture of multiple Chung-Lu models to generate high clustering. Similarly, the Lancichinetti-Fortunato-Radicchi (LFR) model (Lancichinetti et al., 2008) generates graphs with community structures by shifting edge probabilities to intra-community pairs on top of Chung-Lu. We empirically validate that EPGMs with binding (we report the results based on Chung-Lu; one may achieve even better performance with binding based on other edge-probability models, as shown in Table 1) achieve comparable performance in generating high-clustering graphs, with much higher variability (i.e., low overlap; recall that high variability is important for RGMs; see Definition 3.1 and Remark 3.2). See Table 4 for the results on *Hams* and *Fcbk*, and see Appendix E.6 for more details with full results and discussions on deep graph generative models (Rendsburg et al., 2020; Simonovsky & Komodakis, 2018; You et al., 2018).

**Extra experimental results.** Due to the page limit, the full results are in Appendix E. Our fitting algorithms also assign different node-sampling probabilities to different nodes (See Appendix E.1). Moreover, as mentioned in Remark 5.11, for parallel binding, we can fit and control the number of (non-)isolated nodes; see Appendix C for the theoretical analyses and experimental results.

## 7 CONCLUSION AND DISCUSSIONS

In this work, we show that realization beyond edge independence can better reproduce common patterns while ensuring high tractability and variability. We formally define EPGMs and show their basic properties (Section 4). Notably, even with edge dependency, EPGMs maintain the same variability (Property 4.7). We propose a pattern-reproducing, tractable, and flexible realization framework called *binding* (Algorithm 1) with two practical variants: local binding (Algorithm 2) and parallel binding (Algorithm 3). We derive tractability results (Theorems 5.8 and B.5) on the closed-form subgraph densities, and propose efficient parameter fitting (Section 5.4; Lemmas 5.12-5.15). We conduct extensive experiments to show the empirical power of EPGMs with binding (Section 6).
**Limitations and future directions.** EPGMs with binding generate more isolated nodes than EIGMs due to higher variance. Fortunately, we can address the limitation by fitting and controlling the number of isolated nodes with the tractability results, as mentioned in Remark 5.11. The performance of EPGMs depends on both the underlying edge probabilities and the way to realize (i.e., sample from) them. In this work, we focus on the latter, while finding valuable edge probabilities is an independent problem. Notably, as shown in Section 6.5, it is possible to jointly optimize both edge probabilities and their realization. As discussed in Remark 5.4, binding only covers a subset of EPGMs, and we will explore the other types of EPGMs (e.g., EPGMs with lower subgraph densities) in the future. Combining binding with other mechanisms in existing edge-dependent RGMs to create even stronger RGMs is another interesting future direction.

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

## A    FLOWCHART

Below, we provide a flowchart of this work, summarizing the main ideas and contents.

Target: To find random graph models that have high tractability and generate graphs with common patterns in real-world graphs and high variability

Background: EIGMs with edge-independent realization methods have high tractability but cannot generate graphs with high triangle densities

Idea: To consider edge-dependent realization methods

We define the concept of EPGMs (Section 4) but considering all the possible cases is impractical

Idea: To find sub-cases with meaningful realization methods, specifically aiming to improve triangle densities

We propose specific realization schemes called binding and derive the tractability results on motif probabilities (Section 5) but we want to further improve the efficiency of computing motif probabilities in practice

Idea: To reduce the number of distinct node groups for calculation by considering node equivalence

We analyze the node equivalence in various edge-probabilities models and derive the reduced time complexities for computing motif probabilities (Section 5.4)

## B    PROOFS

In this section, we show the proofs of our theoretical results.

### B.1    EPGMs

**Proposition 4.2** (EIGMs are special EPGMs). *For any $p$, the EIGM w.r.t. $p$ is an EPGM w.r.t. $p$, i.e., $f_p^{EI} \in \mathcal{F}(p)$.*

*Proof.* By the definition of EIGMs,

$$
\Pr_{f_p^{\mathrm{EI}}}[(u,v)]
$$
$$
= \sum_{G \in \mathcal{G}(V)} f_p^{\mathrm{EI}}(G)\mathbf{1}[(u,v) \in G]
$$
$$
= \sum_{(u,v) \in G \in \mathcal{G}(V)} f_p^{\mathrm{EI}}(G)
$$
$$
= \sum_{(u,v) \in G \in \mathcal{G}(V)} p(u,v) \prod_{(u,v) \neq (u^+,v^+) \in G} p(u^+,v^+) \prod_{(u^-,v^-) \notin G} (1 - p(u^-,v^-))
$$
$$
= p(u,v), \forall u,v,
$$

completing the proof.    $\square$

**Proposition 4.3** (EPGMs are general). *For any $f \colon \mathcal{G}(V) \to [0,1]$, there exists $p \colon \binom{V}{2} \to [0,1]$ such that $f \in \mathcal{F}(p)$.*

*Proof.* Let $p\colon \binom{V}{2} \to [0,1]$ be that $p(u,v) = \Pr_f[(u,v)], \forall u,v \in V$, then by Definition 4.1, $f \in \mathcal{F}(p)$. $\square$

**Proposition 4.4** (Upper bound of edge-group probabilities)**.** *For any* $p\colon \binom{V}{2} \to [0,1]$ *and any edge group* $P \subseteq \binom{V}{2}$, $\Pr_f[P \subseteq E(G)] \le \min_{(u,v)\in P} p(u,v), \forall f \in \mathcal{F}(p)$.

*Proof.* By definition, $\Pr_f[(u,v)] = p(u,v), \forall (u,v)$. Hence,

$$\Pr_f[P \subseteq E(G)] = \Pr_f[\bigwedge_{(u,v)\in P} (u,v) \in G]$$

$$\le \min_{(u,v)\in P} \Pr_f[(u,v)]$$

$$= \min_{(u,v)\in P} p(u,v),$$

where we have used the fact that $\bigwedge_{(u,v)\in P}(u,v) \in G$ is a subevent of $(u,v) \in G$ for any $(u,v) \in P$. $\square$

**Proposition 4.7** (EPGMs have constant expected degrees and overlap)**.** *For any* $p\colon \binom{V}{2} \to [0,1]$, *the expected degree of each node and the overlap of all the EPGMs w.r.t.* $p$ *are constant. Specifically,* $\mathbb{E}_f[d(v;G)] = \sum_{u\in V} p(u,v)$ *and* $\mathrm{Ov}(f) = \frac{\sum_{u,v\in V} p^2(u,v)}{\sum_{u,v\in V} p(u,v)}, \forall f \in \mathcal{F}(p)$.

*Proof.* By linearity of expectation,

$$\mathbb{E}_f[d(v;G)] = \sum_{u\in V} \Pr[u \in N(v)] = \sum_{u\in V} \Pr[(u,v) \in G] = \sum_{u\in V} p(u,v),$$

which does not depend on anything else but $p$.
By Definition 3.1,

$$\mathrm{Ov}(f)$$

$$= \frac{\mathbb{E}_{G',G''\sim f} |E(G') \cap E(G'')|}{\mathbb{E}_f |E(G)|}$$

$$= \frac{\sum_{u,v} \Pr[(u,v) \in G' \wedge (u,v) \in G'']}{\sum_{u,v} \Pr[(u,v) \in G]}$$

$$= \frac{\sum_{u,v} \Pr[(u,v) \in G'] \Pr[(u,v) \in G'']}{\sum_{u,v} \Pr[(u,v) \in G]}$$

$$= \frac{\sum_{u,v\in V} p^2(u,v)}{\sum_{u,v\in V} p(u,v)}, \forall f \in \mathcal{F}(p),$$

where we have used linearity of expectation and the independence between $G'$ and $G''$, completing the proof. $\square$

**Corollary 4.6.** *For any* $p\colon \binom{V}{2} \to [0,1]$, $\mathbb{E}_f[\triangle(G)] \le \sum_{\{u,v,w\}\in\binom{V}{3}} \min(p(u,v),p(u,w),p(v,w)), \forall f \in \mathcal{F}(p)$, *where* $\triangle(G)$ *is the number of triangles in* $G$.

*Proof.* By linearity of expectation and Property 4.4,

$$\mathbb{E}_f[\triangle(G)] = \sum_{\{u,v,w\}\in\binom{V}{3}} \Pr_f[\{(u,v),(u,w),(v,w) \in E(G)\}]$$

$$\le \sum_{\{u,v,w\}\in\binom{V}{3}} \min(p(u,v),p(u,w),p(v,w)).$$

$\square$

**Proposition 5.2** (Binding produces EPGMs). *For any $p\colon \binom{V}{2} \to [0,1]$ and any pair partition $\mathcal{P}$, $f^{BD}_{p;\mathcal{P}} \in \mathcal{F}(p)$.*

*Proof.* For each pair $(u,v)$, the existence of the corresponding edge is determined in the "binding" procedure on the group $P$ such that $(u,v) \in P$ (Lines 2 and 3), where $(u,v)$ is added into $\hat{E}$ and thus $E$ if and only if $s \leq \hat{p}(u,v) = p(u,v)$ (Line 9), which happens with probability $p(u,v)$ since $s \sim \mathcal{U}(0,1)$. $\qquad\square$

**Proposition 5.3** (Binding produces higher edge-group probabilities). *For any $p\colon \binom{V}{2} \to [0,1]$, any pair partition $\mathcal{P}$, and any $P \subseteq \binom{V}{2}$, $\mathrm{Pr}_{f^{BD}_{p;\mathcal{P}}}[P \subseteq E(G)] \geq \mathrm{Pr}_{f^{EI}_{p}}[P \subseteq E(G)]$.*

*Proof.* Let $\mathcal{P}'$ be a partition of $P$ such that $\mathcal{P}' \coloneqq \{P_0 \cap P\colon P_0 \in \mathcal{P}, P_0 \cap P \neq \emptyset\}$. Then

$$\mathrm{Pr}_{f^{BD}_{p;\mathcal{P}}}[P \subseteq E(G)] = \prod_{P' \in \mathcal{P}'} \min_{(u,v) \in P'} p(u,v)$$

$$= \prod_{(u,v) \in P\colon \exists P' \in \mathcal{P}', (u,v) = \arg\min_{(u',v') \in P'} p(u,v)} p(u,v)$$

$$\geq \prod_{(u,v) \in P} p(u,v),$$

since each $p(u,v) \leq 1$. $\qquad\square$

### B.2 MAXIMAL BINDING

As mentioned in Remark 4.5, the upper bound in Property 4.4 is tight, i.e., we can find EPGMs achieving the upper bound.

Indeed, we shall show below in Lemma B.2 that, as mentioned in Section 5.1, *maximal binding* (i.e., binding with all the pairs bound together $\mathcal{P} = \{\binom{V}{2}\}$) achieves the upper bound.

In order to prove Lemma B.2, let us prove the following lemma first.

**Lemma B.1** (The graph distribution with maximal binding). *For any $p\colon \binom{V}{2} \to [0,1]$, we first index the pairs (i.e., assign each pair a number) in $\binom{V}{2}$ in the descending order w.r.t. probabilities, i.e.,*

$$\binom{V}{2} = \{(u_1,v_1), (u_2,v_2), \ldots, (u_M,v_M)\}$$

*with $M = \binom{|V|}{2}$ such that*

$$p(u_1,v_1) \geq p(u_2,v_2) \geq \cdots \geq p(u_M,V_M),$$

*then the graph distribution with maximal binding is*

$$f^{BD}_{p;\{\binom{V}{2}\}}(G) = \begin{cases} 1 - p(u_1,v_1), & \text{if } G = (V, \emptyset), \\ p(u_M,v_M), & \text{if } G = (V, \binom{V}{2}), \\ p(u_i,v_i) - p(u_{i+1},v_{i+1}), & \text{if } G = (V, \{(u_j,v_j)\colon 1 \leq j \leq i\}), \forall i \in [M-1], \\ 0, & \text{otherwise.} \end{cases}$$

*Proof.* With $\mathcal{P} = \{\binom{V}{2}\}$, all the edge existences are determined by the same random variable $s$. Hence, if a pair $(u,v)$ exists, then all the pairs $(u',v')$ with $p(u',v') \geq p(u,v)$ must exist. The possible outputs are either $G = (V, \emptyset)$ or $G = (V, \{(u_j,v_j)\colon 1 \leq j \leq i\})$ for some $i \in [M]$. The case $G = (V, \emptyset)$ happens when $s > \max_{u,v \in V} p(u,v) = p(u_1,v_1)$ with probability $1 - p(u_1,v_1)$. The case $G = (V, \binom{V}{2})$ happens when $s \leq \min_{u,v \in V} p(u,v) = p(u_M,v_M)$ with probability $p(P_{\binom{|V|}{2}})$. For each remaining case $G = (V, \{(u_j,v_j)\colon 1 \leq j \leq i\}$ with $i \in [M-1]$, it happens when $p(u_{i+1},v_{i+1}) < s \leq p(u_i,v_i)$ with probability $p(u_i,v_i) - p(u_{i+1},v_{i+1})$. $\qquad\square$

**Lemma B.2** (Maximal binding achieves maximum edge-group probabilities)**.** *For any* $p\colon \binom{V}{2} \to [0,1]$ *and any edge-group* $P \subseteq \binom{V}{2}$, *we have*

$$\Pr_{f^{BD}_{p;\{\binom{V}{2}\}}}[P \subseteq E(G)] = \min_{(u,v)\in P} p(u,v), \forall f \in \mathcal{F}(p),$$

*where* $f^{BD}_{p;\mathcal{P}}$ *denotes the RGM defined by* $f^{BD}_{p;\mathcal{P}}(G) = \Pr[$*Algorithm 1 outputs G with inputs p and* $\mathcal{P}]$.

*Proof.* By Lemma B.1, in a graph $G$ generated by $f^{\mathrm{BD}}_{p;\{\binom{V}{2}\}}$, $P \subseteq E(G)$ if and only if $\arg\min_{(u,v)\in P} p(u,v) \in G.$, which happens with probability $\min_{(u,v)\in P} p(u,v)$. $\qquad\square$

### B.3 LOCAL BINDING

**Proposition 5.5** (Local binding produces EPGMs)**.** *For any* $p\colon \binom{V}{2} \to [0,1]$, $g\colon V \to [0,1]$ *and* $R \in \mathbb{N}$, $f^{LB}_{p;g,R} \in \mathcal{F}(p)$.

*Proof.* For each pair $(u,v)$, $\Pr_{f^{\mathrm{LB}}_{p;g,R}}[(u,v)] = \sum_{\mathcal{P}} \Pr_{\mathcal{P}\sim g}[\mathcal{P}] \Pr_{f^{\mathrm{BD}}_{p;\mathcal{P}}}[(u,v)]$. By Proposition 5.2, $\Pr_{f^{\mathrm{BD}}_{p;\mathcal{P}}}[(u,v)] = p(u,v), \forall \mathcal{P}$. Hence, $\Pr_{f^{\mathrm{LB}}_{p;g,R}}[(u,v)] = \sum_{\mathcal{P}} \Pr_{\mathcal{P}\sim g}[\mathcal{P}] p(u,v) = p(u,v)$. $\qquad\square$

**Theorem 5.7** (Time complexities of graph generation with local binding)**.** *Given* $p\colon \binom{V}{2} \to [0,1]$, $g\colon V \to [0,1]$, *and* $R \in \mathbb{N}$, $f^{LB}_{p;g,R}$ *generates a graph in* $O(R\left(\sum_{v\in V} g(v)\right)^2 + |V|^2)$ *time with high probability, with the worst case* $O(R|V|^2)$.

*Proof.* We have at most $R$ rounds of sampling and binding, where each round samples at most $|V|$ nodes and thus at most $\binom{|V|}{2}$ pairs. More specifically, the number of nodes sampled in each round is $\sum_{v\in V} g(v)$ in expectation, and thus $O(\sum_{v\in V} g(v))$ with high probability (e.g., you can use a Chernoff bound). Hence, it takes $O(R\sum_{v\in V} g(v))$ time with high probability, and at most $O(\binom{|V|}{2}R)$ time for the $R$ rounds. The number of remaining pairs is at most $\binom{|V|}{2}$ so dealing with them takes $O(\binom{|V|}{2})$ time. For the generation, we need to enumerate all the node groups and each pair in each group. Since the partition is disjoint, i.e., each pair is in exactly one group, each pair is visited exactly once, which takes $O(\binom{|V|}{2})$ time. In conclusion, generating a graph takes $O(R\sum_{v\in V} g(v) + |V|^2)$ with high probability, and $O(\binom{|V|}{2}R)$ time in the worst case. $\qquad\square$

**Theorem 5.8** (Tractable motif probabilities with local binding)**.** *For any* $p\colon \binom{V}{2} \to [0,1]$, $g\colon V \to [0,1]$, $R \in \mathbb{N}$, *and* $V' = \{u,v,w\} \in \binom{V}{3}$, *we can compute the closed-form* $\Pr_{f^{LB}_{p;g,R}}[E(G[V']) = E^*], \forall E^* \subseteq \binom{V'}{2}$ *as a function w.r.t.* $p$, $g$, *and* $R$.

*Proof.* The overall idea is that we (1) consider all the sub-cases of how all the pairs $\binom{V'}{2}$ are partitioned and grouped during the whole process, (2) compute the motif probabilities conditioned on each sub-case, and (3) finally take the summation of the motif probabilities in all the sub-cases.

We first consider all the cases of how all the pairs are sampled and grouped until $\binom{V'}{2}$ are fully determined. We divided the cases w.r.t. how the pairs in $\binom{V'}{2}$ are eventually grouped by the sampled node sets. First let us define some "short-cut" variables:

- the probability that among $V'$, exactly $V^*$ is sampled together in a round
$$p_g(V^*) \coloneqq \Pr_g[\{u,v,w\} \cap V_s = V^*] = \prod_{v\in V^*} g(v) \prod_{v'\notin V^*}(1 - g(v')), \forall V^* \subseteq V'$$

- the probability that among $V'$, at least two nodes (and thus at least one pair) are sampled together in a round
$$p_g(\mathcal{V}_{\geq 2}) \coloneqq \sum_{V^*:\, |V^*|\geq 2} p_g(V^*) = p_g(\{u,v\}) + p_g(\{u,w\}) + p_g(\{v,w\}) + p_g(\{u,v,w\})$$
$$= g(u)g(v)(1 - g(w)) + g(u)g(w)(1 - g(v))$$
$$+ g(v)g(w)(1 - g(u)) + g(u)g(v)g(w)$$

- the probability that among $V'$, at most one node (and thus no pair) is sampled together in a round

$$p_g(\mathcal{V}_{<2}) := 1 - p_g(\mathcal{V}_{\geq 2})$$

WLOG, we assume that $p(u,v) \geq p(u,w) \geq p(v,w)$.

$\boldsymbol{\{u, v, w\}.}$ The first time any pair in $\binom{V'}{2}$ is sampled in the $R$ rounds is when $u$, $v$, and $w$ are sampled by $g$ together, which happens with probability

$$q(\{u,v,w\}) = p_g(V') + p_g(\mathcal{V}_{<2})p_g(V') + p_g^2(\mathcal{V}_{<2})p_g(V') + \cdots + p_g^{R-1}(\mathcal{V}_{<2})p_g(V')$$

$$= \prod_{i=0}^{R-1} p_g^i(\mathcal{V}_{<2})p_g(V') = \frac{1 - p_g^R(\mathcal{V}_{<2})}{1 - p_g(\mathcal{V}_{<2})}p_g(V'),$$

where each term $p_g^i(\mathcal{V}_{<2})p_g(V')$ is the probability that in the first $i$ rounds at most one node among $V'$ is sampled and $V'$ is sampled altogether in the $(i+1)$-th round. Conditioned on that, it generates

- $\{(u,v), (u,w), (v,w)\}$ with probability $p(v,w)$; when the random variable $s$ in binding satisfies $s \leq p(v,w)$,
- $\{(u,v), (u,w)\}$ with probability $p(u,w) - p(v,w)$; when $p(v,w) < s \leq p(u,w)$,
- $\{(u,v)\}$ with probability $p(u,v) - p(u,w)$; when $p(u,w) < s \leq p(u,v)$, and
- $\emptyset$ with probability $1 - p(u,v)$; when $s > p(u,v)$.

$\boldsymbol{\{u, v\} \to \{u, v, w\}.}$ All the pairs in $\binom{V'}{2}$ are covered in twice in the $R$ rounds. At the first time, $u$ and $v$ are sampled together by $g$ but not $w$. At the second time, $u$, $v$, and $w$ are sampled together by $g$. This happens with probability

$$q(\{u,v\} \to \{u,v,w\}) = p_g(V') + (p_g(\mathcal{V}_{<2}) + p_g(\{u,v\}))\,p_g(V') + \cdots$$

$$+ (p_g(\mathcal{V}_{<2}) + p_g(\{u,v\}))^{R-1}\,p_g(V') - q(\{u,v,w\})$$

$$= \sum_{i=0}^{R-1} (p_g(\mathcal{V}_{<2}) + p_g(\{u,v\}))^i\,p_g(V') - q(\{u,v,w\})$$

$$= \left(\frac{1 - (p_g(\mathcal{V}_{<2}) + p_g(\{u,v\}))^R}{1 - (p_g(\mathcal{V}_{<2}) + p_g(\{u,v\}))} - \frac{1 - p_g^R(\mathcal{V}_{<2})}{1 - p_g(\mathcal{V}_{<2})}\right)p_g(V'),$$

where $(p_g(\mathcal{V}_{<2}) + p_g(\{u,v\}))^i\,p_g(V')$ is the probability that in the first $i$ rounds we either sample no pair between $V'$ or just $(u,v)$, and we sample $V'$ altogether in the $(i+1)$-th round, and $q(\{u,v,w\})$ is subtracted to exclude the cases where $(u,v)$ is not sampled in the first $i$ rounds. In such cases, when $(u,v)$ is sampled for the first time, we decide the existence of $(u,v)$, and then after that, when $V'$ is sampled altogether for the first time, we decide the existences of the remaining two pair $(u,w)$ and $(v,w)$. Hence, conditioned on that, it generates

- $\{(u,v), (u,w), (v,w)\}$ with probability $p(u,v)p(v,w)$; when $s_1 \leq p(u,v)$ in the round $(u,v)$ is sampled for the first time and $s_2 \leq p(v,w)$ in the round $V'$ is sampled altogether for the first time,
- $\{(u,v), (u,w)\}$ with probability $p(u,v)\,(p(u,w) - p(v,w))$; when $s_1 \leq p(u,v)$ and $p(v,w) < s_2 \leq p(u,w)$,
- $\{(u,v)\}$ with probability $p(u,v)\,(1 - p(u,w))$; when $s_1 \leq p(u,v)$ and $s_2 > p(u,w)$,
- $\{(u,w), (v,w)\}$ with probability $(1 - p(u,v))\,p(v,w)$; when $s_1 > p(u,v)$ and $s_2 \leq p(v,w)$,
- $\{(u,w)\}$ with probability $(1 - p(u,v))\,(p(u,w) - (v,w))$; when $s_1 > p(u,v)$ and $p(v,w) < s_2 \leq p(u,w)$, and
- $\emptyset$ with probability $(1 - p(u,v))\,(1 - p(u,w))$; when $s_1 > p(u,v)$ and $s_2 > p(u,w)$.

$\boldsymbol{\{u, w\} \to \{u, v, w\}.}$ Similarly, this happens with probability

$$q(\{u,w\} \to \{u,v,w\}) = \left(\frac{1 - (p_g(\mathcal{V}_{<2}) + p_g(\{u,w\}))^R}{1 - (p_g(\mathcal{V}_{<2}) + p_g(\{u,w\}))} - \frac{1 - p_g^R(\mathcal{V}_{<2})}{1 - p_g(\mathcal{V}_{<2})}\right)p_g(V')$$

Conditioned on that, it generates

- $\{(u,v),(u,w),(v,w)\}$ with probability $p(u,w)p(v,w)$; when $s_1 \le p(u,w)$ and $s_2 \le p(v,w)$,
- $\{(u,v),(u,w)\}$ with probability $p(u,w)\left(p(u,v)-p(v,w)\right)$; when $s_1 \le p(u,w)$ and $p(v,w) < s_2 \le p(u,v)$,
- $\{(u,w)\}$ with probability $p(u,w)\left(1-p(u,v)\right)$; when $s_1 \le p(u,w)$ and $s_2 > p(u,v)$,
- $\{(u,v),(v,w)\}$ with probability $\left(1-p(u,w)\right)p(v,w)$; when $s_1 > p(u,w)$ and $s_2 \le p(v,w)$,
- $\{(u,v)\}$ with probability $\left(1-p(u,w)\right)\left(p(u,v)-(v,w)\right)$; when $s_1 > p(u,w)$ and $p(v,w) < s_2 \le p(u,v)$, and
- $\emptyset$ with probability $\left(1-p(u,w)\right)\left(1-p(u,v)\right)$; when $s_1 > p(u,w)$ and $s_2 > p(u,v)$.

$\underline{\boldsymbol{\{v,w\} \to \{u,v,w\}.}}$ Similarly, this happens with probability

$$q(\{v,w\} \to \{u,v,w\}) = \left( \frac{1 - (p_g(\mathcal{V}_{<2}) + p_g(\{v,w\}))^R}{1 - (p_g(\mathcal{V}_{<2}) + p_g(\{v,w\}))} - \frac{1 - p_g^R(\mathcal{V}_{<2})}{1 - p_g(\mathcal{V}_{<2})} \right) p_g(V')$$

Conditioned on that, it generates

- $\{(u,v),(u,w),(v,w)\}$ with probability $p(v,w)p(u,w)$; when $s_1 \le p(v,w)$ and $s_2 \le p(u,w)$,
- $\{(u,v),(v,w)\}$ with probability $p(v,w)\left(p(u,v)-p(u,w)\right)$; when $s_1 \le p(v,w)$ and $p(u,w) < s_2 \le p(u,v)$,
- $\{(v,w)\}$ with probability $p(v,w)\left(1-p(u,v)\right)$; when $s_1 \le p(v,w)$ and $s_2 > p(u,v)$,
- $\{(u,v),(u,w)\}$ with probability $\left(1-p(v,w)\right)p(u,w)$; when $s_1 > p(v,w)$ and $s_2 \le p(u,w)$,
- $\{(u,v)\}$ with probability $\left(1-p(v,w)\right)\left(p(u,v)-(u,w)\right)$; when $s_1 > p(v,w)$ and $p(u,w) < s_2 \le p(u,v)$, and
- $\emptyset$ with probability $\left(1-p(v,w)\right)\left(1-p(u,v)\right)$; when $s_1 > p(v,w)$ and $s_2 > p(u,v)$.

**The remaining cases.** Three edges are determined independently. This happens with the remaining probability

$$q_{indep} = 1 - q(\{u,v,w\}) - q(\{u,v\} \to \{u,v,w\}) - q(\{u,w\} \to \{u,v,w\}) - q(\{v,w\} \to \{u,v,w\})$$

Conditioned on that, it generates each $E^* \subseteq \binom{V'}{2}$ with probability

$$\prod_{(x,y)\in E^*} p(x,y) \prod_{(x',y')\in \binom{V'}{2}\setminus E^*} (1 - p(x',y')).$$

Taking the summation of all the sub-cases gives the results as follows.

$\underline{\boldsymbol{E^* = \{(u,v),(u,w),(v,w)\}}}$

$$\begin{aligned}
\Pr\nolimits_{f_{p;g,R}^{\mathrm{LB}}}[E(G[V']) = \{(u,v),(u,w),(v,w)\}] = \ & q(\{u,v,w\})p(v,w) + \\
& q(\{u,v\} \to \{u,v,w\})p(u,v)p(v,w) + \\
& q(\{u,w\} \to \{u,v,w\})p(u,w)p(v,w) + \\
& q(\{v,w\} \to \{u,v,w\})p(v,w)p(u,w) + \\
& q_{indep}p(u,v)p(u,w)p(v,w)
\end{aligned}$$

$\underline{\boldsymbol{E^* = \{(u,v),(u,w)\}}}$

$$\begin{aligned}
\Pr\nolimits_{f_{p;g,R}^{\mathrm{LB}}}[E(G[V']) = \{(u,v),(u,w)\}] = \ & q(\{u,v,w\})\left(p(u,w)-p(v,w)\right) + \\
& q(\{u,v\} \to \{u,v,w\})p(u,v)\left(p(u,w)-p(v,w)\right) + \\
& q(\{u,w\} \to \{u,v,w\})p(u,w)\left(p(u,v)-p(v,w)\right) + \\
& q(\{v,w\} \to \{u,v,w\})\left(1-p(v,w)\right)p(u,w) + \\
& q_{indep}p(u,v)p(u,w)\left(1-p(v,w)\right)
\end{aligned}$$

$\underline{\boldsymbol{E^* = \{(u,v),(v,w)\}}}$

$$\begin{aligned}
\Pr\nolimits_{f_{p;g,R}^{\mathrm{LB}}}[E(G[V']) = \{(u,v),(v,w)\}] = \ & q(\{u,w\} \to \{u,v,w\})\left(1-p(u,w)\right)p(v,w) + \\
& q(\{v,w\} \to \{u,v,w\})p(v,w)\left(p(u,v)-p(u,w)\right) + \\
& q_{indep}p(u,v)p(v,w)\left(1-p(u,w)\right)
\end{aligned}$$

$\underline{\boldsymbol{E^* = \{(u, w), (v, w)\}}}$

$$\Pr{}_{f_{p;g,R}^{\mathrm{LB}}}[E(G[V']) = \{(u, w), (v, w)\}] = q(\{u, v\} \to \{u, v, w\})\left(1 - p(u, v)\right)p(v, w) +$$
$$q_{indep}p(u, w)p(v, w)\left(1 - p(u, v)\right)$$

$\underline{\boldsymbol{E^* = \{(u, v)\}}}$

$$\Pr{}_{f_{p;g,R}^{\mathrm{LB}}}[E(G[V']) = \{(u, v)\}] = q(\{u, v, w\})\left(p(u, v) - p(u, w)\right) +$$
$$q(\{u, v\} \to \{u, v, w\})p(u, v)\left(1 - p(u, w)\right) +$$
$$q(\{u, w\} \to \{u, v, w\})\left(1 - p(u, w)\right)\left(p(u, v) - p(v, w)\right) +$$
$$q(\{v, w\} \to \{u, v, w\})\left(1 - p(v, w)\right)\left(p(u, v) - p(u, w)\right) +$$
$$q_{indep}p(u, v)\left(1 - p(u, w)\right)\left(1 - p(v, w)\right)$$

$\underline{\boldsymbol{E^* = \{(u, w)\}}}$

$$\Pr{}_{f_{p;g,R}^{\mathrm{LB}}}[E(G[V']) = \{(u, w)\}] = q(\{u, v\} \to \{u, v, w\})\left(1 - p(u, v)\right)\left(p(u, w) - p(v, w)\right) +$$
$$q(\{u, w\} \to \{u, v, w\})p(u, w)\left(1 - p(u, v)\right) +$$
$$q_{indep}p(u, w)\left(1 - p(u, v)\right)\left(1 - p(v, w)\right)$$

$\underline{\boldsymbol{E^* = \{(v, w)\}}}$

$$\Pr{}_{f_{p;g,R}^{\mathrm{LB}}}[E(G[V']) = \{(u, w)\}] = q(\{v, w\} \to \{u, v, w\})p(v, w)\left(1 - p(u, v)\right) +$$
$$q_{indep}p(v, w)\left(1 - p(u, v)\right)\left(1 - p(u, w)\right)$$

$\underline{\boldsymbol{E^* = \emptyset}}$

$$\Pr{}_{f_{p;g,R}^{\mathrm{LB}}}[E(G[V']) = \{(u, w)\}] = q(\{u, v, w\})\left(1 - p(u, v)\right) +$$
$$q(\{u, v\} \to \{u, v, w\})\left(1 - p(u, v)\right)\left(1 - p(u, w)\right) +$$
$$q(\{u, w\} \to \{u, v, w\})\left(1 - p(u, w)\right)\left(1 - p(u, v)\right) +$$
$$q(\{v, w\} \to \{u, v, w\})\left(1 - p(v, w)\right)\left(1 - p(u, v)\right) +$$
$$q_{indep}\left(1 - p(u, v)\right)\left(1 - p(u, w)\right)\left(1 - p(v, w)\right)$$

$\square$

**Discussion on higher orders.** As mentioned in Remark 5.9, the reasoning in the proof above can be extended to higher orders. When the order of motifs increases, enumerating the cases of how all the pairs are sampled and grouped becomes more and more challenging. When considering 3-motifs, we are essentially considering the possible sequences of subsets up to order 3, where (1) each sequence should cover all the node pairs, and (2) each subset in the sequence should cover at least one pair that has not been covered by the subsets before it. The high-level idea would be similar, but the number increases exponentially:

- for 3-motifs, we need to consider 16 cases, 4 of which involve edge dependency, as shown above;
- for 4-motifs, we need to consider 16205 cases, 5261 of which involve edge dependency.

The above numbers are obtained using a recursive search. In principle, we can also derive the variance of the number of 3-motif by considering the probabilities of 6-motifs, since the co-existence of two 3-motifs involves motifs up to order 6. We leave the efficient computation for higher-order motifs as a future direction.

**Theorem 5.10** (Time complexity of computing motif probabilities with local binding)**.** *Given* $p\colon \binom{V}{2} \to [0, 1]$, $g\colon V \to [0, 1]$, *and* $R \in \mathbb{N}$, *computing* $\Pr_{f_{p;g,R}^{\mathrm{LB}}}[E(G[V']) = E^*]$ *takes* $O(|V|^3)$ *time in total for all* $E^* \subseteq \binom{V'}{2}$ *and* $V' \in \binom{V}{3}$.

*Proof.* For computing motif probabilities, we need to enumerate all triplets $V' = \{u, v, w\} \in \binom{V}{3}$ and compute the motif probability for each 3-motif. For each motif, the calculation only involves arithmetic operations, which takes $O(1)$ time since the formulae are fixed. In conclusion, computing 3-motif probabilities takes $O(\binom{|V|}{3})$ time. $\square$

## B.4 Parallel binding

**Proposition B.3** (Parallel binding produces EPGMs). *For any* $p \colon \binom{V}{2} \to [0,1]$, $g \colon V \to [0,1]$, *and* $R \in \mathbb{N}$, $f_{p;g,R}^{PB} \in \mathcal{F}(p)$.

*Proof.* For each pair $(u,v)$, if $\frac{1-(1-p(u,v))^{1/R}}{g(u)g(v)} \leq 1$, i.e., $p(u,v) \leq 1 - (1 - g(u)g(v))^R$, then $p_{rem}(u,v) = 0$ and

$$\Pr_{f_{p;g,R}^{PB}}[(u,v)] = 1 - \Pr[(u,v) \text{ not inserted in the } R \text{ rounds}] \Pr[(u,v) \text{ not inserted when dealing with } p_{rem}]$$
$$= 1 - (1 - g(u)g(v)r(u,v))^R(1 - p_{rem})$$
$$= 1 - (1 - p(u,v))$$
$$= p(u,v).$$

Otherwise, if $p(u,v) > 1 - (1 - g(u)g(v))^R$, then $r(u,v) = 1$ and

$$\Pr_{f_{p;g,R}^{PB}}[(u,v)] = 1 - \Pr[(u,v) \text{ not inserted in the } R \text{ rounds}] \Pr[(u,v) \text{ not inserted when dealing with } p_{rem}]$$
$$= 1 - (1 - g(u)g(v)r(u,v))^R(1 - p_{rem})$$
$$= 1 - (1 - g(u)g(v))^R \frac{1 - p(u,v)}{(1 - g(u)g(v))^R}$$
$$= 1 - (1 - p(u,v))$$
$$= p(u,v).$$

$\square$

**Theorem B.4** (Time complexities of graph generation with parallel binding). *Given* $p \colon \binom{V}{2} \to [0,1]$, $g \colon V \to [0,1]$, *and* $R \in \mathbb{N}$, $f_{p;g,R}^{PB}$ *generates a graph in* $O\left(R\left(\sum_{v \in V} g(v)\right)^2 + |V|^2\right)$ *time with high probability, with the worst case* $O(R|V|^2)$.

*Proof.* We have at most $R$ rounds of sampling and binding, where each round samples at most $|V|$ nodes and thus at most $\binom{|V|}{2}$ pairs. More specifically, the number of nodes sampled in each round is $\sum_{v \in V} g(v)$ in expectation, and thus $O(\sum_{v \in V} g(v))$ with high probability (e.g., one can use a Chernoff bound). Hence, it takes $O(R\sum_{v \in V} g(v))$ time with high probability, and at most $O(\binom{|V|}{2}R)$ time for the $R$ rounds. The number of pairs with $p_{rem} > 0$ is at most $\binom{|V|}{2}$ so dealing with them takes $O(\binom{|V|}{2})$ time. In conclusion, generating a graph takes $O(R\sum_{v \in V} g(v) + |V|^2)$ with high probability, and $O(\binom{|V|}{2}R)$ time in the worst case. $\square$

**Theorem B.5** (Tractable motif probabilities with parallel binding). *For any* $p \colon \binom{V}{2} \to [0,1]$, $g \colon V \to [0,1]$, $R \in \mathbb{N}$, *and* $V' = \{u,v,w\} \in \binom{V}{3}$, *we can compute the closed-form* $\Pr_{f_{p;g,R}^{PB}}[E(G[V']) = E^*], \forall E^* \subseteq \binom{V'}{2}$ *as a function w.r.t.* $p$, $g$, *and* $R$.

*Proof.* The overall idea is that we (1) compute the probabilities of each subset of $\binom{V'}{2}$ being inserted in each round and (2) accumulate the probabilities in $R$ rounds to obtain the final motif probabilities.

We first compute the probability of each subset of $\binom{V}{2}$ being inserted in each round. We divide the cases w.r.t. different sets of sampled nodes $V_s \cap V'$. First, let us define some "short-cut" variables:

- the probability that among $V'$, exactly $V^*$ is sampled together in a round

$$p_g(V^*) \coloneqq \Pr_g[\{u,v,w\} \cap V_s = V^*] = \prod_{v \in V^*} g(v) \prod_{v' \notin V^*} (1 - g(v)), \forall V^* \subseteq V'$$

- the probability that among $V'$, at least two nodes (and thus at least one pair) are sampled together in a round

$$p_g(\mathcal{V}_{\geq 2}) \coloneqq \sum_{V^* \colon |V^*| \geq 2} p_g(V^*) = p_g(\{u,v\}) + p_g(\{u,w\}) + p_g(\{v,w\}) + p_g(\{u,v,w\})$$
$$= g(u)g(v)(1 - g(w)) + g(u)g(w)(1 - g(v)) + g(v)g(w)(1 - g(u))$$

- the probability that among $V'$, at most one node (and thus no pair) is sampled together in a round

$$p_g(\mathcal{V}_{<2}) := 1 - p_g(\mathcal{V}_{\geq 2})$$

- the variables $r$ and $p_{rem}$ are defined as in Algorithm 3.

WLGO, we assume that $p(u,v) \geq p(u,w) \geq p(v,w)$.

__$V_s = \{u, v, w\}$.__ This happens with probability $p_g(V')$. Conditioned on that, it generates

- $\{(u,v), (u,w), (v,w)\}$ with probability $r(v,w)$; when $s \leq r(v,w)$,
- $\{(u,v), (u,w)\}$ with probability $r(u,w) - r(v,w)$; when $r(v,w) < s \leq r(u,w)$,
- $\{(u,v)\}$ with probability $r(u,v) - r(u,w)$; when $r(u,w) < s \leq r(u,v)$, and
- $\emptyset$ with probability $1 - r(u,v)$; when $s > r(u,v)$.

__$V_s = \{u, v\}$.__ This happens with probability $p_g(\{u,v\})$. Conditioned on that, it generates

- $\{(u,v)\}$ with probability $r(u,v)$; when $s \leq r(u,v)$, and
- $\emptyset$ with probability $1 - r(u,v)$l when $s > r(u,v)$.

__$V_s = \{u, w\}$.__ This happens with probability $p_g(\{u,w\})$. Conditioned on that, it generates

- $\{(u,w)\}$ with probability $r(u,w)$; when $s \leq r(u,w)$,
- $\emptyset$ with probability $1 - r(u,w)$; when $s > r(u,w)$.

__$V_s\{v, w\}$.__ This happens with probability $p_g(\{v,w\})$. Conditioned on that, it generates

- $\{(v,w)\}$ with probability $r(v,w)$; when $s \leq r(v,w)$,
- $\emptyset$ with probability $1 - r(v,w)$; when $s > r(v,w)$.

__The remaining cases (i.e., $|V_s \cap V'| \leq 1$).__ This happens with probability $p_g(\mathcal{V}_{<2})$. Conditioned on that, it generates

- $\emptyset$ with probability 1.

__Summary for each round.__ Let $p_{round}(E^*)$ denote the probability of $E^*$ being generated in each round, for each $E^* \subseteq \binom{V'}{2}$. We have

- $p_{round}(\{(u,v), (u,w), (v,w)\}) = p_g(V')r(v,w)$,
- $p_{round}(\{(u,v), (u,w)\}) = p_g(V')(r(u,w) - r(v,w))$,
- $p_{round}(\{(u,v)\}) = p_g(V')(r(u,v) - r(u,w)) + p_g(\{u,v\})r(u,v)$,
- $p_{round}(\{(u,w)\}) = p_g(\{u,w\})r(u,w)$,
- $p_{round}(\{(v,w)\}) = p_g(\{v,w\})r(v,w)$, and
- $p_{round}(\emptyset) = 1 - p_g(V')r(u,v) - p_g(\{u,v\})r(u,v) - p_g(\{u,w\})r(u,w) - p_g(\{v,w\})r(v,w)$.

We are now ready to compute the motif probabilities.

__$E^* = \emptyset$.__ This happens when $\emptyset$ is generated in all $R$ rounds and for the remaining probabilities $p_{rem}$, with probability

$$\Pr_{f_{p;g,R}^{\text{PB}}}[E(G[V']) = \emptyset] = (p_{round}(\emptyset))^R (1 - p_{rem}(u,v))(1 - p_{rem}(u,w))(1 - p_{rem}(v,w)).$$

__$E^* = \{(u,v)\}$.__ This happens when either $\emptyset$ or $\{(u,v)\}$ is generated in all $R$ rounds and for $p_{rem}$, and $(u,v)$ is generated in at least one round, which has probability

$$\Pr_{f_{p;g,R}^{\text{PB}}}[E(G[V']) = \{(u,v)\}]$$
$$= (p_{round}(\emptyset))^R p_{rem}(u,v)(1 - p_{rem}(u,w))(1 - p_{rem}(v,w)) +$$
$$((p_{round}(\emptyset) + p_{round}(\{(u,v)\}))^R - (p_{round}(\emptyset))^R)(1 - p_{rem}(u,w))(1 - p_{rem}(v,w)),$$

where $((p_{round}(\emptyset) + p_{round}(\{(u,v)\}))^R - (p_{round}(\emptyset))^R)$ is the probability that in the $R$ rounds, only $(u,v)$ is inserted.

$\underline{\boldsymbol{E^* = \{(u,w)\}.}}$ Similarly, this happens with probability

$$\Pr_{f_{p;g,R}^{\text{PB}}}[E(G[V']) = \{(u,w)\}]$$

$$= (p_{round}(\emptyset))^R p_{rem}(u,w)(1 - p_{rem}(u,v))(1 - p_{rem}(v,w))+$$

$$((p_{round}(\emptyset) + p_{round}(\{(u,w)\}))^R - (p_{round}(\emptyset))^R)(1 - p_{rem}(u,v))(1 - p_{rem}(v,w)).$$

$\underline{\boldsymbol{E^* = \{(v,w)\}.}}$ Similarly, this happens with probability

$$\Pr_{f_{p;g,R}^{\text{PB}}}[E(G[v']) = \{(u,w)\}]$$

$$= (p_{round}(\emptyset))^R p_{rem}(v,w)(1 - p_{rem}(u,v))(1 - p_{rem}(u,w))+$$

$$((p_{round}(\emptyset) + p_{round}(\{(v,w)\}))^R - (p_{round}(\emptyset))^R)(1 - p_{rem}(u,v))(1 - p_{rem}(u,w)).$$

$\underline{\boldsymbol{E^* = \{(u,v),(u,w)\}.}}$ This happens when one among $\emptyset$, $\{(u,v)\}$, $\{(u,w)\}$, and $\{(u,v),(u,w)\}$ is generated in all $R$ rounds and for $R_{rem}$, while excluding the cases ending up with $\emptyset$, $\{(u,v)\}$, or $\{(u,w)\}$. This happens with probability

$$\Pr_{f_{p;g,R}^{\text{PB}}}[E(G[V']) = \{(u,v),(u,w)\}]$$

$$= (p_{round}(\emptyset))^R p_{rem}(u,v) p_{rem}(u,w)(1 - p_{rem}(v,w))+$$

$$((p_{round}(\emptyset) + p_{round}(\{(u,v)\}))^R - (p_{round}(\emptyset))^R) p_{rem}(u,w)(1 - p_{rem}(v,w))+$$

$$((p_{round}(\emptyset) + p_{round}(\{(u,w)\}))^R - (p_{round}(\emptyset))^R) p_{rem}(u,v)(1 - p_{rem}(v,w))+$$

$$\tilde{p}(\{(u,v),(u,w)\}; R)(1 - p_{rem}(v,w)),$$

where

$$\tilde{p}(\{(u,v),(u,w)\}; R)$$

$$= (p_{round}(\emptyset) + p_{round}(\{(u,v)\}) + p_{round}(\{(u,w)\}) + p_{round}(\{(u,v),(u,w)\}))^R -$$

$$(p_{round}(\emptyset) + p_{round}(\{(u,v)\}))^R -$$

$$(p_{round}(\emptyset) + p_{round}(\{(u,w)\}))^R +$$

$$(p_{round}(\emptyset))^R$$

is the probability that exactly $(u,v)$ and $(u,w)$ are inserted in the $R$ rounds, using the inclusion-exclusion principle.

$\underline{\boldsymbol{E^* = \{(u,v),(v,w)\}.}}$ Similarly, this happens with probability

$$\Pr_{f_{p;g,R}^{\text{PB}}}[E(G[V']) = \{(u,v),(v,w)\}]$$

$$= (p_{round}(\emptyset))^R p_{rem}(u,v) p_{rem}(v,w)(1 - p_{rem}(u,w))+$$

$$((p_{round}(\emptyset) + p_{round}(\{(u,v)\}))^R - (p_{round}(\emptyset))^R) p_{rem}(v,w)(1 - p_{rem}(u,w))+$$

$$((p_{round}(\emptyset) + p_{round}(\{(v,w)\}))^R - (p_{round}(\emptyset))^R) p_{rem}(u,v)(1 - p_{rem}(u,w))+$$

$$\tilde{p}(\{(u,v),(v,w)\}; R)(1 - p_{rem}(u,w)),$$

where

$$\tilde{p}(\{(u,v),(v,w)\}; R) = (p_{round}(\emptyset) + p_{round}(\{(u,v)\}) + p_{round}(\{(v,w)\}))^R -$$

$$(p_{round}(\emptyset) + p_{round}(\{(u,v)\}))^R -$$

$$(p_{round}(\emptyset) + p_{round}(\{(v,w)\}))^R +$$

$$(p_{round}(\emptyset))^R.$$

Note that $p_{round}(\{(u,v),(v,w)\}) = 0$.

$\underline{\boldsymbol{E^* = \{(u, w), (v, w)\}}}.$ Similarly, this happens with probability

$$\Pr_{f_{p;g,R}^{\text{PB}}}[E(G[V']) = \{(u, w), (v, w)\}]$$

$$= (p_{round}(\emptyset))^R p_{rem}(u, w) p_{rem}(v, w)(1 - p_{rem}(u, v)) +$$
$$((p_{round}(\emptyset) + p_{round}(\{(u, w)\}))^R - (p_{round}(\emptyset))^R) p_{rem}(v, w)(1 - p_{rem}(u, v)) +$$
$$((p_{round}(\emptyset) + p_{round}(\{(v, w)\}))^R - (p_{round}(\emptyset))^R) p_{rem}(u, w)(1 - p_{rem}(u, v)) +$$
$$\tilde{p}(\{(u, w), (v, w)\}; R)(1 - p_{rem}(u, v)),$$

where

$$\tilde{p}(\{(u, w), (v, w)\}; R) = ((p_{round}(\emptyset) + p_{round}(\{(u, w)\}) + p_{round}(\{(v, w)\}))^R -$$
$$(p_{round}(\emptyset) + p_{round}(\{(u, w)\}))^R -$$
$$(p_{round}(\emptyset) + p_{round}(\{(v, w)\}))^R +$$
$$(p_{round}(\emptyset))^R)$$

Note that $p_{round}(\{(u, w), (v, w)\}) = 0$.

$\underline{\boldsymbol{E^* = \{(u, v), (u, w), (v, w)\}}}.$ This happens with the remaining probability, i.e.,

$$\Pr_{f_{p;g,R}^{\text{PB}}}[E(G[V']) = \{(u, v), (u, w), (v, w)\}] = 1 - \sum_{E' \subsetneq \binom{V'}{2}} \Pr_{f_{p;g,R}^{\text{PB}}}[E(G[V']) = E'].$$

$\square$

**Discussion on higher orders.** Similar to the counterpart for local binding, the reasoning in the proof above can be extended to higher orders. When the order of motifs increases, both considering the cases in each round and accumulating them in multiple rounds become increasingly challenging. For the cases in each round, we first need to consider more cases of $V_s$, i.e., all the subsets of $V'$. For accumulating the probabilities, for each $E^*$, we first need to consider all the cases (i.e., all the subsets of $E^*$) in each round that can accumulate to $E*$, and we need to use the inclusion-exclusion principle to avoid counting some sub-motifs multiple times, where again all the subsets of $E^*$ need to be considered. Hence, for motifs of order $k$, the number of cases is at least $O(2^{\binom{k}{2}})$.

**Theorem B.6** (Time complexity of computing motif probabilities with parallel binding)**.** *Given* $p \colon \binom{V}{2} \to [0, 1]$, $g \colon V \to [0, 1]$, *and* $R \in \mathbb{N}$, *computing* $\Pr_{f_{p;g,R}^{\text{PB}}}[E(G[V']) = E^*]$ *takes* $O(|V|^3)$ *time in total for all* $E^* \subseteq \binom{V'}{2}$ *and* $V' \in \binom{V}{3}$.

*Proof.* For computing motif probabilities, we need to enumerate all triplets $V' = \{u, v, w\} \in \binom{V}{3}$ and compute the motif probability for each 3-motif. For each motif, the calculation only involves arithmetic operations, which takes $O(1)$ time since the formulae are fixed. In conclusion, computing 3-motif probabilities takes $O(\binom{|V|}{3})$ time. $\square$

### B.5 FITTING

#### B.5.1 THE ERDŐS-RÉNYI (ER) MODEL

**Definition.** The Erdős-Rényi (ER) model (Erdős & Rényi, 1959) outputs edge probabilities with two parameters: $n_0$ and $p_0$, and the output is $p_{n_0, p_0}^{ER}$ with $p_{n_0, p_0}^{ER}(u, v) = p_0, \forall u, v \in \binom{V}{2}$ with $V = [n_0]$. Given a graph $G = (V = [n], E)$, ER outputs $n_0 = n$ and $p_0 = \frac{2|E|}{n(n-1)}$.

**Lemma 5.12** (Reduced time complexity with ER)**.** *Given* $n_0 \in \mathbb{N}$, $p_0 \in [0, 1]$, $g_0 \in [0, 1]$, *and* $R \in \mathbb{N}$, *computing both* $\Pr_{f_{p;g,R}^{\text{LB}}}[E(G[V']) = E^*]$ *and* $\Pr_{f_{p;g,R}^{\text{PB}}}[E(G[V']) = E^*]$ *takes* $O(1)$ *times in total for all* $E^* \subseteq \binom{V'}{2}$ *and* $V' \in \binom{V}{3}$ *with* $p = p_{n_0, p_0}^{ER}$ *and* $g(v) = g_0, \forall v \in V = [n_0]$.

*Proof.* When $p(u, v) \equiv v_0$ and $g(v) \equiv g_0$, both $\Pr_{f_{p;g,R}^{\text{LB}}}[E(G[V']) = E^*]$ and $\Pr_{f_{p;g,R}^{\text{PB}}}[E(G[V']) = E^*]$ become the same functions for all $V' \in \binom{V}{3}$, which only involve arithmetic operations on $p_0$ and $g_0$ and thus take $O(1)$ time for computation. Since the functions are the

same for all $V' \in \binom{V}{3}$, we only need to calculate for a single $V'$. Hence, the total time complexity is still $O(1)$. The detailed formulae are as follows.

**Local binding.** Fix any $V' \in \binom{V}{3}$, we have

$$p_g(V^*) = g_0^{|V^*|}(1-g_0)^{3-|V^*|}, \forall V^* \subseteq V',$$

$$p_g(\mathcal{V}_{\geq 2}) = 3g_0^2(1-g_0) + g_0^3,$$

and

$$p_g(\mathcal{V}_{<2}) = 3g_0(1-g_0)^2 + (1-g_0)^3.$$

Hence

$$q(\{u,v,w\}) = \frac{1 - \left(3g_0(1-g_0)^2 + (1-g_0)^3\right)^R}{3g_0^2(1-g_0) + g_0^3} g_0^3,$$

$$q_2 := q(\{u,v\} \to \{u,v,w\}) = q(\{u,w\} \to \{u,v,w\}) = q(\{v,w\} \to \{u,v,w\})$$

$$= \left( \frac{1 - \left(3g_0(1-g_0)^2 + (1-g_0)^3 + g_0^2(1-g_0)\right)^R}{2g_0^2(1-g_0) + g_0^3} - \frac{1 - \left(3g_0(1-g_0)^2 + (1-g_0)^3\right)^R}{3g_0^2(1-g_0) + g_0^3} \right),$$

and

$$q_{indep} = 1 - q(\{u,v,w\}) - 3q_2.$$

$\underline{\boldsymbol{E^* = \{(u,v),(u,w),(v,w)\}}}$

$$\Pr_{f_{p;g,R}^{\text{LB}}}[E(G[V']) = \{(u,v),(u,w),(v,w)\}] = q(\{u,v,w\})p_0 + 3q_2 p_0^2 + q_{indep}p_0^3$$

$\underline{\boldsymbol{|E^*| = 2}}$

For each $E^*$ with $|E^*| = 2$, i.e., $E^* = \{(u,v),(u,w)\}$ or $\{(u,v),(v,w)\}$ or $(u,w),(v,w)$, we have

$$\Pr_{f_{p;g,R}^{\text{LB}}}[E(G[V']) = E^*] = q_2 p_0(1-p_0) + q_{indep}p_0^2(1-p_0)$$

$\underline{\boldsymbol{|E^*| = 1}}$

For each $E^*$ with $|E^*| = 1$, i.e., $E^* = \{(u,v)\}$ or $\{(u,w)\}$ or $(v,w)$, we have

$$\Pr_{f_{p;g,R}^{\text{LB}}}[E(G[V']) = E^*] = q_2 p_0(1-p_0) + q_{indep}p_0(1-p_0)^2$$

$\underline{\boldsymbol{E^* = \emptyset}}$

$$\Pr_{f_{p;g,R}^{\text{LB}}}[E(G[V']) = \{(u,w)\}] = q(\{u,v,w\})(1-p_0) + 3q_2(1-p_0)^2 + q_{indep}(1-p_0)^3$$

$\square$

### B.5.2 THE CHUNG-LU (CL) MODEL

**Definition.** The Chung-Lu (CL) model (Chung & Lu, 2002) outputs edge probabilities with a sequence of expected degrees $D = (d_1, d_2, \ldots, d_n)$, and the output is $p_D^{CL}$ with $p_D^{CL}(u,v) = \min(\frac{d_u d_v}{\sum_{i=1}^n d_i}, 1), \forall u,v \in \binom{V}{2}$ with $V = [n]$. Given a graph $G = (V = [n], E)$, CL outputs $d_i = d(i; G)$ for each node $i \in V$.

**Lemma 5.13** (Reduced time complexity with CL). *Given $D = (d_1, d_2, \ldots, d_n)$, $g_d$ for $d \in \{d_1, d_2, \ldots, d_n\}$, and $R \in \mathbb{N}$, computing both $\Pr_{f_{p;g,R}^{\text{LB}}}[E(G[V']) = E^*]$ and $\Pr_{f_{p;g,R}^{\text{PB}}}[E(G[V']) = E^*]$ for all $E^* \subseteq \binom{V'}{2}$ and $V' \in \binom{[n]}{3}$ takes $O(k_{deg}^3)$ times with $p = p_D^{CL}$ and $g(i) = g_{d_i}, \forall i \in [n]$.*

*Proof.* The key idea is that given $V' = \{i,j,k\} \in \binom{V}{3}$, both the three edge probabilities (i.e., $p(i,j)$, $p(i,k)$, and $p(j,k)$) and the three node-sampling probabilities (i.e., $g(i)$, $g(j)$, and $g(k)$) are fully determined by the degrees of the three nodes.

Hence, we only need to calculate motif probabilities for each degree combination instead of each node combination. Since we have $k_{deg}$ different degrees, the total number of degree combinations of

size 3 is $O(k_{deg}^3)$, and the calculation for each combination takes $O(1)$ time on arithmetic operations with fixed formulae. In conclusion, the total time complexity is $O(k_{deg}^3)$.

Some details are as follows. Let $k_{deg} = \{d_1, d_2, \ldots, d_n\} = \{\tilde{d}_1, \tilde{d}_2, \ldots, \tilde{d}_{k_{deg}}\}$, and let $n_i$ denote the number of nodes with degree $\tilde{d}_i$, for $i \in [k_{deg}]$. Given three degrees $\tilde{d}_i$, $\tilde{d}_j$, and $\tilde{d}_k$, we have

- $n_i n_j n_k$ such combinations, when $i \neq j$, $i \neq k$, and $j \neq k$
- $\binom{n_i}{2} n_k$ such combinations, when $i = j$ and $i \neq k$; similarly for $i = k$ and $i \neq j$ or $j = k$ and $i \neq j$
- $\binom{n_i}{3}$ such combinations, when $i = j = k$.

$\square$

### B.5.3 THE STOCHASTIC BLOCK (SB) MODEL

**Definition.** Given a graph $G = (V = [n], E)$ and a node partition $f_B \colon [n] \to [c]$ with $c \in \mathbb{N}$, let $V_i = \{v \in V \colon f_B(v) = i\}$ denote the set of nodes partitioned in the $i$-th group for $i \in [c]$. The fitting of the edge probabilities in the stochastic block (SB) model gives $p_B \colon [c] \times [c] \to [0, 1]$ with $p_B(i, i) = \frac{|E(G[V_i])|}{\binom{|V_i|}{2}}$ and $p_B(i, j) = \frac{|E \cap \{(v, v') \colon v \in V_i, v' \in V_j\}|}{|V_i||V_j|}$, for $i \neq j \in [c]$.

**Lemma 5.14** (Reduced time complexity with SB). *Given $f_B \colon [n_0] \to [c]$, $f_B \colon [n_0] \to [c]$, $g_i$ for $i \in [c]$, and $R \in \mathbb{N}$, computing both $\Pr_{f_{p;g,R}^{LB}}[E(G[V']) = E^*]$ and $\Pr_{f_{p;g,R}^{PB}}[E(G[V']) = E^*]$ takes $O(c^3)$ times in total for all $E^* \subseteq \binom{V'}{2}$ and $V' \in \binom{[n]}{3}$ with $p = p_{f_B, p_B}^{SB}$ and $g(v) = g_{f_B(v)}$ for each $v \in V = [n]$.*

*Proof.* The key idea is that given $V' = \{i, j, k\} \in \binom{V}{3}$, both the three edge probabilities (i.e., $p(i, j)$, $p(i, k)$, and $p(j, k)$) and the three node-sampling probabilities (i.e., $g(i)$, $g(j)$, and $g(k)$) are fully determined by the membership the three nodes, i.e., $f_B(i)$, $f_B(j)$, and $f_B(k)$.

Hence, we only need to calculate motif probabilities for each membership combination instead of each node combination. Since we have $c$ different groups, the total number of degree combinations of size 3 is $O(c^3)$, and the calculation for each combination takes $O(1)$ time on arithmetic operations with fixed formulae. In conclusion, the total time complexity is $O(c^3)$.

Some details are as follows. Let $n_i = |V_i|$ denote the number of nodes in the $i$-th group. Given three group membership indicators $i$, $j$, and $k$, we have

- $n_i n_j n_k$ such combinations, when $i \neq j$, $i \neq k$, and $j \neq k$
- $\binom{n_i}{2} n_k$ such combinations, when $i = j$ and $i \neq k$; similarly for $i = k$ and $i \neq j$ or $j = k$ and $i \neq j$
- $\binom{n_i}{3}$ such combinations, when $i = j = k$.

$\square$

### B.5.4 THE STOCHASTIC KRONECKER (KR) MODEL

**Definition B.7** (Kronecker product and Kronecker power). Given two matrices $A \in \mathbb{R}^{m \times n}$ and $B \in \mathbb{R}^{p \times q}$, the Kronecker product between $A$ and $B$ is

$$\mathbf{A} \otimes \mathbf{B} = \begin{bmatrix} a_{11}b_{11} & a_{11}b_{12} & \cdots & a_{11}b_{1q} & \cdots & \cdots & a_{1n}b_{11} & a_{1n}b_{12} & \cdots & a_{1n}b_{1q} \\ a_{11}b_{21} & a_{11}b_{22} & \cdots & a_{11}b_{2q} & \cdots & \cdots & a_{1n}b_{21} & a_{1n}b_{22} & \cdots & a_{1n}b_{2q} \\ \vdots & \vdots & \ddots & \vdots & & & \vdots & \vdots & \ddots & \vdots \\ a_{11}b_{p1} & a_{11}b_{p2} & \cdots & a_{11}b_{pq} & \cdots & \cdots & a_{1n}b_{p1} & a_{1n}b_{p2} & \cdots & a_{1n}b_{pq} \\ \vdots & \vdots & & \vdots & \ddots & & \vdots & \vdots & & \vdots \\ \vdots & \vdots & & \vdots & & \ddots & \vdots & \vdots & & \vdots \\ a_{m1}b_{11} & a_{m1}b_{12} & \cdots & a_{m1}b_{1q} & \cdots & \cdots & a_{mn}b_{11} & a_{mn}b_{12} & \cdots & a_{mn}b_{1q} \\ a_{m1}b_{21} & a_{m1}b_{22} & \cdots & a_{m1}b_{2q} & \cdots & \cdots & a_{mn}b_{21} & a_{mn}b_{22} & \cdots & a_{mn}b_{2q} \\ \vdots & \vdots & \ddots & \vdots & & & \vdots & \vdots & \ddots & \vdots \\ a_{m1}b_{p1} & a_{m1}b_{p2} & \cdots & a_{m1}b_{pq} & \cdots & \cdots & a_{mn}b_{p1} & a_{mn}b_{p2} & \cdots & a_{mn}b_{pq}. \end{bmatrix}$$

Given $k \in \mathbb{N}$, the $k$ Kronecker power of $A$ is

$$\underbrace{A \otimes (A \cdots (A \otimes (A \otimes A)))}_{k-1 \text{ times of Kronecker products}}.$$

**Definition.** The stochastic Kronecker (KR) model (Leskovec et al., 2010) outputs edge probabilities with a seed matrix $\theta \in [0,1]^{2 \times 2}$ and $k_{KR} \in \mathbb{N}$,[11] and the output $p_{\theta,k_{KR}}^{KR}$ is the $k_{KR}$-th Kronecker power of $\theta$.

**Lemma B.8** (Node equivalence in KR). *Given $\theta \in [0,1]^{2 \times 2}$, $k_{KR} \in \mathbb{N}$, $g_i$ for $0 \leq i \leq k_{KR}$, and $R \in \mathbb{N}$, computing both $\Pr_{f_{p;g,R}^{lB}}[E(G[V']) = E^*]$ and $\Pr_{f_{p;g,R}^{PB}}[E(G[V']) = E^*]$ takes $O(k_{KR}^7)$ times in total for all $E^* \subseteq \binom{V'}{2}$ and $V' \in \binom{[n]}{3}$ with $p = p_{\theta,k_{KR}}^{KR}$ and $g(v) = g_i$ with $i$ being the number of ones in the binary representation of $v - 1$, for each $v \in [2^{k_{KR}}]$.*

*Proof.* A square binary matrix $P \in \{0,1\}^{n \times n}$ for some $n \in \mathbb{N}$ is a permutation matrix if exactly one entry in each row or column of $P$ is 1, i.e., $\sum_k P_{ik} = \sum_k P_{kj} = 1, \forall i,j \in [n]$.

With binary node labels, given two nodes

$$u = (u_1 u_2 \cdots u_{k_{KR}})_2$$

and

$$v = (v_1 v_2 \cdots v_{k_{KR}})_2,$$

we have

$$\theta_{uv}^{(k_{KR})} = \prod_{i=1}^{k_{KR}} \theta_{u_i v_i},$$

which implies that for any permutation $\pi \in S_{k_{KR}}$,

$$\theta_{uv}^{(k_{KR})} = \theta_{\pi(u)\pi(v)}^{(k_{KR})}, \forall u,v,$$

where with a slight abuse of notation,

$$\pi(u) = (u_{\pi(1)} u_{\pi(2)} \cdots u_{\pi(k_{KR})})_2$$

and

$$\pi(v) = (v_{\pi(1)} v_{\pi(2)} \cdots v_{\pi(k_{KR})})_2.$$

On the other hand, for any two nodes with the same number of ones in the binary representations, we can find a permutation $\pi$ between the two binary representations by seeing them as sequences. Let $P = P_\pi \in \{0,1\}^{2^{k_{KR}} \times 2^{k_{KR}}}$ with $P_{ij} = 1$ if and only if $\pi$ converts the binary presentation of $i - 1$ to that of $j - 1$, and we have $P^\top \theta_{uv}^{(k_{KR})} P = \theta_{uv}^{(k_{KR})}$. $\qquad \square$

---

[11]We consider the commonly used 2-by-2 seed matrices.

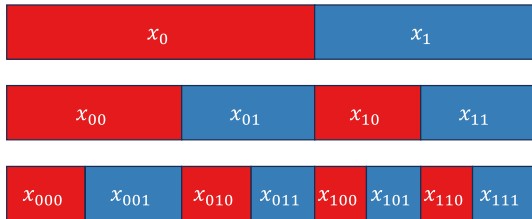

Figure 2: The node combinations in KR.

*Remark* B.9. The equivalence in KR is slightly weaker than that in the other three models (ER, CL, and SB). Specifically, in the other three models, "two nodes $i$ and $j$ are equivalent" means that, when you swap $i$ and $j$ while keeping the other nodes unchanged, the RGM is kept unchanged. For KR, the equivalence is weaker in that you have to swap $i$ and $j$ together with all the other nodes w.r.t. a permutation. This is also why the reduced time complexity is $O(k_{KR}^7)$ instead of $O(k_{KR}^3)$ in Lemma 5.15.

**Lemma 5.15** (Reduced time complexity with KR). *Given $\theta \in [0,1]^{2\times 2}$, $k_{KR} \in \mathbb{N}$, $g_i$ for $0 \leq i \leq k_{KR}$, and $R \in \mathbb{N}$, computing both $\Pr_{f_{p;g,R}^{LB}}[E(G[V']) = E^*]$ and $\Pr_{f_{p;g,R}^{PB}}[E(G[V']) = E^*]$ for all $E^* \subseteq \binom{V'}{2}$ and $V' \in \binom{[n]}{3}$ takes $O(k_{KR}^7)$ times with $p = p_{\theta,k_{KR}}^{KR}$ and $g(v) = g_i$ with $i$ being the number of ones in the binary representation of $v - 1$, for each $v \in [2^{k_{KR}}]$.*

*Proof.* We divide node combinations w.r.t the binary node labels. As shown in the proof of Lemma B.8, node combinations are equivalent with permutations on the binary node labels. Hence, in each equivalent class of node combinations, we can consider only the one with the form as shown in Figure 2, where each number ($x_0$, $x_1$, $x_{00}$, etc.) represents the number of zeros and ones. Here,

- the first node $v_1$ (more precisely, its binary node representation) has $x_0$ zeros first and then $x_1$ ones,
- the second node $v_2$ has $x_{00}$ zeros, then $x_{01}$ ones, then $x_{10}$ zeros, and finally $x_{11}$ ones, and
- the third node $v_3$ has $x_{000}$ zeros, then $x_{001}$ ones, then $x_{010}$ zeros, then $x_{011}$ ones, then $x_{100}$ zeros, then $x_{101}$ ones, then $x_{110}$ zeros, and finally $x_{111}$ ones.

As indicated in the figure, we have

- $x_0 + x_1 = k_{KR}$
- $x_{00} + x_{01} = x_0$, $x_{10} + x_{11} = x_1$
- $x_{000} + x_{001} = x_{00}$, $x_{010} + x_{011} = x_{01}$, $x_{100} + x_{101} = x_{10}$, and $x_{110} + x_{111} = x_{11}$.

The number of different equivalent classes is upper-bounded by

$$\sum_{x_0=0}^{k_{KR}} \sum_{x_{00}=0}^{x_0} \sum_{x_{10}=0}^{k_{KR}-x_0} \sum_{x_{000}=0}^{x_{00}} \sum_{x_{010}=0}^{x_0-x_{00}} \sum_{x_{100}=0}^{x_{10}} \sum_{x_{110}=0}^{k_{KR}-x_0-x_{10}} 1$$

$$= \frac{(k_{KR}+1)(k_{KR}+2)(k_{KR}+3)(k_{KR}+4)(k_{KR}+5)(k_{KR}+6)(k_{KR}+7)}{5040} = O(k_{KR}^7).$$

For each equivalent class, the calculation only involves arithmetic with a fixed formula and thus takes $O(1)$ time. Note that the Kronecker power can be computed beforehand with much lower time complexity, i.e., $o(k_{KR}^7)$ (Seroussi & Ma, 1983). In conclusion, the total time complexity is $O(k_{KR}^7)$. $\qquad\square$

## C ON (NON-)ISOLATED NODES

### C.1 TRACTABLE NUMBER OF (NON-)ISOLATED NODES WITH PARALLEL BINDING

**Theorem C.1** (Tractable number of (non-)isolated nodes with parallel binding). *For any $p\colon \binom{V}{2} \to [0,1]$, $g\colon V \to [0,1]$, $R \in \mathbb{N}$, we can compute the closed-form (w.r.t. $p$, $g$, and $R$) $\mathbb{E}_{f_{p;g,R}^{PB}}[|\{v \in G\colon d(v;G) \geq 1\}|]$.*

*Proof.* By the linearity of expectation,

$$\mathbb{E}_{f^{\mathrm{PB}}_{p;g,R}}[|\{v \in G \colon d(v;G) \geq 1\}|] = \sum_{v \in V} \Pr_{f^{\mathrm{PB}}_{p;g,R}}[d(v;G) \geq 1].$$

Hence, we only need to compute the probability of each node $v$ being (non-)isolated. A node $v$ is isolated if and only if no edge incident to $v$ is inserted in each round. In each round, when $v$ is sampled, i.e., $v \in V_s$, the probability that no edge incident to $v$ is inserted is $1 - \max_{u \in V_s} p(u, v)$. Let $p_{iso}(v)$ denote the aforementioned probability and sort $V \setminus \{v\} = \{u_1, u_2, \ldots, u_{n-1}\}$ with $n = |V|$ and $p(u_1, v) \geq p(u_2, v) \geq \cdots \geq p(u_{n-1}, v)$. We have

$$p_{iso}(v) = (1 - \Pr[v \in V_s]) + \Pr[v \in V_s](1 - \mathbb{E}_{f^{\mathrm{PB}}_{p;g,R}}[\max_{u \in V_s} p(u, v)]) = 1 - g(v)\mathbb{E}_{f^{\mathrm{PB}}_{p;g,R}}[\max_{u \in V_s} p(u, v)],$$

where

$$\mathbb{E}_{f^{\mathrm{PB}}_{p;g,R}}[\max_{u \in V_s} p(u, v)]$$

$$= \Pr[u_1 \in V_s]p(u_1, v) + \Pr[u_1 \notin V_s \wedge u_2 \in V_s]p(u_2, v) + \cdots + \Pr[(\bigwedge_{i=1}^{n-2} u_i \notin V_s) \wedge u_{n-1} \in V_s]p(u_{n-1}, v)$$

$$= g(u_1)p(u_1, v) + (1 - g(u_1))g(u_2)p(u_2, v) + \cdots + \prod_{i=1}^{n-2}(1 - g(u_i))g(u_{n-1})p(u_{n-1}, v).$$

Finally, the probability that $v$ is isolated after $R$ rounds and dealing with $p_{rem}$ is

$$\tilde{p}_{iso}(v) = (p_{iso}(v))^R(1 - p_{rem}(v)),$$

and thus the expected number of non-isolated nodes is

$$\mathbb{E}_{f^{\mathrm{PB}}_{p;g,R}}[|\{v \in G \colon d(v;G) \geq 1\}|] = \sum_{v \in V}(1 - \tilde{p}_{iso}(v)).$$

$\square$

**The expected number of degree-1 nodes** We can extend the reasoning above to compute the expected number of degree-1 nodes. Fix a node $v$, for each node $u_k$, we shall compute the probability that no other $(u_{k'}, v)$ with $k' \neq k$ is inserted, denoted by $p_s(v; u_k)$, which is the probability of $v$ being isolated plus the probability of $v$ being only adjacent to $u_k$. In other words, we compute the probability of $v$ being isolated while ignoring $u_k$. We have

$$p_s(v; u_k) = (1 - g(v)) + g(v)\tilde{p}_s(v; u_k),$$

where

$$\tilde{p}_s(v; u_k) = g(u_1)(1 - p(u_1, v)) +$$
$$(1 - g(u_1))g(u_2)(1 - p(u_2, v)) + \cdots +$$
$$\prod_{i=1}^{k-2}(1 - g(u_i))g(u_{k-1})(1 - p(u_{k-1}, v)) +$$
$$\prod_{i=1}^{k-1}(1 - g(u_i))g(u_k)\hat{p}_s(v; u_k) +$$
$$\prod_{i=1}^{k+1}(1 - g(u_i))g(u_{k+1})(1 - p(u_{k+1}, v)) + \cdots +$$
$$\prod_{i=1}^{n-2}(1 - g(u_i))g(u_{n-1})p(u_{n-1}, v) +$$
$$\prod_{i=1}^{n-1}(1 - g(u_i))$$

with

$$\hat{p}_s(v; u_k) = g(u_{k+1})(1 - p(u_{k+1}, v)) +$$
$$(1 - g(u_{k+1}))g(u_{k+2})(1 - p(u_{k+2}, v)) + \cdots +$$
$$\prod_{i=k+1}^{n-2}(1 - g(u_i))g(u_{n-1})(1 - p(u_{n-1}, v)) +$$
$$\prod_{i=k+1}^{n-1}(1 - g(u_i)).$$

Finally, the probability of $v$ being degree-1 is

$$\sum_{i=1}^{n-1}(p_s(v; u_i) - p_{iso}(v)).$$

**Theorem C.2** (Time complexity of computing the expected number of (non-)isolated nodes with parallel binding). *Given* $p\colon \binom{V}{2} \to [0, 1]$, $g\colon V \to [0, 1]$, *and* $R \in \mathbb{N}$, *computing* $\mathbb{E}_{f_{p;g,R}^{PB}}[|\{v \in G\colon d(v; G) \geq 1\}|]$ *takes* $O(|V|^2 \log |V|)$ *time.*

*Proof.* For computing the expected number of non-isolated nodes, for each node $v$, we need to first sort the other nodes $u \in V \setminus \{v\}$ w.r.t. $p(u, v)$, which takes $O(|V| \log |V|)$ times. After that, the calculation only arithmetic operations, which takes $O(1)$ time since the formulae are fixed. Hence, for each node $v$ it takes $O(\log |V|)$ times. In conclusion, for all the nodes in $V$, it takes $O(|V| \log |V|)$ time in total. $\square$

*Remark* C.3. Considering node equivalence (see Section 5.4) can also be used to reduce the time complexity of computing the number of (non-)isolated nodes.

## C.2 EXPERIMENTAL RESULTS

Since we have the tractability results on the number of (non-)isolated nodes, we can also fit and control the number of (non-)isolated nodes with our binding schemes. Specifically, in our main experiments, the objective of fitting is merely the number of triangles. Here, we further consider variants with the fitting objective including both the number of triangles and the number of (non-)isolated nodes, trying to preserve both numbers as the ground truth.

In Table 5, for each dataset and each model, we compare the ground-truth graph, the corresponding EIGM, and the following two variants of EPGMs:

1. PARABDG: parallel binding with the number of triangles as the objective
2. PARABDG-N: parallel binding with both the number of triangles and the number of (non-)isolated nodes[12]

and report the following statistics of the generated graphs:

1. $n_{ni}$: the number of non-isolated nodes
2. $\triangle$: the number of triangles
3. GCC: the global clustering coefficient
4. ALCC: the average clustering coefficient

As in the main text, the statistics are averaged on 100 random trials, i.e., 100 generated graphs.

For ER, we relax both the number of total nodes and the uniform edge probability, i.e., $n_0$ and $p_0$, for fitting. For the other three models (CL, SB, and KR), we still use the edge probabilities obtained from the original model and only add an additional term to the objective.

As shown in the results, in most cases, PARABDG generates graphs with fewer non-isolated nodes compared to the ground truth, and PARABDG-n well fits the number of non-isolated nodes while still improving clustering compared to EIGMs. Notably, since the total number of nodes for KR can only

---

[12]We only have tractability results with parallel binding.

Table 5: The number of non-isolated nodes and clustering metrics of graphs generated by different realization methods. The number of non-isoalted nodes $n_{ni}$ and the number of triangles ($\triangle$) are normalized. For each dataset and each model, the best result is in bold and the second best is underlined.

| dataset | | *Hams* | | | | *Fcbk* | | | | *Polb* | | | |
|---|---|---|---|---|---|---|---|---|---|---|---|---|---|
| metric | | $n_{ni}$ | $\triangle$ | GCC | ALCC | $n_{ni}$ | $\triangle$ | GCC | ALCC | $n_{ni}$ | $\triangle$ | GCC | ALCC |
| model | GROUNDT | 1.000 | 1.000 | 0.229 | 0.540 | 1.000 | 1.000 | 0.519 | 0.606 | 1.000 | 1.000 | 0.226 | 0.320 |
| ER | EDGEIND | **1.000** | 0.013 | 0.008 | 0.008 | **1.000** | 0.009 | 0.011 | 0.011 | **1.000** | 0.034 | 0.022 | **0.022** |
| | PARABDG | 0.812 | 0.988 | **0.385** | **0.640** | 0.555 | **1.002** | 0.574 | **0.815** | 0.801 | 1.025 | **0.412** | 0.659 |
| | PARABDG-N | 0.996 | **0.990** | 0.481 | 0.748 | 1.007 | 0.584 | **0.594** | **0.835** | 1.007 | **1.012** | 0.532 | 0.787 |
| CL | EDGEIND | **0.964** | 0.299 | 0.067 | 0.058 | **0.988** | 0.124 | 0.064 | 0.063 | **0.944** | 0.792 | 0.183 | 0.173 |
| | PARABDG | 0.771 | **1.000** | **0.185** | **0.471** | 0.656 | **1.006** | **0.336** | **0.626** | 0.789 | **1.010** | **0.221** | 0.468 |
| | PARABDG-N | 0.959 | 0.257 | 0.027 | 0.069 | 0.969 | 1.098 | 0.125 | 0.151 | 0.935 | 0.794 | 0.135 | **0.219** |
| SB | EDGEIND | **0.996** | 0.263 | 0.080 | 0.038 | **1.000** | 0.153 | 0.145 | 0.080 | **0.975** | 0.478 | 0.145 | 0.164 |
| | PARABDG | 0.719 | **0.993** | **0.241** | **0.521** | 0.608 | **1.035** | **0.529** | **0.557** | 0.899 | **1.010** | **0.183** | **0.251** |
| | PARABDG-N | 0.991 | 1.168 | 0.154 | 0.092 | 1.000 | 1.036 | 0.423 | 0.204 | 0.953 | 0.475 | 0.094 | 0.217 |
| KR | EDGEIND | **0.996** | 0.185 | 0.039 | 0.060 | 1.014 | 0.052 | 0.035 | 0.042 | 1.598 | 0.101 | 0.040 | 0.075 |
| | PARABDG | 0.856 | **0.997** | **0.165** | **0.394** | 0.781 | **0.971** | **0.347** | **0.605** | 1.194 | 0.942 | **0.219** | 0.420 |
| | PARABDG-N | 0.996 | 0.301 | 0.028 | 0.099 | **1.000** | 0.953 | 0.254 | 0.262 | **0.987** | **0.976** | 0.268 | **0.368** |

| dataset | | *Spam* | | | | *Cepg* | | | | *Scht* | | | |
|---|---|---|---|---|---|---|---|---|---|---|---|---|---|
| metric | | $n_{ni}$ | $\triangle$ | GCC | ALCC | $n_{ni}$ | $\triangle$ | GCC | ALCC | $n_{ni}$ | $\triangle$ | GCC | ALCC |
| model | GROUNDT | 1.000 | 1.000 | 0.145 | 0.286 | 1.000 | 1.000 | 0.321 | 0.447 | 1.000 | 1.000 | 0.377 | 0.350 |
| ER | EDGEIND | **1.000** | 0.005 | **0.003** | **0.003** | **1.000** | 0.037 | 0.033 | 0.033 | **1.000** | 0.027 | 0.029 | **0.029** |
| | PARABDG | 0.783 | **0.993** | 0.401 | 0.663 | 0.688 | **0.968** | **0.508** | **0.750** | 0.617 | **0.991** | 0.559 | 0.794 |
| | PARABDG-N | 1.006 | 1.009 | 0.526 | 0.787 | 1.008 | 0.832 | 0.606 | 0.839 | 1.002 | 0.669 | **0.604** | **0.839** |
| CL | EDGEIND | 0.906 | 0.496 | 0.072 | 0.060 | **0.953** | 0.683 | 0.230 | 0.223 | **0.964** | 0.644 | 0.245 | 0.234 |
| | PARABDG | 0.700 | **1.007** | **0.131** | **0.436** | 0.698 | **0.999** | **0.310** | **0.578** | 0.866 | **1.135** | **0.294** | **0.610** |
| | PARABDG-N | **0.908** | 0.445 | 0.033 | 0.071 | 0.927 | 0.725 | 0.198 | **0.334** | 0.932 | 0.639 | 0.200 | **0.347** |
| SB | EDGEIND | **0.982** | 0.528 | 0.094 | 0.036 | **0.994** | 0.662 | 0.258 | 0.200 | **0.992** | 0.644 | 0.272 | 0.128 |
| | PARABDG | 0.685 | **0.994** | **0.158** | **0.356** | 0.911 | **1.047** | 0.333 | **0.363** | 0.792 | **0.975** | **0.340** | **0.437** |
| | PARABDG-N | 0.957 | 0.537 | 0.070 | 0.109 | 0.990 | 1.056 | **0.329** | 0.202 | 0.972 | 0.956 | 0.292 | 0.205 |
| KR | EDGEIND | 1.438 | 0.061 | 0.014 | 0.025 | 1.210 | 0.132 | 0.069 | 0.120 | 1.953 | 0.032 | 0.033 | 0.052 |
| | PARABDG | 1.024 | 1.049 | 0.161 | **0.378** | 1.043 | **1.001** | 0.279 | **0.461** | 1.211 | 1.069 | 0.346 | **0.581** |
| | PARABDG-N | **0.995** | **0.981** | **0.161** | 0.385 | **0.996** | 1.118 | **0.296** | 0.478 | **0.997** | **1.030** | **0.370** | 0.640 |

be a power of the seed-matrix size (i.e., a power of 2 in our experiments), the corresponding EIGM generates graphs with too many non-isolated nodes in many cases, while PARABDG-n generates graphs with a more similar number of non-isolated nodes (i.e., closer to the ground truth). Moreover, it is also known that even without binding, some models may suffer from the problem of isolated nodes, e.g., CL (Brissette & Slota, 2021; Brissette et al., 2022) and KR (Mahdian & Xu, 2007; Seshadri et al., 2013).

Overall, the results validate that, our tractability results allow practitioners to fit the number of non-isolated nodes (if that is one of their main concerns) while improving other aspects, e.g., clustering.

# D ADDITIONAL DISCUSSIONS

## D.1 GENERAL GRAPHS

As mentioned in Section 2, we focus on undirected unweighted graphs without self-loops following common settings for random graph models in the main text. Below, we shall discuss different more general cases.

**Directed edges and self-loops.** In our binding schemes (Algorithms 1 to 3), if we consider directed edges and/or self-loops, we can further consider them after sampling a group of nodes. Regarding theoretical analysis, we can further consider subgraphs (motifs) with directed edges and self-loops (Milo et al., 2002) and the high-level ideas still apply.

**Weighted edges.** Our graph generation algorithms only determine the (in)existence of edges and we may need additional schemes to generate edge weights. For example, we can use algorithms that generate proper edge weights when given graph topology (Bu et al., 2023). Since in our graph generation algorithms, nodes (and thus edges) can be sampled multiple times, an alternative way to

have edge weights is to allow each edge to be inserted multiple times and use the times of repetition as edge weights.

## D.2 OVERLAP-RELATED TRIANGLE-DENSITY RESULTS

As mentioned in Section 3.1, Chanpuriya et al. (2024) have recently extended their theoretical analysis to other categories of RGMs. In addition to EIGMs, they further considered two other categories: node independent graph models (NIGMs) and fully dependent graph models (FDGMs). Between the two, FDGMs means any distribution of graphs, i.e., any RGM, is allowed.

They only discussed general overlap-related triangle-density upper bounds in those categories of RGMs, without detailed tractability results for practical graph generations. Specifically, their graph generation algorithm is based on maximal clique enumeration (MCE). However, given a graph, MCE itself can take exponential time (Eblen et al., 2012).

Also, what we focus on in this work, i.e., the category of binding-based EPGMs, is a subset of EPGMs and are not "fully general" as FDGMs. On the other hand, NIGMs are associated with node embeddings, where we have a node embedding space (i.e., a distribution) $\mathcal{E}$ and a symmetric function $e\colon \mathcal{E} \times \mathcal{E} \to [0, 1]$, and each node $i$ has a node embedding $\boldsymbol{x}_i$ sampled from $\mathcal{E}$ i.i.d., and each edge $(i, j)$ exists with probability $e(\boldsymbol{x}_i, \boldsymbol{x}_j)$ independently. Our binding-based EPGMs do not fall in this category either.

## D.3 SUBSET SAMPLING

As mentioned in Footnote 7 in Section 5.2, we use independent *node* sampling (yet still with *edge* dependency) which is simple, tractable, and works well. Specifically, independent node sampling allows us to easily compute the marginal probability of each node binding sampled in each round, which is involved in the derivation of our tractability results. Also, as shown in our experiments, with binding schemes using independent node sampling, we still achieve significant empirical improvement over EIGMs. In the most general case, considering the sampling probabilities of all $2^{|V|}$ subsets would be intractable. Recently, a line of works has been proposed for tractable and differentiable subset sampling (Xie & Ermon, 2019; Pervez et al., 2023; Ahmed et al., 2023; Sutter et al., 2023), and exploring more flexible node sampling schemes is an interesting future direction to be explored.

## D.4 PRACTICAL MEANING OF BINDING

As we mentioned in Section 5.2, local binding (and parallel binding as a parallel version) binds node pairs *locally among a group of nodes* (instead of some irrelevant node pairs). Such node pairs are structurally related, and are expected to be meaningfully related in the corresponding real-world systems. We shall discuss two specific real-world scenarios below.

**Group interactions in social networks.** In typical social networks, nodes represent people, and edges represent social communications/relations between people. Each group "bound together" by our binding algorithms can represent a group interaction, e.g., an offline social event (meeting, conference, party) or an online social event (group chat, Internet forum, online game). In such social events, people gather together and the communications/relations between them likely co-occur. Certainly, not necessarily all people in such events would communicate with each other, e.g., some people are more familiar with each other. This is exactly the point of considering binding with various edge probabilities (instead of just inserting cliques).

Specifically, the random variable $s$ represents the overall "social power" of an event, while individual edge probabilities $p(u, v)$'s represent some local factors (e.g., their personal relationship) between each pair of people. A line of research studies group interactions in social networks (Felmlee & Faris, 2013; Levorato, 2014; Purushotham & Jay Kuo, 2015; Jang et al., 2016; Li et al., 2020; Iacopini et al., 2022).

**Gene functional associations in gene networks.** In typical gene networks, nodes represent genes, and edges represent gene functional associations, i.e., connections between genes that contribute jointly to a biological function. Each group "bound together" by our binding algorithms can represent a biological function, since typically (1) a single biological function involves multiple genes (Plomin, 1990; Anastassiou, 2007; Naoumkina et al., 2010) (represented by a group of nodes bound together), and (2) the same biological function may involve different genes in different

Table 6: The basic statistics of the datasets.

| dataset | $|V|$ | $|E|$ | # triangles | GCC | ALCC |
|---------|-------|-------|-------------|-----|------|
| *Hams* | 2,000 | 16,097 | 157,953 | 0.229 | 0.540 |
| *Fcbk* | 4,039 | 88,234 | 4,836,030 | 0.519 | 0.606 |
| *Polb* | 1,222 | 16,717 | 303,129 | 0.226 | 0.320 |
| *Spam* | 4,767 | 37,375 | 387,051 | 0.145 | 0.286 |
| *Cepg* | 1,692 | 47,309 | 2,353,812 | 0.321 | 0.447 |
| *Scht* | 2,077 | 63,023 | 4,192,980 | 0.377 | 0.350 |

cases (Gottesman & Hanson, 2005; Pritykin et al., 2015; Storey et al., 2007) (represented by the probabilistic nature of binding).

**On parallel binding.** Specifically, as mentioned in Section 5.3, compared to local binding where each pair can only participate in a single group, parallel binding allows each pair to participate in multiple groups (in different rounds). This is also true for real-world group interactions, where different groups overlap and intersect with each other (Lee et al., 2021; LaRock & Lambiotte, 2023).

## E  ADDITIONAL DETAILS OF THE EXPERIMENTS

### E.1  EXPERIMENTAL SETTINGS

**Datasets.** We use six real-world datasets from three different domains: (1) social networks *hamsterster (Hams)* and *facebook (Fcbk)*, (2) web graphs *polblogs (Polb)* and *spam (Spam)*, and (3) biological graphs *CE-PG (Cepg)* and *SC-HT (Scht)*.

The datasets are available online (Rossi & Ahmed, 2015; Leskovec & Krevl, 2014):

- *hamsterster (Hams)* (Hamsterster) is available at `https://networkrepository.com/soc-hamsterster.php`
- *facebook (Fcbk)* (Leskovec & Mcauley, 2012) is available at `https://snap.stanford.edu/data/ego-Facebook.html`
- *polblogs (Polb)* (Adamic & Glance, 2005) is available at `https://networks.skewed.de/net/polblogs`
- *spam (Spam)* (Castillo et al., 2008) is available at `https://networkrepository.com/web-spam.php`
- *CE-PG (Cepg)* (Cho et al., 2014) is available at `https://networkrepository.com/bio-CE-PG.php`
- *SC-HT (Scht)* (Cho et al., 2014) is available at `https://networkrepository.com/bio-SC-HT.php`

In Table 6, we show the basic statistics (e.g., the numbers of nodes and edges) of the datasets.

We provide the formal definitions of some basic statistics below.

**Definition E.1** (Clustering coefficients). Given $G = (V, E)$, the number of wedges (i.e., open triangles) is $n_w(G) = \sum_{v \in V} \binom{d(v)}{2}$. The *global clustering coefficient* (GCC) of $G$ is defined as

$$\text{GCC}(G) = \frac{3\triangle(G)}{n_w(G)},$$

where $\triangle(G)$ is the number of triangles in $G$ and it is multiplied by 3 because each triangle corresponds to three wedges (consider three different nodes as the center of the wedge). The *average local clustering coefficient* (ALCC) of $G$ is defined as

$$\text{ALCC}(G) = \sum_{v \,:\, d(v) \geq 2} \frac{\triangle(v; G)}{\binom{d(v)}{2}},$$

where $\triangle(v; G)$ is the number of triangles involving $v$ in $G$.

**Models.** The Erdős-Rényi (ER) model outputs edge probabilities with two parameters: $n_0$ and $p_0$, and the output is $p_{n_0,p_0}^{ER}$ with $p_{n_0,p_0}^{ER}(u,v) = p_0, \forall u, v \in \binom{V}{2}$ with $V = [n_0]$. Given a graph $G = (V = [n], E)$, the standard fitting of ER gives $n_0 = n$ and $p_0 = \frac{|E|}{\binom{|V|}{2}}$.

The Chung-Lu (CL) model outputs edge probabilities with a sequence of expected degrees $D = (d_1, d_2, \ldots, d_n)$, and the output is $p_D^{CL}$ with $p_D^{CL}(u, v) = \min(\frac{d_u d_v}{\sum_{i=1}^{n} d_i}, 1), \forall u, v \in \binom{V}{2}$ with $V = [n]$. Given a graph $G = (V = [n], E)$, the standard fitting of CL gives $d_i = d(i; G)$ for each node $i \in V$.

The stochastic block (SB) model outputs edge probabilities with (1) a partition of nodes which can be represented by an assignment function $f_B \colon [n_0] \to [c]$ with $n_0$ nodes and $c$ blocks and (2) the edge probability between each pair of blocks (including between two identical blocks), which can be represented by $p_B \colon [c] \times [c] \to [0, 1]$, and the output is $p_{f_B, p_B}^{SB}$ with $p_{f_B, p_B}^{SB}(u, v) = p_B(f_B(u), f_B(v)), \forall u, v \in [n_0]$. In our experiments, we use the Python library Graspologic (Chung et al., 2019) which contains a fitting algorithm for SB. Specifically, it uses spectral embedding (Von Luxburg, 2007; Sussman et al., 2012; Rohe et al., 2011) and a Gaussian mixture model (Reynolds et al., 2009) to obtain node partitions.

The stochastic Kronecker (KR) model outputs edge probabilities with a seed matrix $\theta \in [0, 1]^{2 \times 2}$ and a Kronecker power $k_{KR} \in \mathbb{N}$, and the output is $p_{\theta, k_{KR}}^{KR}$ with $p_{\theta, k_{KR}}^{KR}(u, v) = \theta_{uv}^{(k_{KR})}, \forall u, v \in \binom{V}{2}$ with $V = [2^{k_{KR}}]$, where $\theta^{(k_{KR})} \in [0, 1]^{2^{k_{KR}} \times 2^{k_{KR}}}$ is the $k_{KR}$-th Kronecker power of $\theta$. In our experiments, we use `kronfit` (Leskovec et al., 2010) proposed by the original authors of KR.

**Fitting.** For fitting the parameters for our binding schemes, we use the Adam optimizer (Kingma & Ba, 2015) with learning rate $\eta = 0.001$ and $n_{ep} = 10,000$ epochs for training. In our experiments, we consistently use $R = 100,000$ rounds for both of our binding schemes. By default, the input edge probabilities $p$ are provided and fixed as described above. By default, the objective is the expected number of triangles. More specifically, it is

$$(1 - \frac{\mathbb{E}_{f_{p;g,R}^{X}}[\triangle(G)]}{\triangle(G_{input})})^2$$

where

$$\mathbb{E}_{f_{p;g,R}^{X}}[\triangle(G)] = \sum_{V' \in \binom{V}{3}} \Pr_{f_{p;g,R}^{X}}[E(G[V']) = \binom{V'}{2}]$$

is the expected number of triangles in a generated graph with $X \in \{\text{LOCLBDG}, \text{PARABDG}\}$ indicating the binding scheme, and $\triangle(G_{input})$ is the ground-truth number of triangles in the input graph.

We observe that our fitting algorithms assign different node-sampling probabilities to different nodes, which implies that different nodes have different levels of importance in binding. In Figure 3, for the CL model and for each dataset, we show the relations between nodes' degrees and their node-sampling probabilities in LOCLBDG and PARABDG. For LOCLBDG, we observe strong positive correlations between node degrees and node-sampling probabilities. For PARABDG, similar trends are observed, but the patterns are quite different. Also, we can observe that the node-sampling probabilities for PARABDG are overall lower than those for LOCLBDG, as mentioned in Section 6.4.

**Hardware and software.** All the experiments of fitting are run on a machine with two Intel Xeon® Silver 4210R (10 cores, 20 threads) processors, a 512GB RAM, and RTX A6000 (48GB) GPUs. A single GPU is used for each fitting process. The code for fitting is written in Python, using Pytorch (Paszke et al., 2019). All the experiments of graph generation are run on a machine with one Intel i9-10900K (10 cores, 20 threads) processor, a 64GB RAM. The code for generation is written in C++, compiled with G++ with O2 optimization and OpenMP (Dagum & Menon, 1998) parallelization.

### E.2  P1: CLUSTERING

As mentioned in Section 6.2, the results in Table 1 are averaged on 100 random trials. In Table 7, we show the full results with standard deviations. With binding, the variance is higher since the covariances between edges are higher with dependency. We also compute the mean squared errors w.r.t. each metric. The results are in Table 8. Notably, for graph generators, variability is desirable in many cases (Moreno et al., 2018; Stamm et al., 2023).

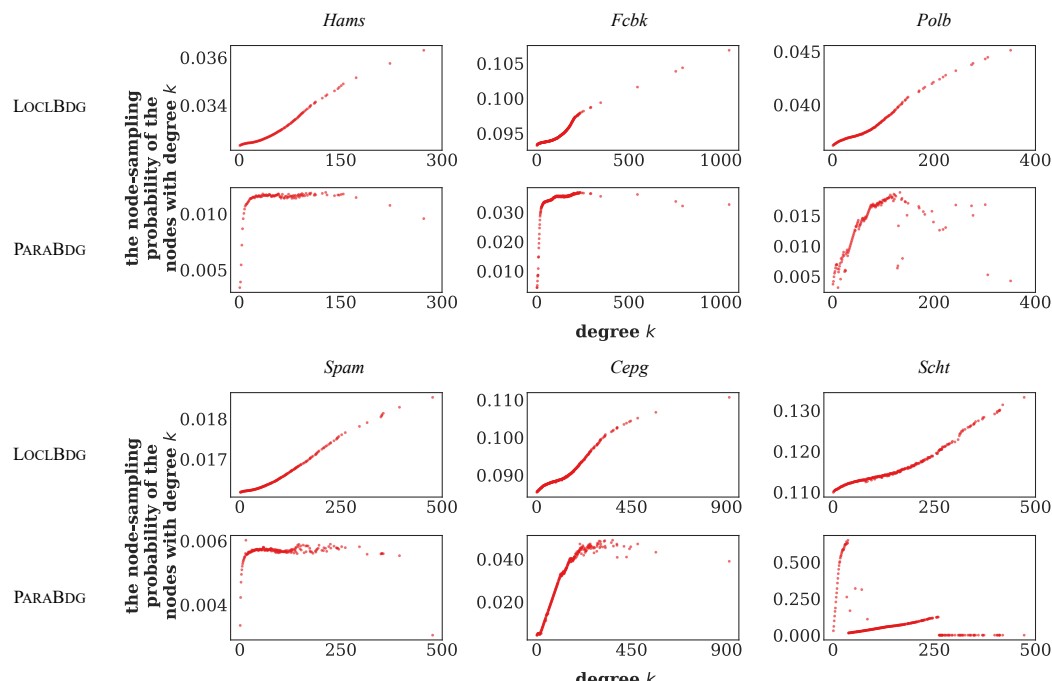

Figure 3: The relations between node degrees and node-sampling probabilities.

### E.3 P2: DEGREES, DISTANCES, AND OTHER GRAPH STATISTICS

**Definition E.2** (Paths and distance). Given a graph $G = (V, E)$, a sequences of nodes $(v_1, v_2, \ldots, v_t)$ consisting of $t$ distinct nodes is a *path* between $v_1$ and $v_t$, if $(v_i, v_{i+1}) \in E, \forall i \in [t-1]$, and $t$ is called the length of the path. Given two nodes $u, v \in V$, the *distance* between $u$ and $v$ is the length of the shortest path between $u$ and $v$.

**Definition E.3** (Connected components). Given a graph $G = (V, E)$, and two nodes $u, v \in V$, we say $u$ and $v$ are in the same *connected component*, if and only if there exists at least one path between $u$ and $v$. This relation of "being in the same connected component" forms equivalent classes among the nodes, and each equivalent class is a connected component. A *largest connected component* is a connected component with the largest size (i.e., the number of nodes in it).[13]

In Figure 4, for each dataset (each column) and each model (each row), we compare the degree distributions and distance distributions in the ground-truth graph and the graphs generated with each realization method, supplementing Figure 1.

In Table 9, we provide the detailed numerical results w.r.t. degrees and distances. Specifically, for each dataset, each mode, and each realization method, we report the following statistics:

- the results of the linear regression of node degrees $k$ and the number of nodes with each degree $k$ on a log-log scale: the fit slope (the exponent $\alpha$ in the corresponding power-law fitting) and the $r$ value (the strength of a power law)
- the average path length (APL) and the 90%-effective diameter ($d_{eff}$) in the largest connected component[14]

With binding, the generated graphs are overall closer to ground truth w.r.t. some other graph metrics: modularity (Newman, 2006), conductance (Gleich, 2006), core numbers (Seidman, 1983), average vertex betweenness (Freeman, 1977), average edge betweenness (Brandes, 2008), and natural connectivity (Chan et al., 2014). See Tables 11 to 16 for the detailed results. Modularity is computed

---

[13]A graph may contain several equal-size largest connected components, but it rarely happens for real-world graphs.

[14]The average path length is the average distance of the pairs in the largest connected component, and the 90%-effective diameter is the minimum distance $d$ such that at least 90% of the pairs in the largest connected component have distances at most $d$.

Table 7: The clustering metrics of graphs generated by different realization methods, with the standard deviations. The number of triangles ($\triangle$) is normalized.

| dataset | | Hams | | | Fcbk | | | Polb | | |
|---|---|---|---|---|---|---|---|---|---|---|
| metric | | $\triangle$ | GCC | ALCC | $\triangle$ | GCC | ALCC | $\triangle$ | GCC | ALCC |
| model | GROUNDT | 1.000 | 0.229 | 0.540 | 1.000 | 0.519 | 0.606 | 1.000 | 0.226 | 0.320 |
| | EDGEIND | 0.013 | 0.008 | 0.008 | 0.009 | 0.011 | 0.011 | 0.034 | 0.022 | 0.022 |
| | (std) | 0.001 | 0.000 | 0.000 | 0.000 | 0.000 | 0.000 | 0.001 | 0.000 | 0.000 |
| ER | LOCLBDG | 0.997 | 0.321 | 0.236 | 1.010 | 0.448 | 0.223 | 0.955 | 0.336 | 0.247 |
| | (std) | 0.279 | 0.028 | 0.022 | 0.445 | 0.077 | 0.042 | 0.320 | 0.038 | 0.032 |
| | PARABDG | 0.988 | 0.385 | 0.640 | 1.002 | 0.574 | 0.815 | 1.025 | 0.412 | 0.659 |
| | (std) | 0.081 | 0.014 | 0.018 | 0.155 | 0.036 | 0.026 | 0.135 | 0.022 | 0.028 |
| | EDGEIND | 0.299 | 0.067 | 0.058 | 0.124 | 0.064 | 0.063 | 0.792 | 0.183 | 0.173 |
| | (std) | 0.010 | 0.002 | 0.002 | 0.002 | 0.001 | 0.001 | 0.017 | 0.002 | 0.005 |
| CL | LOCLBDG | 0.992 | 0.165 | 0.255 | 1.026 | 0.255 | 0.305 | 1.002 | 0.214 | 0.341 |
| | (std) | 0.353 | 0.030 | 0.026 | 1.033 | 0.095 | 0.050 | 0.132 | 0.008 | 0.021 |
| | PARABDG | 1.000 | 0.185 | 0.471 | 1.006 | 0.336 | 0.626 | 1.010 | 0.221 | 0.468 |
| | (std) | 0.144 | 0.013 | 0.013 | 0.261 | 0.035 | 0.018 | 0.068 | 0.003 | 0.009 |
| | EDGEIND | 0.263 | 0.080 | 0.038 | 0.153 | 0.145 | 0.080 | 0.478 | 0.145 | 0.164 |
| | (std) | 0.007 | 0.001 | 0.001 | 0.002 | 0.001 | 0.000 | 0.012 | 0.002 | 0.004 |
| SB | LOCLBDG | 1.039 | 0.219 | 0.240 | 0.934 | 0.429 | 0.331 | 0.994 | 0.237 | 0.355 |
| | (std) | 0.419 | 0.042 | 0.026 | 0.732 | 0.086 | 0.074 | 0.386 | 0.025 | 0.037 |
| | PARABDG | 0.993 | 0.241 | 0.521 | 1.035 | 0.529 | 0.557 | 1.010 | 0.183 | 0.251 |
| | (std) | 0.118 | 0.013 | 0.012 | 0.504 | 0.064 | 0.042 | 1.819 | 0.076 | 0.054 |
| | EDGEIND | 0.185 | 0.039 | 0.060 | 0.052 | 0.035 | 0.042 | 0.101 | 0.040 | 0.075 |
| | (std) | 0.006 | 0.001 | 0.002 | 0.001 | 0.000 | 0.001 | 0.003 | 0.001 | 0.003 |
| KR | LOCLBDG | 1.095 | 0.152 | 0.230 | 0.927 | 0.239 | 0.270 | 1.061 | 0.141 | 0.234 |
| | (std) | 0.580 | 0.047 | 0.028 | 1.090 | 0.117 | 0.048 | 2.234 | 0.106 | 0.054 |
| | PARABDG | 0.997 | 0.165 | 0.394 | 0.971 | 0.347 | 0.605 | 0.942 | 0.219 | 0.420 |
| | (std) | 0.210 | 0.021 | 0.016 | 0.395 | 0.055 | 0.017 | 0.601 | 0.075 | 0.035 |

| dataset | | Spam | | | Cepg | | | Scht | | |
|---|---|---|---|---|---|---|---|---|---|---|
| metric | | $\triangle$ | GCC | ALCC | $\triangle$ | GCC | ALCC | $\triangle$ | GCC | ALCC |
| model | GROUNDT | 1.000 | 0.145 | 0.286 | 1.000 | 0.321 | 0.447 | 1.000 | 0.377 | 0.350 |
| | EDGEIND | 0.005 | 0.003 | 0.003 | 0.037 | 0.033 | 0.033 | 0.027 | 0.029 | 0.029 |
| | (std) | 0.000 | 0.000 | 0.000 | 0.001 | 0.000 | 0.000 | 0.000 | 0.000 | 0.000 |
| ER | LOCLBDG | 0.993 | 0.336 | 0.234 | 1.016 | 0.397 | 0.258 | 1.012 | 0.420 | 0.251 |
| | (std) | 0.158 | 0.022 | 0.013 | 0.557 | 0.083 | 0.057 | 0.687 | 0.094 | 0.063 |
| | PARABDG | 0.993 | 0.401 | 0.663 | 0.968 | 0.508 | 0.750 | 0.991 | 0.559 | 0.794 |
| | (std) | 0.047 | 0.010 | 0.011 | 0.183 | 0.039 | 0.038 | 0.198 | 0.043 | 0.035 |
| | EDGEIND | 0.496 | 0.072 | 0.060 | 0.683 | 0.230 | 0.223 | 0.644 | 0.245 | 0.234 |
| | (std) | 0.010 | 0.001 | 0.002 | 0.008 | 0.001 | 0.004 | 0.006 | 0.001 | 0.003 |
| CL | LOCLBDG | 1.028 | 0.124 | 0.260 | 0.996 | 0.293 | 0.430 | 1.036 | 0.318 | 0.469 |
| | (std) | 0.214 | 0.016 | 0.019 | 0.241 | 0.018 | 0.033 | 0.367 | 0.028 | 0.042 |
| | PARABDG | 1.007 | 0.131 | 0.436 | 0.999 | 0.310 | 0.578 | 1.135 | 0.294 | 0.610 |
| | (std) | 0.074 | 0.006 | 0.011 | 0.107 | 0.004 | 0.010 | 1.290 | 0.079 | 0.033 |
| | EDGEIND | 0.528 | 0.094 | 0.036 | 0.662 | 0.258 | 0.200 | 0.644 | 0.272 | 0.128 |
| | (std) | 0.013 | 0.002 | 0.001 | 0.008 | 0.002 | 0.002 | 0.006 | 0.001 | 0.001 |
| SB | LOCLBDG | 0.985 | 0.152 | 0.223 | 0.986 | 0.323 | 0.415 | 1.034 | 0.354 | 0.386 |
| | (std) | 0.171 | 0.018 | 0.017 | 0.450 | 0.037 | 0.046 | 0.368 | 0.034 | 0.042 |
| | PARABDG | 0.994 | 0.158 | 0.356 | 1.047 | 0.333 | 0.363 | 0.975 | 0.340 | 0.437 |
| | (std) | 0.110 | 0.013 | 0.017 | 0.541 | 0.085 | 0.056 | 0.298 | 0.045 | 0.030 |
| | EDGEIND | 0.061 | 0.014 | 0.025 | 0.132 | 0.069 | 0.120 | 0.032 | 0.033 | 0.052 |
| | (std) | 0.002 | 0.000 | 0.001 | 0.002 | 0.001 | 0.002 | 0.001 | 0.000 | 0.001 |
| KR | LOCLBDG | 0.943 | 0.118 | 0.187 | 0.990 | 0.175 | 0.312 | 1.444 | 0.181 | 0.277 |
| | (std) | 0.759 | 0.055 | 0.028 | 2.112 | 0.098 | 0.077 | 3.610 | 0.132 | 0.079 |
| | PARABDG | 1.049 | 0.161 | 0.378 | 1.001 | 0.279 | 0.461 | 1.069 | 0.346 | 0.581 |
| | (std) | 0.319 | 0.032 | 0.017 | 0.757 | 0.098 | 0.044 | 1.165 | 0.152 | 0.035 |

after obtaining partitions using the Louvain algorithm (Blondel et al., 2008). Conductance is computed after obtaining bi-partitions using the Kernighan-Lin bisection algorithm (Kernighan & Lin,

Table 8: The mean squared errors w.r.t. clustering metrics of graphs generated by different realization methods. The number of triangles ($\triangle$) is normalized.

| dataset | | *Hams* | | | *Fcbk* | | | *Polb* | | |
|---------|---------|---------|------|------|---------|------|------|---------|------|------|
| metric | | $\triangle$ | GCC | ALCC | $\triangle$ | GCC | ALCC | $\triangle$ | GCC | ALCC |
| model | GROUNDT | 0.000 | 0.000 | 0.000 | 0.000 | 0.000 | 0.000 | 0.000 | 0.000 | 0.000 |
| ER | EDGEIND | 0.974 | 0.049 | 0.283 | 0.983 | 0.258 | 0.354 | 0.934 | 0.042 | 0.089 |
| | LOCLBDG | 0.078 | 0.009 | 0.093 | 0.199 | 0.011 | 0.148 | 0.104 | 0.013 | 0.007 |
| | PARABDG | 0.007 | 0.024 | 0.010 | 0.024 | 0.004 | 0.044 | 0.019 | 0.035 | 0.115 |
| CL | EDGEIND | 0.492 | 0.026 | 0.233 | 0.767 | 0.207 | 0.295 | 0.044 | 0.002 | 0.022 |
| | LOCLBDG | 0.125 | 0.005 | 0.082 | 1.068 | 0.079 | 0.093 | 0.017 | 0.000 | 0.001 |
| | PARABDG | 0.021 | 0.002 | 0.005 | 0.068 | 0.035 | 0.001 | 0.005 | 0.000 | 0.022 |
| SB | EDGEIND | 0.544 | 0.022 | 0.252 | 0.718 | 0.140 | 0.276 | 0.273 | 0.007 | 0.025 |
| | LOCLBDG | 0.177 | 0.002 | 0.091 | 0.539 | 0.015 | 0.081 | 0.149 | 0.001 | 0.002 |
| | PARABDG | 0.014 | 0.000 | 0.001 | 0.255 | 0.004 | 0.004 | 3.303 | 0.008 | 0.008 |
| KR | EDGEIND | 0.664 | 0.036 | 0.230 | 0.898 | 0.234 | 0.317 | 0.809 | 0.034 | 0.060 |
| | LOCLBDG | 0.346 | 0.008 | 0.097 | 1.194 | 0.092 | 0.115 | 4.989 | 0.018 | 0.010 |
| | PARABDG | 0.044 | 0.005 | 0.022 | 0.157 | 0.033 | 0.000 | 0.364 | 0.006 | 0.011 |

| dataset | | *Spam* | | | *Cepg* | | | *Scht* | | |
|---------|---------|---------|------|------|---------|------|------|---------|------|------|
| metric | | $\triangle$ | GCC | ALCC | $\triangle$ | GCC | ALCC | $\triangle$ | GCC | ALCC |
| model | GROUNDT | 0.000 | 0.000 | 0.000 | 0.000 | 0.000 | 0.000 | 0.000 | 0.000 | 0.000 |
| ER | EDGEIND | 0.990 | 0.020 | 0.080 | 0.927 | 0.083 | 0.171 | 0.947 | 0.121 | 0.103 |
| | LOCLBDG | 0.025 | 0.037 | 0.003 | 0.310 | 0.013 | 0.039 | 0.473 | 0.011 | 0.014 |
| | PARABDG | 0.002 | 0.066 | 0.143 | 0.035 | 0.037 | 0.093 | 0.039 | 0.035 | 0.198 |
| CL | EDGEIND | 0.254 | 0.005 | 0.051 | 0.100 | 0.008 | 0.050 | 0.126 | 0.017 | 0.014 |
| | LOCLBDG | 0.046 | 0.001 | 0.001 | 0.058 | 0.001 | 0.001 | 0.136 | 0.004 | 0.016 |
| | PARABDG | 0.006 | 0.000 | 0.023 | 0.012 | 0.000 | 0.017 | 1.682 | 0.013 | 0.069 |
| SB | EDGEIND | 0.223 | 0.003 | 0.062 | 0.114 | 0.004 | 0.061 | 0.127 | 0.011 | 0.049 |
| | LOCLBDG | 0.030 | 0.000 | 0.004 | 0.202 | 0.001 | 0.003 | 0.136 | 0.002 | 0.003 |
| | PARABDG | 0.012 | 0.000 | 0.005 | 0.295 | 0.007 | 0.010 | 0.089 | 0.003 | 0.008 |
| KR | EDGEIND | 0.882 | 0.017 | 0.068 | 0.754 | 0.064 | 0.107 | 0.936 | 0.118 | 0.089 |
| | LOCLBDG | 0.579 | 0.004 | 0.011 | 4.462 | 0.031 | 0.024 | 13.233 | 0.056 | 0.012 |
| | PARABDG | 0.104 | 0.001 | 0.009 | 0.573 | 0.011 | 0.002 | 1.361 | 0.024 | 0.054 |

1970). In most cases, the metrics in the graphs generated with binding are closer to the ground truth, indicating that binding improves the generation quality in various aspects.

### E.4 GRAPH GENERATION SPEED

In Table 10, for each dataset and each model, we report the running time of graph generation (averaged on 100 random trials) using EDGEIND, LOCLBDG, PARABDG, and serialized PARABDG without parallelization (PARABDG-S). The algorithmic details of EDGEIND for each model are as follows:

- We try to find an optimized and fast algorithm for each model in C++
- For ER, we use the Boost Graph Library (Siek et al., 2001)
- For CL, we use NetworKit (Staudt et al., 2016)
- For SB, we use online code in a GitHub repo[15]
- For KR, we use `krongen` in SNAP (Leskovec & Sosič, 2016)

Consistent with our observation in Section 6.4, EDGEIND is fastest with the simplest algorithmic nature, and between the two binding schemes, PARABDG is noticeably faster than LOCLBDG, and is even faster with parallelization.

---

[15]https://github.com/ntamas/blockmodel

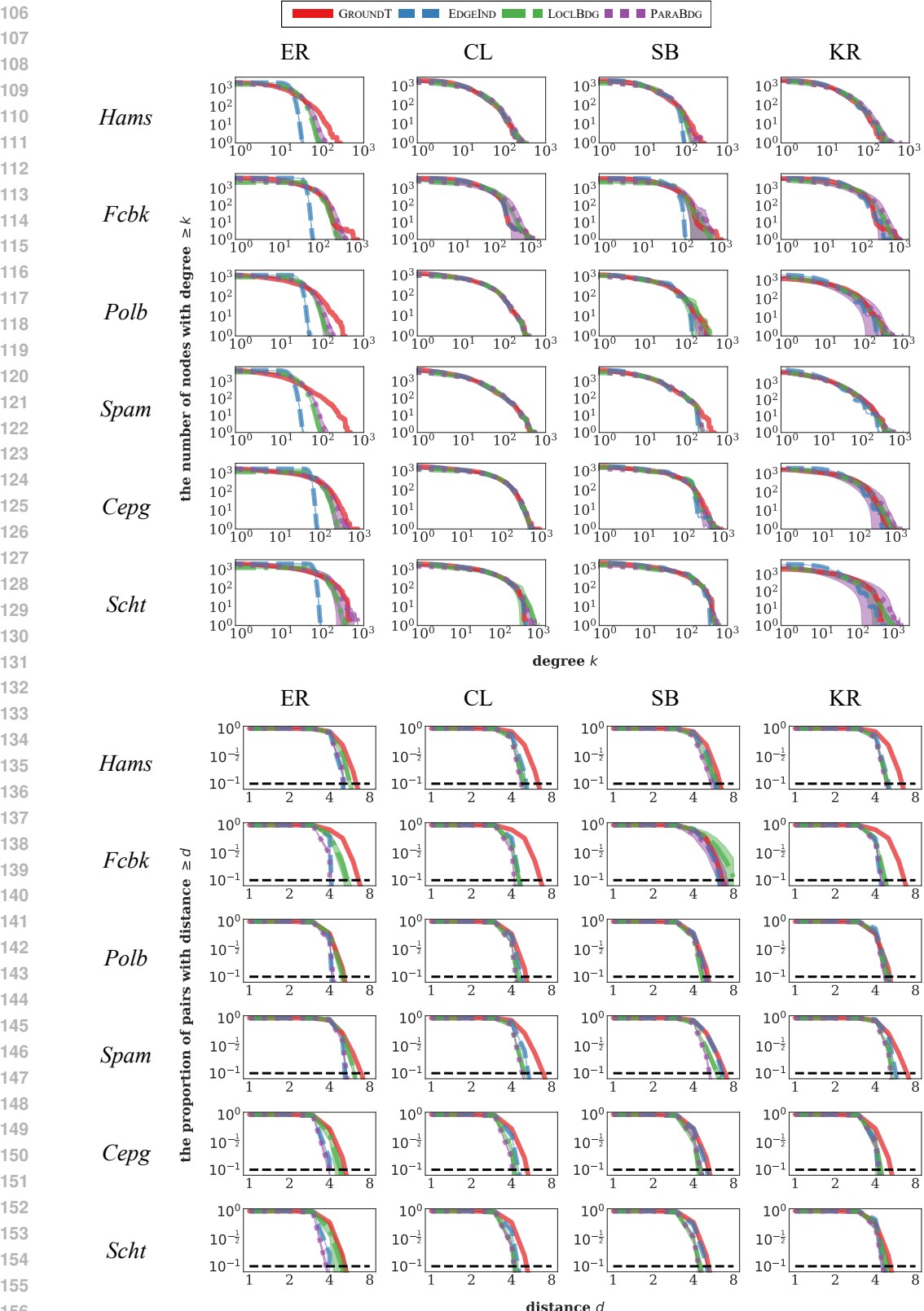

Figure 4: The degree (top) and distance (bottom) distributions of graphs generated by different realization methods. All the plots are in a log-log scale. Each shaded area represents one standard deviation.

Table 9: The numerical results regarding degrees and distances of graphs generated by different realization methods.

| dataset | | Hams | | | | Fcbk | | | | Polb | | | |
|---|---|---|---|---|---|---|---|---|---|---|---|---|---|
| metric | | $\alpha$ | $r$ | APL | $d_{\text{eff}}$ | $\alpha$ | $r$ | APL | $d_{\text{eff}}$ | $\alpha$ | $r$ | APL | $d_{\text{eff}}$ |
| model | GROUNDT | -1.432 | -0.934 | 3.589 | 5.000 | -1.180 | -0.900 | 3.693 | 5.000 | -1.069 | -0.921 | 2.738 | 4.000 |
| ER | EDGEIND | -0.058 | -0.008 | 3.004 | 4.000 | -0.046 | -0.005 | 2.606 | 3.000 | 0.009 | 0.007 | 2.507 | 3.000 |
| | LOCLBDG | -1.301 | -0.850 | 3.254 | 4.060 | -1.076 | -0.869 | 2.892 | 3.950 | -0.978 | -0.828 | 2.703 | 3.570 |
| | PARABDG | -0.958 | -0.553 | 2.996 | 4.000 | -2.338 | -0.797 | 2.262 | 3.000 | -1.136 | -0.663 | 2.416 | 3.000 |
| CL | EDGEIND | -1.414 | -0.927 | 2.938 | 4.000 | -1.185 | -0.898 | 2.608 | 3.000 | -1.055 | -0.920 | 2.585 | 3.000 |
| | LOCLBDG | -1.262 | -0.935 | 2.772 | 3.390 | -1.058 | -0.917 | 2.493 | 3.000 | -0.974 | -0.906 | 2.414 | 3.000 |
| | PARABDG | -1.282 | -0.924 | 2.713 | 3.000 | -0.980 | -0.877 | 2.331 | 3.000 | -0.968 | -0.900 | 2.373 | 3.000 |
| SB | EDGEIND | -1.211 | -0.853 | 3.309 | 4.000 | -0.600 | -0.399 | 3.507 | 5.000 | -0.967 | -0.766 | 2.717 | 4.000 |
| | LOCLBDG | -1.263 | -0.905 | 3.193 | 4.420 | -1.028 | -0.823 | 4.276 | 6.480 | -0.959 | -0.884 | 2.525 | 3.020 |
| | PARABDG | -1.209 | -0.872 | 3.000 | 4.070 | -0.409 | -0.294 | 3.429 | 5.190 | -0.954 | -0.824 | 2.595 | 3.430 |
| KR | EDGEIND | -1.359 | -0.909 | 2.856 | 3.990 | -1.185 | -0.806 | 2.566 | 3.000 | -1.332 | -0.912 | 2.848 | 3.940 |
| | LOCLBDG | -1.272 | -0.937 | 2.764 | 3.320 | -1.134 | -0.924 | 2.613 | 3.090 | -1.174 | -0.924 | 2.715 | 3.300 |
| | PARABDG | -1.301 | -0.934 | 2.742 | 3.010 | -1.104 | -0.915 | 2.499 | 3.000 | -1.164 | -0.928 | 2.661 | 3.050 |

| dataset | | Spam | | | | Cepg | | | | Scht | | | |
|---|---|---|---|---|---|---|---|---|---|---|---|---|---|
| metric | | $\alpha$ | $r$ | APL | $d_{\text{eff}}$ | $\alpha$ | $r$ | APL | $d_{\text{eff}}$ | $\alpha$ | $r$ | APL | $d_{\text{eff}}$ |
| model | GROUNDT | -1.495 | -0.947 | 3.794 | 5.000 | -0.917 | -0.907 | 2.711 | 4.000 | -0.950 | -0.860 | 2.772 | 4.000 |
| ER | EDGEIND | -0.054 | -0.008 | 3.384 | 4.000 | -0.067 | -0.009 | 2.119 | 3.000 | -0.078 | -0.011 | 2.135 | 3.000 |
| | LOCLBDG | -1.551 | -0.856 | 3.601 | 4.840 | -0.843 | -0.821 | 2.482 | 3.210 | -0.848 | -0.825 | 2.532 | 3.340 |
| | PARABDG | -1.069 | -0.541 | 3.312 | 4.000 | -1.858 | -0.765 | 2.033 | 2.490 | -2.274 | -0.800 | 1.981 | 2.000 |
| CL | EDGEIND | -1.477 | -0.943 | 3.119 | 4.000 | -0.918 | -0.897 | 2.415 | 3.000 | -0.964 | -0.905 | 2.430 | 3.000 |
| | LOCLBDG | -1.364 | -0.944 | 2.850 | 3.440 | -0.789 | -0.866 | 2.195 | 3.000 | -0.802 | -0.875 | 2.215 | 3.000 |
| | PARABDG | -1.389 | -0.940 | 2.811 | 3.000 | -0.715 | -0.809 | 2.096 | 3.000 | -0.779 | -0.825 | 2.201 | 3.000 |
| SB | EDGEIND | -1.448 | -0.893 | 3.729 | 5.000 | -0.715 | -0.644 | 2.650 | 4.000 | -0.790 | -0.749 | 2.661 | 4.000 |
| | LOCLBDG | -1.441 | -0.938 | 3.274 | 4.550 | -0.718 | -0.803 | 2.318 | 3.030 | -0.713 | -0.814 | 2.289 | 3.000 |
| | PARABDG | -1.445 | -0.931 | 3.021 | 4.000 | -0.713 | -0.661 | 2.397 | 3.000 | -0.756 | -0.793 | 2.307 | 3.000 |
| KR | EDGEIND | -1.602 | -0.929 | 3.466 | 4.000 | -0.976 | -0.807 | 2.303 | 3.000 | -1.338 | -0.882 | 2.747 | 3.000 |
| | LOCLBDG | -1.457 | -0.951 | 3.177 | 4.000 | -1.010 | -0.909 | 2.343 | 3.000 | -1.100 | -0.912 | 2.649 | 3.300 |
| | PARABDG | -1.498 | -0.953 | 3.126 | 4.000 | -0.978 | -0.904 | 2.313 | 3.000 | -1.023 | -0.891 | 2.522 | 3.000 |

Table 10: The time (in seconds) for graph generation with different realization methods.

| dataset | | Hams | Fcbk | Polb | Spam | Cepg | Scht |
|---|---|---|---|---|---|---|---|
| ER | EDGEIND | <0.05 | <0.05 | <0.05 | <0.05 | <0.05 | <0.05 |
| | LOCLBDG | 3.2 | 7.9 | 2.2 | 7.7 | 3.7 | 4.9 |
| | PARABDG | <0.05 | <0.05 | <0.05 | 0.1 | <0.05 | <0.05 |
| | PARABDG-S | 0.2 | 0.1 | <0.05 | 0.8 | <0.05 | <0.05 |
| CL | EDGEIND | <0.05 | <0.05 | <0.05 | <0.05 | <0.05 | <0.05 |
| | LOCLBDG | 4.0 | 48.2 | 2.4 | 9.3 | 6.3 | 11.5 |
| | PARABDG | 0.3 | 1.1 | 0.2 | 0.7 | 0.3 | 1.5 |
| | PARABDG-S | 3.0 | 9.4 | 1.8 | 6.8 | 2.6 | 13.9 |
| SB | EDGEIND | 0.1 | 0.1 | 0.1 | 0.1 | 0.1 | 0.1 |
| | LOCLBDG | 4.0 | 177.6 | 4.0 | 8.9 | 10.3 | 10.6 |
| | PARABDG | 0.3 | 6.2 | 0.9 | 0.7 | 1.0 | 0.7 |
| | PARABDG-S | 3.1 | 33.7 | 8.6 | 7.2 | 9.8 | 6.6 |
| KR | EDGEIND | 0.1 | 0.1 | <0.05 | 0.1 | <0.05 | 0.1 |
| | LOCLBDG | 4.7 | 49.0 | 16.6 | 28.5 | 81.0 | 200.1 |
| | PARABDG | 0.3 | 1.7 | 0.5 | 1.6 | 0.9 | 6.8 |
| | PARABDG-S | 3.2 | 12.6 | 5.2 | 14.2 | 10.5 | 31.3 |

We upscale the hamsterster (*Hams*) dataset by duplicating the whole graphs multiple times.

- The original dataset contains $|V| = 2000$ nodes.
- With 32GB RAM, all the proposed methods can run with $|V| = 128000$ ($64\times$ of the original graph).

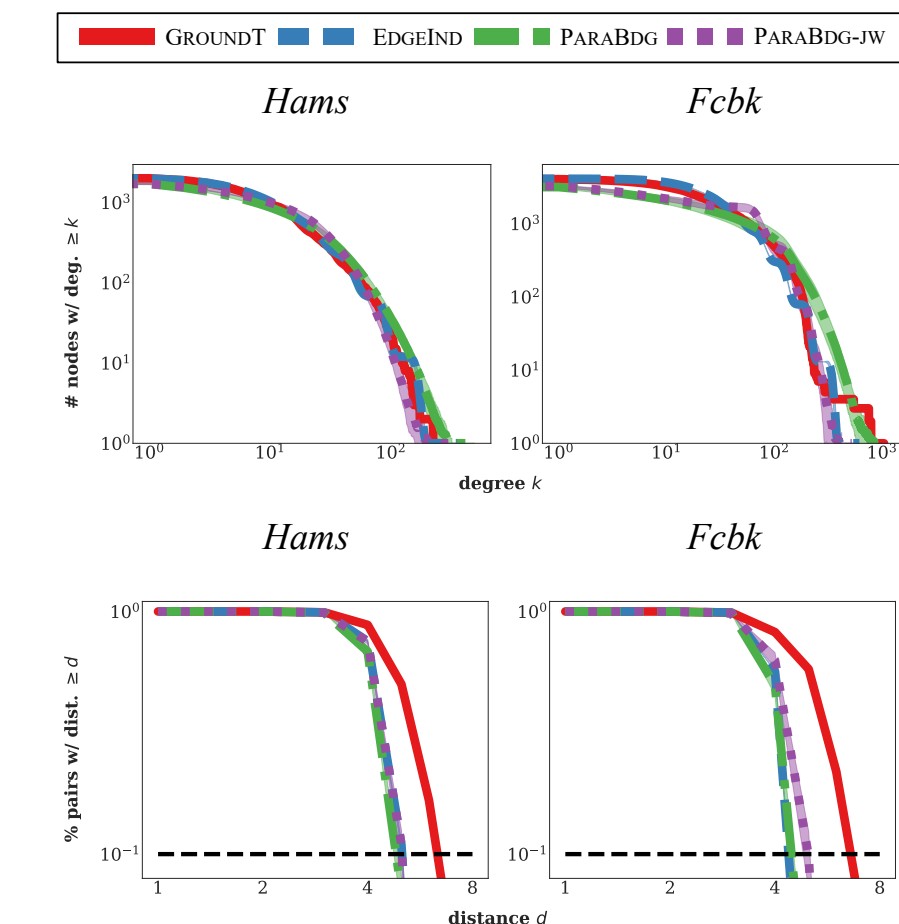

Figure 5: The degree (top) and distance (bottom) distributions of graphs generated by different realization methods. Each shaded area represents one standard deviation.

Table 11: The modularity in the graphs generated by different realization methods.

| | dataset | *Hams* | *Fcbk* | *Polb* | *Spam* | *Cepg* | *Scht* |
|---|---|---|---|---|---|---|---|
| model | GROUNDT | 0.474 | 0.777 | 0.427 | 0.462 | 0.434 | 0.253 |
| ER | EDGEIND | 0.210 | 0.120 | 0.155 | 0.205 | 0.104 | 0.099 |
| | LOCLBDG | 0.394 | 0.443 | 0.353 | 0.440 | 0.321 | 0.369 |
| | PARABDG | 0.365 | 0.517 | 0.323 | 0.394 | 0.392 | 0.430 |
| CL | EDGEIND | 0.193 | 0.107 | 0.127 | 0.180 | 0.082 | 0.078 |
| | LOCLBDG | 0.325 | 0.343 | 0.184 | 0.303 | 0.184 | 0.205 |
| | PARABDG | 0.301 | 0.332 | 0.152 | 0.271 | 0.118 | 0.262 |
| SB | EDGEIND | 0.317 | 0.756 | 0.423 | 0.370 | 0.407 | 0.208 |
| | LOCLBDG | 0.386 | 0.751 | 0.422 | 0.396 | 0.417 | 0.235 |
| | PARABDG | 0.375 | 0.741 | 0.482 | 0.432 | 0.466 | 0.263 |
| KR | EDGEIND | 0.190 | 0.114 | 0.193 | 0.254 | 0.107 | 0.142 |
| | LOCLBDG | 0.322 | 0.357 | 0.335 | 0.424 | 0.248 | 0.313 |
| | PARABDG | 0.314 | 0.367 | 0.420 | 0.411 | 0.304 | 0.385 |

See Table 17 for the detailed results.

To handle even large graphs, we further provide an alternative implementation with parallel binding (PARABDG), where we

- Save the memory usage by considering the classes of node pairs with the same probability.

Table 12: The conductance in the graphs generated by different realization methods.

| dataset | | *Hams* | *Fcbk* | *Polb* | *Spam* | *Cepg* | *Scht* |
|---|---|---|---|---|---|---|---|
| model | GROUNDT | 0.131 | 0.012 | 0.079 | 0.147 | 0.075 | 0.556 |
| ER | EDGEIND | 0.330 | 0.394 | 0.369 | 0.327 | 0.407 | 0.411 |
| | LOCLBDG | 0.235 | 0.181 | 0.271 | 0.201 | 0.311 | 0.251 |
| | PARABDG | 0.226 | 0.188 | 0.253 | 0.212 | 0.241 | 0.226 |
| CL | EDGEIND | 0.744 | 0.830 | 0.869 | 0.831 | 0.901 | 0.911 |
| | LOCLBDG | 0.444 | 0.265 | 0.816 | 0.492 | 0.813 | 0.809 |
| | PARABDG | 0.540 | 0.453 | 0.826 | 0.687 | 0.847 | 0.326 |
| SB | EDGEIND | 0.261 | 0.067 | 0.081 | 0.207 | 0.090 | 0.615 |
| | LOCLBDG | 0.222 | 0.017 | 0.080 | 0.186 | 0.083 | 0.597 |
| | PARABDG | 0.228 | 0.021 | 0.086 | 0.245 | 0.067 | 0.472 |
| KR | EDGEIND | 0.814 | 0.776 | 0.863 | 0.828 | 0.883 | 0.853 |
| | LOCLBDG | 0.411 | 0.406 | 0.420 | 0.265 | 0.474 | 0.216 |
| | PARABDG | 0.432 | 0.282 | 0.211 | 0.288 | 0.359 | 0.208 |

Table 13: The max core number in the graphs generated by different realization methods.

| dataset | | *Hams* | *Fcbk* | *Polb* | *Spam* | *Cepg* | *Scht* |
|---|---|---|---|---|---|---|---|
| model | GROUNDT | 24.0 | 115.0 | 36.0 | 35.0 | 80.0 | 100.0 |
| ER | EDGEIND | 11.0 | 32.7 | 19.5 | 10.9 | 42.9 | 46.9 |
| | LOCLBDG | 29.5 | 120.9 | 42.6 | 33.9 | 94.3 | 117.3 |
| | PARABDG | 18.7 | 70.4 | 28.1 | 20.9 | 61.7 | 79.8 |
| CL | EDGEIND | 16.9 | 43.7 | 33.5 | 25.6 | 66.7 | 79.4 |
| | LOCLBDG | 30.6 | 104.3 | 35.9 | 35.3 | 76.6 | 96.2 |
| | PARABDG | 24.4 | 105.7 | 35.3 | 27.1 | 73.1 | 96.9 |
| SB | EDGEIND | 21.4 | 71.8 | 33.9 | 37.6 | 99.8 | 96.4 |
| | LOCLBDG | 31.4 | 88.7 | 34.8 | 40.4 | 85.1 | 98.4 |
| | PARABDG | 26.3 | 121.3 | 37.4 | 38.4 | 107.8 | 109.0 |
| KR | EDGEIND | 15.5 | 32.0 | 15.9 | 13.0 | 36.2 | 25.0 |
| | LOCLBDG | 31.6 | 98.4 | 33.7 | 37.9 | 68.8 | 84.9 |
| | PARABDG | 26.3 | 107.6 | 38.5 | 37.7 | 86.1 | 109.4 |

Table 14: The average vertex betweenness (normalized w.r.t. the ground-truth value) in the graphs generated by different realization methods.

| dataset | | *Hams* | *Fcbk* | *Polb* | *Spam* | *Cepg* | *Scht* |
|---|---|---|---|---|---|---|---|
| model | GROUNDT | 1.000 | 1.000 | 1.000 | 1.000 | 1.000 | 1.000 |
| ER | EDGEIND | 0.794 | 0.610 | 0.863 | 0.876 | 0.666 | 0.647 |
| | LOCLBDG | 0.975 | 0.888 | 1.065 | 1.078 | 0.938 | 1.103 |
| | PARABDG | 0.954 | 0.945 | 1.038 | 1.089 | 0.944 | 0.954 |
| CL | EDGEIND | 0.790 | 0.605 | 0.940 | 0.835 | 0.893 | 0.817 |
| | LOCLBDG | 0.903 | 0.808 | 0.998 | 0.965 | 0.983 | 0.919 |
| | PARABDG | 0.873 | 0.755 | 0.985 | 0.946 | 0.929 | 0.792 |
| SB | EDGEIND | 0.898 | 0.961 | 0.999 | 1.024 | 0.976 | 0.929 |
| | LOCLBDG | 1.084 | 1.234 | 1.133 | 1.202 | 1.064 | 1.169 |
| | PARABDG | 1.126 | 1.446 | 1.000 | 1.081 | 0.917 | 0.945 |
| KR | EDGEIND | 0.730 | 0.582 | 0.654 | 0.626 | 0.628 | 0.503 |
| | LOCLBDG | 0.809 | 0.751 | 0.817 | 0.754 | 0.715 | 0.668 |
| | PARABDG | 0.818 | 0.757 | 0.815 | 0.762 | 0.804 | 0.715 |

– For ER, it would be all the pairs.
– For CL, each class contains node pairs with the same node degrees.

Table 15: The average edge betweenness (normalized w.r.t. the ground-truth value) in the graphs generated by different realization methods.

| model | dataset | *Hams* | *Fcbk* | *Polb* | *Spam* | *Cepg* | *Scht* |
|---|---|---|---|---|---|---|---|
| model | GROUNDT | 1.000 | 1.000 | 1.000 | 1.000 | 1.000 | 1.000 |
| ER | EDGEIND | 0.863 | 0.719 | 0.995 | 0.911 | 0.962 | 0.902 |
| | LOCLBDG | 1.062 | 1.159 | 1.122 | 1.090 | 1.186 | 1.050 |
| | PARABDG | 0.983 | 0.970 | 1.090 | 1.073 | 1.138 | 0.877 |
| CL | EDGEIND | 0.790 | 0.605 | 0.940 | 0.835 | 0.893 | 0.817 |
| | LOCLBDG | 0.903 | 0.808 | 0.998 | 0.965 | 0.983 | 0.919 |
| | PARABDG | 0.873 | 0.755 | 0.985 | 0.946 | 0.929 | 0.792 |
| SB | EDGEIND | 0.927 | 0.972 | 1.009 | 1.023 | 0.985 | 0.958 |
| | LOCLBDG | 1.055 | 2.142 | 1.358 | 1.297 | 1.274 | 1.480 |
| | PARABDG | 1.233 | 1.406 | 1.029 | 1.170 | 0.935 | 1.129 |
| KR | EDGEIND | 0.770 | 0.704 | 0.689 | 0.666 | 0.742 | 0.566 |
| | LOCLBDG | 0.886 | 1.141 | 0.952 | 0.807 | 1.089 | 1.088 |
| | PARABDG | 0.897 | 1.151 | 0.913 | 0.850 | 1.191 | 1.095 |

Table 16: The natural connectivity (normalized w.r.t. the ground-truth value) in the graphs generated by different realization methods.

| model | dataset | *Hams* | *Fcbk* | *Polb* | *Spam* | *Cepg* | *Scht* |
|---|---|---|---|---|---|---|---|
| model | GROUNDT | 1.000 | 1.000 | 1.000 | 1.000 | 1.000 | 1.000 |
| ER | EDGEIND | 0.863 | 0.719 | 0.995 | 0.911 | 0.962 | 0.902 |
| | LOCLBDG | 1.062 | 1.159 | 1.122 | 1.090 | 1.186 | 1.050 |
| | PARABDG | 0.983 | 0.970 | 1.090 | 1.073 | 1.138 | 0.877 |
| CL | EDGEIND | 0.878 | 0.633 | 1.050 | 0.884 | 0.960 | 0.895 |
| | LOCLBDG | 1.090 | 1.074 | 1.093 | 0.971 | 1.042 | 1.017 |
| | PARABDG | 0.993 | 0.900 | 1.095 | 0.930 | 1.032 | 1.003 |
| SB | EDGEIND | 0.771 | 0.523 | 0.787 | 0.912 | 0.872 | 0.890 |
| | LOCLBDG | 1.119 | 0.716 | 0.892 | 0.951 | 0.933 | 0.936 |
| | PARABDG | 0.869 | 0.864 | 1.070 | 0.926 | 1.000 | 0.924 |
| KR | EDGEIND | 0.789 | 0.475 | 0.518 | 0.427 | 0.561 | 0.316 |
| | LOCLBDG | 1.160 | 0.923 | 0.971 | 1.000 | 1.288 | 0.966 |
| | PARABDG | 0.947 | 0.807 | 1.024 | 0.684 | 0.889 | 0.837 |

Table 17: The results of the scalability experiments when upscaling the input graph (time: seconds).

| model | $|V|$ | 2k | 4k | 8k | 16k | 32k | 64k | 128k |
|---|---|---|---|---|---|---|---|---|
| ER | LOCLBDG | 3.194 | 6.505 | 16.365 | 45.648 | 143.394 | 494.536 | 1859.232 |
| | PARABDG | 0.034 | 0.058 | 0.113 | 0.232 | 0.601 | 1.705 | 5.381 |
| CL | LOCLBDG | 3.962 | 9.595 | 35.364 | 123.902 | 472.281 | 2162.315 | 8402.245 |
| | PARABDG | 0.302 | 0.495 | 1.027 | 2.114 | 4.404 | 11.184 | 31.129 |
| SB | LOCLBDG | 3.989 | 9.493 | 29.557 | 99.167 | 362.930 | 1648.392 | 8398.062 |
| | PARABDG | 0.266 | 0.489 | 0.994 | 2.132 | 5.335 | 14.861 | 45.983 |
| KR | LOCLBDG | 8.611 | 31.241 | 124.453 | 506.921 | 2097.190 | 8680.988 | 33918.420 |
| | PARABDG | 0.428 | 1.209 | 4.277 | 20.339 | 113.452 | 705.571 | 4351.573 |

– For SB, each class contains node pairs from the same blocks.

– For KR, each class contains node pairs with the same binary node labels up to permutation.

• Directly save the generated edges on the hard disk instead of in the RAM.

Table 18: The results of the scalability experiments when upscaling the input graph (time: seconds) using parallel binding (PARABDG) with additional optimization for large graphs.

| model | $|V|$ | 1m | 2m | 4m | 8m | 16m | 32m | 64m |
|-------|-------|-----|-----|-----|-----|-----|-----|-----|
| ER | PARABDG | 5.942 | 12.449 | 28.174 | 60.975 | 121.889 | 262.736 | 490.985 |
| CL | PARABDG | 102.150 | 220.177 | 423.836 | 815.883 | 1685.561 | 3135.217 | 6179.357 |
| SB | PARABDG | 106.026 | 213.722 | 428.980 | 869.002 | 1798.333 | 3829.563 | 8638.938 |
| KR | PARABDG | 105.062 | 219.351 | 439.110 | 875.381 | 1751.339 | 3504.719 | 7014.911 |

Table 19: Additional empirical evaluation on other models.

| dataset | | *Hams* | | | | *Fcbk* | | | | *Polb* | | |
|---------|-----|-----|-----|-----|-----|-----|-----|-----|-----|-----|-----|-----|
| metric | $\triangle$ | GCC | ALCC | overlap | $\triangle$ | GCC | ALCC | overlap | $\triangle$ | GCC | ALCC | overlap |
| GROUNDT | 1.000 | 0.229 | 0.540 | N/A | 1.000 | 0.519 | 0.606 | N/A | 1.000 | 0.226 | 0.320 | N/A |
| EDGEIND-CL | 0.299 | 0.067 | 0.058 | 0.059 | 0.124 | 0.064 | 0.063 | 0.063 | 0.792 | 0.183 | 0.173 | 0.182 |
| LOCLBDG-CL | 0.992 | 0.165 | 0.255 | 0.058 | 1.026 | 0.255 | 0.305 | 0.063 | 1.002 | 0.214 | 0.341 | 0.181 |
| PARABDG-CL | 1.000 | 0.185 | 0.471 | 0.059 | 1.006 | 0.336 | 0.626 | 0.062 | 1.010 | 0.221 | 0.468 | 0.181 |
| PA | 0.198 | 0.049 | 0.049 | 0.047 | 0.120 | 0.061 | 0.061 | 0.062 | 0.324 | 0.100 | 0.101 | 0.097 |
| RGG ($d=1$) | 1.252 | 0.751 | 0.751 | 0.008 | 0.607 | 0.751 | 0.752 | 0.011 | 1.127 | 0.751 | 0.753 | 0.022 |
| RGG ($d=2$) | 1.011 | 0.595 | 0.604 | 0.003 | 0.492 | 0.596 | 0.607 | 0.033 | 0.933 | 0.601 | 0.615 | 0.029 |
| RGG ($d=3$) | 0.856 | 0.491 | 0.513 | 0.003 | 0.421 | 0.494 | 0.518 | 0.033 | 0.807 | 0.503 | 0.534 | 0.029 |
| BTER | 0.991 | 0.290 | 0.558 | 0.538 | 0.880 | 0.525 | 0.605 | 0.680 | 1.028 | 0.342 | 0.375 | 0.501 |
| TCL | 0.280 | 0.075 | 0.126 | 0.223 | 0.223 | 0.117 | 0.094 | 0.192 | 0.490 | 0.138 | 0.160 | 0.411 |
| LFR ($\mu=0.0$) | 1.140 | 0.262 | 0.546 | 0.435 | N/A | N/A | N/A | N/A | 1.114 | 0.252 | 0.414 | 0.336 |
| LFR ($\mu=0.5$) | 0.296 | 0.068 | 0.081 | 0.175 | 0.161 | 0.084 | 0.120 | 0.170 | 0.571 | 0.145 | 0.170 | 0.170 |
| LFR ($\mu=1.0$) | 0.197 | 0.045 | 0.047 | 0.070 | 0.105 | 0.055 | 0.059 | 0.067 | 0.019 | 0.005 | 0.040 | 0.281 |

| dataset | | *Spam* | | | | *Cepg* | | | | *Scht* | | |
|---------|-----|-----|-----|-----|-----|-----|-----|-----|-----|-----|-----|-----|
| metric | $\triangle$ | GCC | ALCC | overlap | $\triangle$ | GCC | ALCC | overlap | $\triangle$ | GCC | ALCC | overlap |
| GROUNDT | 1.000 | 0.145 | 0.286 | N/A | 1.000 | 0.321 | 0.447 | N/A | 1.000 | 0.377 | 0.350 | N/A |
| EDGEIND-CL | 0.496 | 0.072 | 0.060 | 0.067 | 0.683 | 0.230 | 0.223 | 0.232 | 0.644 | 0.245 | 0.234 | 0.243 |
| LOCLBDG-CL | 1.028 | 0.124 | 0.260 | 0.067 | 0.996 | 0.293 | 0.430 | 0.231 | 1.036 | 0.318 | 0.469 | 0.241 |
| PARABDG-CL | 1.007 | 0.131 | 0.436 | 0.067 | 0.999 | 0.310 | 0.578 | 0.231 | 1.135 | 0.294 | 0.610 | 0.237 |
| PA | 0.112 | 0.027 | 0.026 | 0.025 | 0.288 | 0.130 | 0.130 | 0.129 | 0.226 | 0.121 | 0.123 | 0.116 |
| RGG ($d=1$) | 1.144 | 0.750 | 0.750 | 0.003 | 0.834 | 0.752 | 0.754 | 0.033 | 0.678 | 0.752 | 0.754 | 0.029 |
| RGG ($d=2$) | 0.899 | 0.592 | 0.597 | 0.003 | 0.704 | 0.604 | 0.622 | 0.033 | 0.567 | 0.603 | 0.620 | 0.029 |
| RGG ($d=3$) | 0.772 | 0.485 | 0.501 | 0.003 | 0.611 | 0.509 | 0.544 | 0.033 | 0.492 | 0.507 | 0.541 | 0.029 |
| BTER | 1.003 | 0.194 | 0.325 | 0.402 | 0.991 | 0.484 | 0.504 | 0.631 | 0.658 | 0.397 | 0.383 | 0.544 |
| TCL | 0.201 | 0.044 | 0.087 | 0.223 | 0.356 | 0.166 | 0.165 | 0.362 | 0.218 | 0.130 | 0.146 | 0.312 |
| LFR ($\mu=0.0$) | 1.283 | 0.187 | 0.406 | 0.370 | N/A | N/A | N/A | N/A | 1.081 | 0.506 | 0.850 | 0.977 |
| LFR ($\mu=0.5$) | 0.426 | 0.062 | 0.072 | 0.120 | 0.649 | 0.209 | 0.294 | 0.337 | 0.596 | 0.224 | 0.291 | 0.332 |
| LFR ($\mu=1.0$) | 0.332 | 0.048 | 0.042 | 0.081 | 0.516 | 0.166 | 0.217 | 0.303 | 0.476 | 0.179 | 0.212 | 0.292 |

By doing so, we are able to scale to even large graphs. See Table 18 for the detailed results. Notably, parallel binding (PARABDG) is easily parallelizable. We can distribute the generation to multiple machines and finally merge the generated edges, which allows us to handle even larger graphs.

### E.5 JOINT OPTIMIZATION

As shown in Section 6.5, in some "difficult" cases where PARABDG well preserves the number of triangles but not the number of wedges, with joint optimization, PARABDG-JW does better, well preserving both the number of triangles and the number of wedges. In Figure 5, for both *Hams* and *Fcbk*, we compare the degree and distance distributions in the ground-truth graph and in the graphs generated by EDGEIND, PARABDG, and PARABDG-JW. With joint optimization, both degree and distance distributions do not change much (compare PARABDG and PARABDG-JW in Figure 5).

### E.6 ON HIGH-OVERLAP EIGMS, OTHER EDGE-DEPENDENT RGMS, AND MORE

As discussed in Section 3.1, there exist methods that shift edge probabilities by various mechanisms, while they are still essentially EIGMs. Hence, by Theorem 3.3, they inevitably trade-off between variability and the ability to generate high-clustering graphs. Such methods include Binning Chung Lu (BCL) proposed by Mussmann et al. (2015) that uses accept-reject and Block Two-level Erdos-Renyi (BTER) proposed by Kolda et al. (2014b) that uses a mixture of different EIGMs (specifically,

Table 20: The $\rho$ values (i.e., the probability of taking the triangle-forming step) used by TCL for each dataset.

| dataset | Hams | Fcbk | Polb | Spam | Cepg | Scht |
|---------|-------|-------|-------|-------|-------|-------|
| TCL $\rho$ | 0.877 | 0.986 | 0.035 | 0.652 | 0.263 | 0.411 |

Erdos-Renyi and Chung-Lu). Also, as discussed in Section 3.2, there are also existing methods that use additional mechanisms to improve upon existing EIGMs. For example, Pfeiffer et al. (2012) proposed Transitive Chung-Lu (TCL) that uses an additional mechanism to directly insert triangles on top of the original edge-independent Chung-Lu.

**Differences.** In this work, we aim to improve upon EIGMs by further exploring models without assuming edge independency. The key point is to preserve individual edge probabilities and thus have high tractability, but the existing methods usually use mixed models and thus change the underlying edge probabilities. The consequence is that they either have less tractability or less variability (i.e., high overlap; see Theorems 3.2 and 4.7).

• TCL uses an additional mechanism to directly form triangles and is thus less tractable;

• BTER forms many small dense communities and has very high overlap.

As shown in Property 4.7, EPGMs have the same overlap as the corresponding EIGM, i.e., the variability is perfectly maintained even though we introduce edge dependency.

Below, we compare the performance of (1) the original edge-independent Chung-Lu, (2) Chung-Lu with local binding, (3) Chung-Lu with parallel binding, (4) TCL, and (5) BTER.

**Evaluation.** In addition to the clustering-related metrics (the number of triangles, global clustering coefficient, and the average local clustering coefficient) we used in our main experiments, we further compare the "overlap" (see Definition 3.1) of the generated graphs. Roughly, the overlap of a random graph model is the expected proportion of overlapping edges between two randomly generated graphs (i.e., the edges that exist in both randomly generated graphs). Higher overlap values imply lower variability; when overlap approaches 1, the generated graphs are almost identical.

**Implementation.**

• For TCL, we use online Python code;[16]

• For BTER, we use the official MATLAB implementation.[17]

**Results.** In Table 19, we show the detailed results. Overall, we have the following observations.

• For some datasets (e.g., *facebook*), TCL almost-always (i.e., $\rho \approx 1$) uses the mechanism that directly forms triangles. Even so, TCL often fails to well preserve the clustering-related metrics in real-world graphs.

  – TCL mixes two types of steps: (1) original Chung-Lu with probability $(1-\rho)$ and (2) a triangle-forming step with probability $\rho$.

  – See Table 20 for the $\rho$ values used by TCL for each dataset.

• As expected, although BTER generates graphs with high clustering as intended, it has very high overlap, which implies that it well reproduces high-clustering graphs by largely duplicating the input graphs.

• Our methods with binding schemes have the same overlap as the corresponding EIGM, while well preserving clustering-related metrics in real-world graphs.

**Other edge-dependent RMGs.** For the experiments on other edge-dependent RMGs in Section 6.6, we provide more details here.

• For random geometric graphs (RGG), we tried dimensions $d \in \{1, 2, 3\}$, while setting the number of nodes as that in the input graph, and setting the diameter to fit the number of edges in the input graph. Note that the clustering in the generated graph is only determined by the dimension, and smaller dimensions give higher clustering.

---

[16]`https://github.com/pdsteele/socialNetworksProject/blob/master/proj-TransChungLu.py`

[17]`https://www.mathsci.ai/feastpack`

| average $g(v)$ | $\triangle$ | GCC | ALCC |
|---|---|---|---|
| 0 (EIGM) | 179.21 | 0.010 | 0.010 |
| 0.001 | 1957.88 | 0.100 | 0.119 |
| 0.002 | 3721.49 | 0.177 | 0.249 |
| 0.003 | 5499.17 | 0.240 | 0.379 |
| 0.004 | 7323.14 | 0.296 | 0.489 |
| 0.005 | 9489.65 | 0.344 | 0.568 |
| 0.006 | 10796.54 | 0.386 | 0.635 |
| 0.007 | 12742.98 | 0.422 | 0.681 |
| 0.008 | 14342.90 | 0.464 | 0.723 |
| 0.009 | 16122.18 | 0.491 | 0.749 |
| 0.01 | 18116.62 | 0.514 | 0.772 |

(a) ER + PARABDG

| average $g(v)$ | $\triangle$ | GCC | ALCC |
|---|---|---|---|
| 0 (EIGM) | 179.21 | 0.010 | 0.010 |
| 0.001 | 338.67 | 0.019 | 0.018 |
| 0.002 | 1006.9 | 0.054 | 0.047 |
| 0.003 | 1864.64 | 0.092 | 0.088 |
| 0.004 | 2567.84 | 0.121 | 0.125 |
| 0.005 | 3178.68 | 0.143 | 0.151 |
| 0.006 | 3797.42 | 0.165 | 0.171 |
| 0.007 | 4301.58 | 0.183 | 0.187 |
| 0.008 | 5080.94 | 0.202 | 0.200 |
| 0.009 | 5542.13 | 0.218 | 0.210 |
| 0.01 | 6441.86 | 0.236 | 0.222 |

(b) ER + LOCLBDG

Table 21: The clustering metrics of generated graphs without fitting specific graphs using ER as the underlying edge-probability model.

- For preferential attachment (PA), we tried the extended Barabási-Albert model.[18] We set the number of nodes as that in the input graph, and set the parameter $m$ to fit the number of edges in the input graph. We tried $p, q \in \{0, 0.1, 0.2, 0.3\}$. We report the variant that gives the highest clustering.

- For the Lancichinetti-Fortunato-Radicchi (LFR) model, we set the degrees as the ground-truth degrees, set the community sizes as the sizes of the communities detected using the Louvain algorithm, and tried different mixing parameters $\mu \in \{0, 0.5, 1.0\}$.

**Discussions on deep graph generative models.** Recently, deep graph generative models have become more and more popular. Typically, deep graph generative models aim to fit a population of small graphs, while this work focuses on fitting random graph models to individual input graphs. We empirically tested three deep graph generative models: CELL (Rendsburg et al., 2020), Graph-VAE (Simonovsky & Komodakis, 2018), and GrpahRNN (You et al., 2018).

We summarize our empirical observations as follows:

- CELL often fails to generate high clustering, and also generates high overlap (i.e., low variability). CELL is essentially an EIGM. See also the discussions by Chanpuriya et al. (2021).

- GraphVAE learns to duplicate the training graph (i.e., 100% overlap). This is likely because GraphVAE was designed to learn from a population of graphs instead of a single graph, as discussed above.

- GraphRNN often generates graphs with far more edges but still low clustering. This is likely because GraphRNN was designed mainly for relatively small graphs and cannot fit well to individual large graphs.

As discussed by Chanpuriya et al. (2021), several deep graph generative models also output edge probabilities (e.g., CELL), and this work provides a new perspective to potentially enhance them with edge dependency.

### E.7    ON GRAPH GENERATION WITHOUT FITTING SPECIFIC GRAPHS

Instead of fitting specific graphs as done in our main experiments, one can also use the proposed models to generate graphs "from scratch" without specific graphs as references by freely choosing the parameters.

First, one needs to choose the underlying edge probabilities. Typically, one can use an underlying edge-probability model and choose it according to the required properties. For example, if one wants to generate graphs with power-law degree distributions, Chung-Lu with a prescribed power-law degree sequence can be used. Or, if one wants to generate a graph with community structures, the stochastic block model can be used.

---

[18]See, e.g., `https://networkx.org/documentation/stable/reference/generated/networkx.generators.random_graphs.extended_barabasi_albert_graph.html`.

| $\alpha$ | average $g(v)$ | $\triangle$ | GCC | ALCC |
|---|---|---|---|---|
| | 0 (EIGM) | 13668.59 | 0.167 | 0.337 |
| | 0.01 | 12506.14 | 0.153 | 0.493 |
| | 0.02 | 13160.15 | 0.156 | 0.536 |
| | 0.03 | 13844.84 | 0.161 | 0.559 |
| | 0.04 | 15182.06 | 0.172 | 0.568 |
| -0.3 | 0.05 | 15610.28 | 0.168 | 0.584 |
| | 0.06 | 17647.33 | 0.179 | 0.588 |
| | 0.07 | 16757.68 | 0.172 | 0.588 |
| | 0.08 | 16119.25 | 0.173 | 0.593 |
| | 0.09 | 15417.53 | 0.160 | 0.594 |
| | 0.1 | 18102.03 | 0.176 | 0.605 |
| | 0 (EIGM) | 13668.59 | 0.167 | 0.337 |
| | 0.01 | 13051.15 | 0.159 | 0.539 |
| | 0.02 | 14274.04 | 0.171 | 0.585 |
| | 0.03 | 15724.32 | 0.181 | 0.602 |
| | 0.04 | 16188.49 | 0.182 | 0.614 |
| 0 | 0.05 | 19404.04 | 0.200 | 0.622 |
| | 0.06 | 20993.48 | 0.209 | 0.634 |
| | 0.07 | 19845.02 | 0.198 | 0.639 |
| | 0.08 | 23823.32 | 0.215 | 0.634 |
| | 0.09 | 30700.56 | 0.232 | 0.644 |
| | 0.1 | 26477.88 | 0.215 | 0.646 |
| | 0 (EIGM) | 13668.59 | 0.167 | 0.337 |
| | 0.01 | 14245.14 | 0.173 | 0.598 |
| | 0.02 | 17062.43 | 0.195 | 0.643 |
| | 0.03 | 19329.61 | 0.215 | 0.660 |
| | 0.04 | 22821.36 | 0.232 | 0.673 |
| 0.3 | 0.05 | 23128.39 | 0.238 | 0.684 |
| | 0.06 | 28266.25 | 0.250 | 0.697 |
| | 0.07 | 30571.88 | 0.265 | 0.703 |
| | 0.08 | 27047.89 | 0.250 | 0.717 |
| | 0.09 | 38293.91 | 0.286 | 0.728 |
| | 0.1 | 34335.56 | 0.278 | 0.731 |

(a) CL + PARABDG

| $\alpha$ | average $g(v)$ | $\triangle$ | GCC | ALCC |
|---|---|---|---|---|
| | 0 (EIGM) | 13668.59 | 0.167 | 0.337 |
| | 0.01 | 11962.88 | 0.148 | 0.426 |
| | 0.02 | 12417.21 | 0.149 | 0.462 |
| | 0.03 | 12688.25 | 0.153 | 0.475 |
| | 0.04 | 12847.39 | 0.151 | 0.486 |
| -0.3 | 0.05 | 13543.07 | 0.159 | 0.495 |
| | 0.06 | 14457.40 | 0.163 | 0.504 |
| | 0.07 | 13856.21 | 0.155 | 0.511 |
| | 0.08 | 14942.73 | 0.156 | 0.530 |
| | 0.09 | 15551.34 | 0.163 | 0.524 |
| | 0.1 | 14264.12 | 0.154 | 0.532 |
| | 0 (EIGM) | 13668.59 | 0.167 | 0.337 |
| | 0.01 | 12155.44 | 0.152 | 0.433 |
| | 0.02 | 12651.05 | 0.154 | 0.463 |
| | 0.03 | 13348.38 | 0.160 | 0.480 |
| | 0.04 | 13249.86 | 0.157 | 0.495 |
| 0 | 0.05 | 14450.43 | 0.167 | 0.503 |
| | 0.06 | 15668.08 | 0.171 | 0.518 |
| | 0.07 | 14949.55 | 0.169 | 0.519 |
| | 0.08 | 14733.55 | 0.164 | 0.525 |
| | 0.09 | 19401.58 | 0.182 | 0.528 |
| | 0.1 | 18072.88 | 0.182 | 0.529 |
| | 0 (EIGM) | 13668.59 | 0.167 | 0.337 |
| | 0.01 | 12544.92 | 0.154 | 0.433 |
| | 0.02 | 13383.98 | 0.160 | 0.461 |
| | 0.03 | 13901.56 | 0.166 | 0.476 |
| | 0.04 | 15005.39 | 0.175 | 0.493 |
| 0.3 | 0.05 | 16448.43 | 0.181 | 0.506 |
| | 0.06 | 16623.27 | 0.182 | 0.503 |
| | 0.07 | 18159.03 | 0.186 | 0.523 |
| | 0.08 | 16835.26 | 0.185 | 0.522 |
| | 0.09 | 18177.49 | 0.188 | 0.538 |
| | 0.1 | 18459.21 | 0.195 | 0.546 |

(b) CL + LOCLBDG

Table 22: The clustering metrics of generated graphs without fitting specific graphs using CL as the underlying edge-probability model.

Below, we shall discuss the graph statistics of random graphs generated by EPGMs using binding with varying parameters. Let us first provide the parameter ranges.

For the Erdős-Rényi (ER) model:

- The number $n$ of nodes is fixed as 1024.
- The edge probability $p(u, v)$ is 0.01, the same for all the node pairs. The value 0.01 is chosen in the typical range of real-world graphs (Melancon, 2006).
- The node-sampling probability $g(v)$ is the same for all the nodes (as discussed in Section 5.4), with varying values.
- The number $R$ of rounds is 100000, as in our main experiments.

For the Chung-Lu (CL) model:

- The number $n$ of nodes is fixed as 1024.
- The degree sequence $d_v$'s are generated as a power-law sequence with power-law exponent 2, so that the average edge probability $p(u, v)$ is around 0.01. The exponent 2 is chosen in the typical range of real-world graphs (Chakrabarti & Faloutsos, 2006).
- The node-sampling probability $g(v)$ is the same for nodes with the same degree (as discussed in Section 5.4), with varying mean values and varying correlation with degrees. Specifically, for each node $v$, we set the node-sampling probability $g(v)$ proportional to $d(v)^\alpha$ with different $\alpha$ values (-0.3, 0, and 0.3), where $d(v)$ is the degree of node $v$. The $\alpha$ values are chosen so that no node has a node-sampling probability exceeding 1. The node-sampling probabilities are positively (resp., negatively) correlated with node degrees with a positive (resp., negative) $\alpha$ value. When $\alpha = 0$, the node-sampling probability is the same for all the nodes.

| $\alpha$ | average $g(v)$ | $\triangle$ | GCC | ALCC | $\alpha$ | average $g(v)$ | $\triangle$ | GCC | ALCC |
|---|---|---|---|---|---|---|---|---|---|
| | 0 (EIGM) | 297.15 | 0.167 | 0.337 | | 0 (EIGM) | 297.15 | 0.015 | 0.014 |
| | 0.01 | 1070.80 | 0.153 | 0.493 | | 0.01 | 3139.06 | 0.133 | 0.147 |
| | 0.02 | 1857.45 | 0.156 | 0.536 | | 0.02 | 6016.54 | 0.215 | 0.208 |
| | 0.03 | 2587.32 | 0.161 | 0.559 | | 0.03 | 8872.29 | 0.272 | 0.239 |
| | 0.04 | 3393.55 | 0.172 | 0.568 | | 0.04 | 11101.77 | 0.313 | 0.256 |
| -0.5 | 0.05 | 4140.39 | 0.168 | 0.584 | -0.5 | 0.05 | 13093.51 | 0.331 | 0.261 |
| | 0.06 | 4980.47 | 0.179 | 0.588 | | 0.06 | 19008.57 | 0.368 | 0.274 |
| | 0.07 | 5662.74 | 0.172 | 0.588 | | 0.07 | 18992.58 | 0.378 | 0.270 |
| | 0.08 | 6440.77 | 0.173 | 0.593 | | 0.08 | 23138.66 | 0.412 | 0.276 |
| | 0.09 | 7169.38 | 0.160 | 0.594 | | 0.09 | 24280.39 | 0.412 | 0.271 |
| | 0.1 | 7949.71 | 0.176 | 0.605 | | 0.1 | 30652.89 | 0.432 | 0.280 |
| | 0 (EIGM) | 297.15 | 0.167 | 0.337 | | 0 (EIGM) | 297.15 | 0.015 | 0.014 |
| | 0.01 | 1460.17 | 0.159 | 0.539 | | 0.01 | 4257.22 | 0.172 | 0.170 |
| | 0.02 | 2656.48 | 0.171 | 0.585 | | 0.02 | 7764.28 | 0.265 | 0.221 |
| | 0.03 | 3821.19 | 0.181 | 0.602 | | 0.03 | 11887.79 | 0.327 | 0.247 |
| | 0.04 | 4995.17 | 0.182 | 0.614 | | 0.04 | 16886.04 | 0.379 | 0.253 |
| 0 | 0.05 | 6135.86 | 0.200 | 0.622 | 0 | 0.05 | 20868.18 | 0.405 | 0.257 |
| | 0.06 | 7271.69 | 0.209 | 0.634 | | 0.06 | 23889.00 | 0.436 | 0.256 |
| | 0.07 | 8458.58 | 0.198 | 0.639 | | 0.07 | 29247.06 | 0.451 | 0.264 |
| | 0.08 | 9924.68 | 0.215 | 0.634 | | 0.08 | 30123.19 | 0.450 | 0.253 |
| | 0.09 | 10945.15 | 0.232 | 0.644 | | 0.09 | 36971.23 | 0.451 | 0.254 |
| | 0.1 | 12061.47 | 0.215 | 0.646 | | 0.1 | 45597.38 | 0.468 | 0.264 |
| | 0 (EIGM) | 297.15 | 0.167 | 0.337 | | 0 (EIGM) | 297.15 | 0.015 | 0.014 |
| | 0.01 | 1527.10 | 0.173 | 0.598 | | 0.01 | 4348.57 | 0.170 | 0.156 |
| | 0.02 | 2746.08 | 0.195 | 0.643 | | 0.02 | 8368.21 | 0.269 | 0.215 |
| | 0.03 | 3989.85 | 0.215 | 0.660 | | 0.03 | 12679.81 | 0.331 | 0.235 |
| | 0.04 | 5209.86 | 0.232 | 0.673 | | 0.04 | 17480.57 | 0.380 | 0.246 |
| 0.5 | 0.05 | 6516.78 | 0.238 | 0.684 | 0.5 | 0.05 | 19396.23 | 0.412 | 0.242 |
| | 0.06 | 7707.59 | 0.250 | 0.697 | | 0.06 | 24362.75 | 0.418 | 0.247 |
| | 0.07 | 8833.38 | 0.265 | 0.703 | | 0.07 | 30949.97 | 0.453 | 0.250 |
| | 0.08 | 10119.58 | 0.250 | 0.717 | | 0.08 | 33129.43 | 0.437 | 0.252 |
| | 0.09 | 11197.86 | 0.286 | 0.728 | | 0.09 | 36050.41 | 0.463 | 0.253 |
| | 0.1 | 12589.62 | 0.278 | 0.731 | | 0.1 | 41233.18 | 0.458 | 0.260 |

(a) SB + PARABDG  (b) SB + LOCLBDG

Table 23: The clustering metrics of generated graphs without fitting specific graphs using SB as the underlying edge-probability model.

- The number $R$ of rounds is 100000, as in our main experiments.

For the stochastic block (SB) model:

- The number $n$ of nodes is fixed as 1024.
- The number of communities (i.e., blocks) is fixed as 10.
- The community sizes are generated as a power-law with power-law exponent 1.5. The exponent 1.5 is chosen in the typical range of real-world graphs (Fortunato, 2010).
- The intra-community edge probability and inter-community edge probability are the same for different communities, and are chosen so that the average edge probability $p(u, v)$ is around 0.01.
- The node-sampling probability $g(v)$ is the same for nodes with the same community (as discussed in Section 5.4), with varying mean values and varying correlation with community sizes. Specifically, for each node $v$, we set the node-sampling probability $g(v)$ proportional to $s(v)^{\alpha}$ with different $\alpha$ values (-0.5, 0, and 0.5), where $s(v)$ is the size of the community $v$ is in. The $\alpha$ values are chosen so that no node has a node-sampling probability exceeding 1. The node-sampling probabilities are positively (resp., negatively) correlated with community sizes with a positive (resp., negative) $\alpha$ value. When $\alpha = 0$, the node-sampling probability is the same for all the nodes.
- The number $R$ of rounds is 100000, as in our main experiments.

For the stochastic Kronecker (KR) model:

- The number $n$ of nodes is fixed as 1024. Specifically, the seed matrix is two-by-two, and we take the order-10 Kronecker power of the seed matrix.

| $\alpha$ | average $g(v)$ | $\triangle$ | GCC | ALCC |
|---|---|---|---|---|
| -1 | 0 (EIGM) | 1044.88 | 0.031 | 0.033 |
| | 0.01 | 6586.23 | 0.157 | 0.327 |
| | 0.02 | 11809.86 | 0.240 | 0.422 |
| | 0.03 | 17238.21 | 0.305 | 0.471 |
| | 0.04 | 22369.54 | 0.334 | 0.508 |
| | 0.05 | 27207.33 | 0.361 | 0.530 |
| | 0.06 | 37717.33 | 0.409 | 0.556 |
| | 0.07 | 38131.85 | 0.418 | 0.559 |
| | 0.08 | 46682.81 | 0.432 | 0.580 |
| | 0.09 | 52381.99 | 0.435 | 0.595 |
| | 0.1 | 56996.55 | 0.439 | 0.604 |
| 0 | 0 (EIGM) | 1044.88 | 0.031 | 0.033 |
| | 0.01 | 8860.73 | 0.198 | 0.402 |
| | 0.02 | 16677.24 | 0.307 | 0.499 |
| | 0.03 | 23927.22 | 0.373 | 0.549 |
| | 0.04 | 33434.06 | 0.430 | 0.582 |
| | 0.05 | 33908.84 | 0.436 | 0.600 |
| | 0.06 | 51113.67 | 0.488 | 0.624 |
| | 0.07 | 56890.50 | 0.503 | 0.631 |
| | 0.08 | 57262.88 | 0.492 | 0.637 |
| | 0.09 | 73944.66 | 0.517 | 0.656 |
| | 0.1 | 71084.79 | 0.486 | 0.642 |
| 1 | 0 (EIGM) | 1044.88 | 0.031 | 0.033 |
| | 0.01 | 11202.40 | 0.242 | 0.472 |
| | 0.02 | 21129.53 | 0.360 | 0.572 |
| | 0.03 | 30490.92 | 0.443 | 0.619 |
| | 0.04 | 42543.89 | 0.501 | 0.648 |
| | 0.05 | 48769.38 | 0.535 | 0.664 |
| | 0.06 | 57292.62 | 0.562 | 0.665 |
| | 0.07 | 62270.76 | 0.558 | 0.682 |
| | 0.08 | 68569.86 | 0.522 | 0.689 |
| | 0.09 | 99628.36 | 0.565 | 0.708 |
| | 0.1 | 109134.19 | 0.547 | 0.702 |

(a) KR + PARABDG

| $\alpha$ | average $g(v)$ | $\triangle$ | GCC | ALCC |
|---|---|---|---|---|
| -1 | 0 (EIGM) | 1044.88 | 0.031 | 0.033 |
| | 0.01 | 3086.48 | 0.081 | 0.147 |
| | 0.02 | 5166.20 | 0.122 | 0.188 |
| | 0.03 | 7636.03 | 0.158 | 0.211 |
| | 0.04 | 9707.45 | 0.180 | 0.227 |
| | 0.05 | 10435.17 | 0.189 | 0.232 |
| | 0.06 | 13325.55 | 0.207 | 0.243 |
| | 0.07 | 16207.68 | 0.225 | 0.250 |
| | 0.08 | 18169.74 | 0.236 | 0.261 |
| | 0.09 | 21601.60 | 0.234 | 0.267 |
| | 0.1 | 23594.77 | 0.241 | 0.270 |
| 0 | 0 (EIGM) | 1044.88 | 0.031 | 0.033 |
| | 0.01 | 3726.94 | 0.094 | 0.154 |
| | 0.02 | 6239.78 | 0.141 | 0.187 |
| | 0.03 | 8729.61 | 0.176 | 0.209 |
| | 0.04 | 11659.48 | 0.209 | 0.218 |
| | 0.05 | 15550.88 | 0.238 | 0.230 |
| | 0.06 | 17557.07 | 0.241 | 0.232 |
| | 0.07 | 22365.64 | 0.271 | 0.243 |
| | 0.08 | 20931.69 | 0.266 | 0.237 |
| | 0.09 | 20406.49 | 0.240 | 0.237 |
| | 0.1 | 26693.30 | 0.254 | 0.249 |
| 1 | 0 (EIGM) | 1044.88 | 0.031 | 0.033 |
| | 0.01 | 4118.72 | 0.103 | 0.152 |
| | 0.02 | 7056.13 | 0.155 | 0.181 |
| | 0.03 | 10525.82 | 0.196 | 0.201 |
| | 0.04 | 12531.47 | 0.217 | 0.198 |
| | 0.05 | 16213.98 | 0.245 | 0.207 |
| | 0.06 | 22050.84 | 0.276 | 0.221 |
| | 0.07 | 19337.29 | 0.256 | 0.212 |
| | 0.08 | 25260.46 | 0.276 | 0.220 |
| | 0.09 | 24525.96 | 0.263 | 0.213 |
| | 0.1 | 33226.16 | 0.276 | 0.236 |

(b) KR + LOCLBDG

Table 24: The clustering metrics of generated graphs without fitting specific graphs using KR as the underlying edge-probability model.

- The seed matrix is [0.95, 0.63; 0.63, 0.32]. The values in the seed matrix are chosen so that the average edge probability $p(u, v)$ is around 0.01, and the value distribution is similar to those in the original paper of Kronecker Leskovec et al. (2010).
- The node-sampling probability $g(v)$ is the same for nodes with the same number of ones in their binary node labels (as discussed in Section 5.4), with varying mean values and varying correlation with the number of ones. Specifically, for each node $v$, we set the node-sampling probability $g(v)$ proportional to $(i(v) + 1)^\alpha$ with different $\alpha$ values (-1, 0, and 1), where $i(v)$ is the number of ones in the binary node label of $v$. The $\alpha$ values are chosen so that no node has a node-sampling probability exceeding 1. The node-sampling probabilities are positively (resp., negatively) correlated with the number of ones with a positive (resp., negative) $\alpha$ value. When $\alpha = 0$, the node-sampling probability is the same for all the nodes.

In Tables 21 to 24, we show the clustering metrics of graphs generated without fitting specific graphs as described above, with different underlying edge-probability models.

Below, let us discuss the insights we have based on the results. Overall, in line with our theoretical analysis, in most cases, when we increase node-sampling probabilities, the generated graphs have higher clustering. By varying node-sampling probabilities, one can generate graphs with different levels of clustering. Also, with the same node-sampling probabilities, PARABDG generates graphs with higher clustering than LOCLBDG.

There are also interesting observations on the correlation between node-sampling probabilities and some parameters in the underlying edge-probability models, indicated by the value of $\alpha$. For CL, with the same average node-sampling probability, when we make node-sampling probabilities positively correlated to the node degrees, the generated graphs have higher clustering. For SB, with

the same average node-sampling probability, when we make node-sampling probabilities negatively correlated to the node degrees, the generated graphs have relatively lower clustering, while positive correlation and no correlation give similar results. For KR, with the same average node-sampling probability, when we make node-sampling probabilities positively correlated to the number of ones, the generated graphs have higher clustering.

