# OpenReview forum: "Exploring Edge Probability Graph Models Beyond Edge Independency: Concepts, Analyses, and Algorithms"
_ICLR.cc/2025/Conference — Submitted to ICLR 2025_

### Official Review · Reviewer_xVy1 · 2024-10-20

**Soundness:** 3
**Presentation:** 3
**Contribution:** 3
**Rating:** 6
**Confidence:** 4

**Summary:**

This paper proposes a new configurable way to generate random graphs with dependencies between the edges, in a fine-grained way that make it possible to control parameters such as the number of triangles, clustering coefficients, etc.

The simplest and most famous random graph models, such as Erdos-Renyi, have edges appear independently of each other (in Erdos-Renyi for example the probability of each edge to appear is $p$, independently of all others). This kind of behavior makes analysis easier, but on the other hand it provably cannot capture properties of realistic graphs, such as a large number of triangles/motifs, and the tendancy of realistic graphs to be relatively clustered.

More modern random graph models, that are better at capturing real world characteristics, tend to have dependencies between the edges. One example is random geometric graphs, where points are randomly located on a high-dimensional sphere. But still, most of these models are not very flexible, they are controlled by some global parameters but it may be difficult/impossible for the designer to build a graph that matches specific desired properties.

This paper provides several variants of a new and simple method, called binding, to generate random graphs with high variability (i.e., that do not match a single specific pattern) but also being realistic (having high triangle density, high clusterability, etc). The core idea is simple, and stems from the following question: what is the random graph model with all edges appearing with probability $p$ (possibly dependently), and, say, the largest expected number of triangles? It turns out that this is the distribution which is equal to a complete graph with probability $p$, and empty graph otherwise. Binding is a similar process, picking the same "threshold" to groups of random nodes in order to create small cliques / dense graphs. The proposed model allows to work with predetermined edge probabilities, not necessarily uniform.

The authors propose a couple of different binding methods, one that is sequential and another one that is parallelizable and easier to analyze. They prove several theoretical results: for example, that it is possible to efficiently match a target expected number of triangles by a suitable optimization process. The authors also run some experiments demonstrating the verstality of this method in practice, showing ability to generate diverse and realistic degree distributions or obtain a large triangle coefficient. Another experiment shows that the running time is decently fast, especially for the parallelizable version.

**Strengths:**

- An elegant, novel and flexible method to generate random graphs. Although the generated graphs from, say, the local binding model do not yet seem fully realistic to me, this is a good step in the right direction and the basic binding primitive is natural and analyzable.

- The paper is well written in general. The mathematical statements are clear, intuitive, interesting, and believable (I have not fully checked the proofs).

- The topic of "realistic" random graph models is very important and this paper certainly advances the literature (even if by a small step) on this topic.

**Weaknesses:**

- The experimental section does not provide a lot of information beyond the theoretical part of the paper. It is relatively clear to me, without the experiments or even looking at the proofs, that this model will be able to capture a target triangle density, for example. I would have hoped for the experimental section to show other directions in which this model is realistic. (See questions below.)

**Questions:**

My questions are mostly about how "realistic" this random process is:

1. What are some types of real-world network properties you believe *cannot* be captured by the binding processes you outlined in the paper?
Can it capture realistic community structure faithfully?

2. One thing missing from the paper, as outlined in the weaknesses part, is an experimental part showing the this specific model is useful for downstream applications (and not just as proxy for estimating quantities that are practically useful in downstream applications, such as triangle density). I would imagine that one way to instantiate it is by pre-training some graph neural network on this kind of graphs and show that this pretraining helps improve the performance on some applied benchmark. Do you think it will be possible/realistic to design such an experiment? In what context do you think this random model could come up useful?

---

> ### Author Response · Authors · 2024-11-20
>
> Dear Reviewer xVy1,
>
> Thanks for your insightful comments.
>
> Please find our responses to your individual comments below.
>
> We welcome any further comments and questions during the discussion phase.
>
> ### **Updated manuscript**
>
> We have updated the manuscript accordingly. The updated parts are $\textcolor{blue}{\text{highlighted in blue}}$. When we refer to contents in the manuscript, we **refer to the updated manuscript by default**, unless specified otherwise.
>
> - See the updated manuscript PDF at https://openreview.net/pdf?id=xljPZuprBA

---

> ### Author Response · Authors · 2024-11-20
> **<Response 1> On The Value of Experiments**
>
> ## **<Response 1> On The Value of Experiments**
>
> ### **Context: Related Comments**
>
> - The experimental section does not provide a lot of information beyond the theoretical part of the paper. It is relatively clear to me, without the experiments or even looking at the proofs, that this model will be able to capture a target triangle density, for example. I would have hoped for the experimental section to show other directions in which this model is realistic.
>
> ### **Our Response**
>
> **<Clarification 1-1>** The experimental results are beyond the repetition of theoretical results. They **validate (1) the correctness of our fitting and generation algorithms and (2) the capability of the proposed models to capture the level of clustering in real-world graphs**.
>
> **<Clarification 1-2>** The experimental results **also show that although some other metrics (global clustering coefficients and local clustering coefficients) are not directly fit, they are still well reproduced.**
>
> **<Clarification 1-3> In the initial submission, we had results that “show other directions in which this model is realistic”.**
>
> - **Section 6.2: The proposed models also generate graphs with realistic degrees and distances.**
> - **Appendix E.3: The proposed models also generate graphs with realistic modularity and core numbers.**
>     - In most cases, the modularity and core numbers in the graphs generated with binding are closer to those values in the original real-world graphs, compared to graphs generated without binding.
>
> **<Additional Experiments 1-4> During the revision, we further added four more graph metrics: conductance, average vertex/edge betweenness centrality, and natural connectivity.**
>
> - For all four additional network properties, in most cases, their values in the graphs generated with binding are closer to those values in the original real-world graphs, compared to graphs generated without binding.
> - Average vertex/edge betweenness centrality (Ref 1), and natural connectivity (Ref 2) are three different metrics used to measure the robustness and structural resilience of graphs.
>     - **[Ref 1]** Ellens and Kooij. “Graph measures and network robustness.” arXiv:1311.5064.
>     - **[Ref 2]** Chan et al. “Make it or break it: manipulating robustness in large networks.” SDM 2014.
>
> **<Revision 1-5>** We put the results of these additional experiments in the revised manuscript.
>
> - **At the end of Section 6.3 (Lines 455-457)**, we mention the additional network properties (in addition to modularity and core number that were mentioned in the initial submission).
> - The detailed results have been added in **Appendix E.3. See Tables 11 to 16** (Tables 11 and 13 were in the initial submission and the other four tables are newly added)**.**

---

> ### Author Response · Authors · 2024-11-20
> **<Response 2a> On Capturing Realistic Community Structure**
>
> ## **<Response 2a> On Capturing Realistic Community Structure**
>
> ### **Context: Related Comments**
>
> - Can binding capture realistic community structure faithfully?
>
> ### **Our Response**
>
> **<Clarification 2a-1> As mentioned above in our <Response 1>, binding generates more realistic graphs w.r.t. modularity and conductance, implying that binding captures realistic community structure faithfully.**
>
> - Modularity and conductance are both graph metrics used to measure the strength of community structure.

---

> ### Author Response · Authors · 2024-11-20
> **<Response 2b> On Limitations of Binding**
>
> ## **<Response 2b> On Limitations of Binding**
>
> ### **Context: Related Comments**
>
> - What are some types of real-world network properties you believe cannot be captured by the binding processes you outlined in the paper?
>
> ### **Our Response**
>
> **<Clarification 2b-1> In Remark 5.4, we explicitly discussed the limitations of binding.**
>
> - **Remark 5.4:** `There are EPGMs with lower subgraph densities, which are against our motivation to improve upon EIGMs w.r.t. subgraph densities and are out of this work's scope. That said, they may be useful in scenarios where dense subgraphs are unwanted, e.g., disease control.`
> - The contents were originally there in the initial submission but the index numbers changed. It was Remark 5.3 in the initial submission.
>
> **<Clarification 2b-2> In Section 7, we also mentioned this limitation.**
>
> - See “limitations and future directions”: `As discussed in Remark 5.4, binding only covers a subset of EPGMs, and we will explore the other types of EPGMs (e.g., EPGMs with lower subgraph densities) in the future.`

---

> ### Author Response · Authors · 2024-11-20
> **<Response 3> On Downstream Tasks**
>
> ## **<Response 3> On Downstream Tasks**
>
> ### **Context: Related Comments**
>
> - One thing missing from the paper, as outlined in the weaknesses part, is an experimental part showing the this specific model is useful for **downstream applications** (and not just as proxy for estimating quantities that are practically useful in downstream applications, such as triangle density). I would imagine that one way to instantiate it is by pre-training some graph neural network on this kind of graphs and show that this **pretraining** helps improve the performance on some applied benchmark. Do you think it will be possible/realistic to design such an experiment? In what context do you think this random model could come up useful?
>
> ### **Our Response**
>
> Thanks for the insightful comments.
>
> **<Argument 3-1> In our understanding, the applications of random graph models have been well established.**
>
> - Some typical applications, as listed at the beginning of the introduction section, are graph algorithm testing, statistical testing, and graph anonymization.
> - See the following surveys, where the values of random graph models are discussed.
>     - [Survey] Chakrabarti and Faloutsos. "Graph mining: Laws, generators, and algorithms." *ACM computing surveys (CSUR)* 2006.
>     - [Survey] Drobyshevskiy and Turdakov. "Random graph modeling: A survey of the concepts." *ACM computing surveys (CSUR)* 2019.
>     - [Survey] Bonifati et al. "Graph generators: State of the art and open challenges." *ACM computing surveys (CSUR)* 2020.
>
> **<Acknowledgement 3-2>** That said, we do agree that the aforementioned applications are “conventional”, and it would be interesting to see the applications of random graph models in graph **machine learning**, which has not been well explored yet.
>
> **<Preliminary Experiments and Observations 3-3>** Following your suggestions, we design the following preliminary experiments:
>
> - We consider “graph anonymization” in graph learning: The original topology of a real-world graph dataset is not available due to privacy issues, and we only have access to random graphs that are generated according to the topology.
> - We train GNN models on random graphs and test the trained GNN models on the original dataset.
>     - **Note:** The features and labels are kept as the original ones.
> - **Observation 1:** For heterophilious graphs (i.e., graphs with low label homophily), even trained with randomized graph topology, the trained GNN models can still perform well on the original datasets. Surprisingly, in some cases, the performance is even better when trained on randomized graph topology compared to when trained on the original topology. See the detailed results below.
>
> | Dataset | Trained on | ADGN | APPNP | DAGNN | FAGCN | GAT | GATv2 | GCN | GCN2 | GPRGNN | Average |
> | --- | --- | --- | --- | --- | --- | --- | --- | --- | --- | --- | --- |
> | texas | Original Topology | 0.8222 | 0.7989 | 0.6135 | 0.7700 | 0.6046 | 0.6032 | 0.6565 | 0.7859 | 0.8151 | 0.7189 |
> |  | Erdos-Renyi (ER) | 0.7443 | 0.7962 | 0.5968 | 0.6511 | 0.5908 | 0.5889 | 0.5970 | 0.7676 | 0.7754 | 0.6787 |
> |  | ER + Parallel binding | 0.7535 | 0.8014 | 0.6214 | 0.6486 | 0.5992 | 0.6043 | 0.5957 | 0.7808 | 0.7746 | 0.6866 |
> |  | Chung-Lu (CL) | 0.7454 | 0.8059 | 0.6238 | 0.6568 | 0.6268 | 0.6211 | 0.6165 | 0.7797 | 0.7765 | 0.6947 |
> |  | CL + Parallel binding | 0.7519 | 0.8124 | 0.7081 | 0.6530 | 0.6465 | 0.6381 | 0.6270 | 0.7908 | 0.7743 | 0.7113 |
> | cornell | Original Topology | 0.8314 | 0.8016 | 0.6389 | 0.7832 | 0.5822 | 0.5835 | 0.5841 | 0.7884 | 0.7686 | 0.7069 |
> |  | Erdos-Renyi (ER) | 0.8220 | 0.8390 | 0.6247 | 0.7959 | 0.6139 | 0.5902 | 0.5671 | 0.8218 | 0.7614 | 0.7151 |
> |  | ER + Parallel binding | 0.8263 | 0.8347 | 0.6541 | 0.8035 | 0.5620 | 0.5582 | 0.5686 | 0.8075 | 0.7398 | 0.7061 |
> |  | Chung-Lu (CL) | 0.8086 | 0.8382 | 0.6755 | 0.8071 | 0.5935 | 0.5902 | 0.5771 | 0.8051 | 0.7980 | 0.7215 |
> |  | CL + Parallel binding | 0.8261 | 0.8535 | 0.7343 | 0.8263 | 0.6131 | 0.6218 | 0.6314 | 0.8143 | 0.7902 | 0.7457 |
> - **Observation 2:** However, for homophilous graphs (graphs with high label homophily), training with such randomized graph topology gives much lower performance. This is likely because the randomized graph topology does not consider node features or node labels and breaks the label homophily. See more discussions in **<Challenges and Future Directions 3-4>** below.
>
> **<Challenges and Future Directions 3-4>** The random graph models considered in this work do not consider features (attributes) or labels (classes). We believe it is important to incorporate features and labels into random graph models to make them more suitable for application in graph machine learning, where features and labels are involved. We leave the applications of our models (and random graph models in general) to graph machine learning as an interesting future direction.

---

### Official Review · Reviewer_LPrH · 2024-10-28

**Soundness:** 3
**Presentation:** 2
**Contribution:** 3
**Rating:** 6
**Confidence:** 3

**Summary:**

This paper proposes the concept of binding to upgrade edge-independent random graph models, which are RGMs in which each edge is sampled independently. Binding allows for sampling a graph with given marginal probabilities for each edge, but with some dependence between the edges, allowing for more realistic graph distributions in terms certain statistics, like high clustering coefficient. The idea is to partition all possible edges, then to sample a single uniform random variable for each group in the partition, and add edges to the sampled graph if the random variable is less than edge's marginal probability. This adds edge dependence in the sense that if $e$ and $e'$ are in the same partition, and the marginal probability of the former edge is higher, then the sampling of the former guarantees the sampling of the latter. The paper then proposes a scheme for finding a partition at sampling time, as well as a parallelized scheme which relaxes the edge partition to independent edge groupings. Finally, the paper presents theoretical validation of the tractability of their methods, showing that the samples are generated in quadratic time; and empirical validation of the increased quality of samples when using binding, showing that upon fitting parameters of their model to match the input graph's triangle count, samples indeed match the triangle count closely, while not harming how well the node degree and node pair distance distributions are matched.

**Strengths:**

- The central problem that is tackled, alleviating limitations of edge-independent / inhomogeneous ER graphs, is well-motivated in the text.
- The proposed concept of binding can be applied to augment several different edge-independent random graph models.
- Experiments demonstrate the effective of binding at matching triangle on four well-known kinds of RGMs and several well-known graph datasets.
- There is theoretical work guaranteeing the tractability of the sampling algorithms.
- The organization of the paper is generally logical and clear.

**Weaknesses:**

- Some writing could be clearer. For example, at the end of page 1, the authors say they will explore EPGMs, but it is not clear at that point what it is, or whether it is a previously-proposed concept versus something being proposed in this paper. More natural language descriptions of math, e.g., Property 4.4, would facilitate reading.
- Given that binding is perhaps the main concept of this work, its formal definition should be in the main paper, and there should probably be some natural language description of the idea of general binding along with Algorithm 1 (e.g., as given in the summary above). There could also be more description of the modeling implications of binding: Does binding reflect some natural process that gives rise to real-world graphs? On the other hand, could the model assumption in binding be limiting in some way?
- At least in the main paper, the theoretical work gives guarantees about the tractability of the sampling algorithms, but there is no proof that binding can alleviate modeling issues beyond Proposition 5.2, which is fairly vague - "binding produces higher or
equal subgraph densities", but how much higher? E.g., does binding solve the motivating issue in Theorem 3.3 that edge-independent models cannot generate many triangles with low overlap?

**Questions:**

Two kinds of questions are asked above regarding 1) the modeling side of binding, and 2) possible guarantees that binding alleviates the dense subgraph count vs overlap issue that motivates the work.

### Typos / minor
- Lines 3-5 in Algorithm 2: Is this simply a for-loop from $1$ to $R$? That might be slightly clearer if so.
- Line 371 "EGPMs" -> "EPGMs"

---

> ### Author Response · Authors · 2024-11-20
>
> Dear Reviewer LPrH,
>
> Thanks for your insightful comments.
>
> Please find our responses to your individual comments below.
>
> We welcome any further comments and questions during the discussion phase.
>
> ### **Updated manuscript**
>
> We have updated the manuscript accordingly. The updated parts are $\textcolor{blue}{\text{highlighted in blue}}$. When we refer to contents in the manuscript, we **refer to the updated manuscript by default**, unless specified otherwise.
>
> - See the updated manuscript PDF at https://openreview.net/pdf?id=xljPZuprBA

---

> ### Author Response · Authors · 2024-11-20
> **<Response 1> On Writing**
>
> ## **<Response 1> On Writing**
>
> ### **Context: Related Comments**
>
> - Some writing could be clearer. For example, at the end of page 1, the authors say they will explore EPGMs, but it is not clear at that point what it is, or whether it is a previously-proposed concept versus something being proposed in this paper.
> - More natural language descriptions of math, e.g., Property 4.4, would facilitate reading.
>
> ### **Our Response**
>
> Thanks for the advice.
>
> **<Revision 1-1>** In the revised manuscript, we now explicitly mention that we propose the concept of EPGMs in this work and explain what EPGMs are at the end of page 1 (see Lines 48-50).
>
> - `We propose and explore the concept of edge probability graph models (EPGMs), i.e., RGMs that are still based on edge probabilities but do not assume edge independency`
>
> **<Clarification 1-2>** The natural language explanation of Property 4.4 is before it, where we wrote, `What can we obtain for \textit{given} edge probabilities? For this, we obtain the upper bounds of expected subgraph densities in the graphs generated by EPGMs with given edge probabilities.`
>
> **<Revision 1-3>** In the revised manuscript, we now explicitly mention that this part talks about Property 4.4.
>
> - **Revised:** For this, in **Property 4.4**, we obtain the upper bounds of expected subgraph densities in the graphs generated by EPGMs with given edge probabilities.

---

> ### Author Response · Authors · 2024-11-20
> **<Response 2a> On Binding: Definition and Description**
>
> ## **<Response 2a> On Binding: Definition and Description**
>
> ### **Context: Related Comments**
>
> - Given that binding is perhaps the main concept of this work, its formal definition should be in the main paper.
> - There should probably be some natural language description of the idea of general binding along with Algorithm 1 (e.g., as given in the summary above).
>
> ### **Our Response**
>
> Thanks for the advice.
>
> **<Revision 2a-1> In the revised manuscript, we have moved the definition of binding into the main text.** See Definition 5.1 in Section 5.1.
>
> **<Revision 2a-2> We have added some natural language description as you suggested.**
>
> - **Newly added (Lines 194-197):** Binding is the probabilistic process in Algorithm 1, **where edge dependence is imposed in each group of pairs. Specifically, in each group, if a node pair is sampled as an edge, all the pairs with higher edge probabilities must be sampled too.**

---

> ### Author Response · Authors · 2024-11-20
> **<Response 2b> On Binding: Real-World Motivations**
>
> ## **<Response 2b> On Binding: Real-World Motivations**
>
> ### **Context: Related Comments**
>
> - There could also be more description of the modeling implications of binding: Does binding reflect some natural process that gives rise to real-world graphs?
>
> ### **Our Response**
>
> **<Clarification 2b-1>** As mentioned in Section 5.1, Algorithm 1 (general binding) describes a general framework, while our practical algorithms are Algorithms 2 and 3 (local binding and parallel binding).
>
> **<Clarification 2b-2> We have real-world motivations for local binding and parallel binding**, which can be found in Section 5.2 and Appendix D.4.
>
> - In short, binding mimics group interaction in real-world systems.
> - **Section 5.2:** See Lines 237-245 in the revised manuscript (the contents were there in the initial submission but the line numbers changed; See Lines 236-244 in the initial submission).
> - **Appendix D.4:** The section provides more details about real-world motivations of binding, including specific real-world scenarios.

---

> ### Author Response · Authors · 2024-11-20
> **<Response 2c> On Binding: Limitations**
>
> ## **<Response 2c> On Binding: Limitations**
>
> ### **Context: Related Comments**
>
> - On the other hand, could the model assumption in binding be limiting in some way?
>
> ### **Our Response**
>
> **<Clarification 2c-1> Yes. In Remark 5.4, we explicitly discussed the limitations of binding.**
>
> - **Remark 5.4:** `There are EPGMs with lower subgraph densities, which are against our motivation to improve upon EIGMs w.r.t. subgraph densities and are out of this work's scope. That said, they may be useful in scenarios where dense subgraphs are unwanted, e.g., disease control.`
> - The contents were there in the initial submission but the index numbers changed. It was Remark 5.3 in the initial submission.
>
> **<Clarification 2c-2> In Section 7, we also mentioned this limitation.**
>
> - See “limitations and future directions”: `As discussed in Remark 5.4, binding only covers a subset of EPGMs, and we will explore the other types of EPGMs (e.g., EPGMs with lower subgraph densities) in the future.`

---

> ### Author Response · Authors · 2024-11-20
> **<Response 3> How Binding Alleviates “Dense Subgraph Count v.s. Overlap” Issue**
>
> ## **<Response 3> How Binding Alleviates “Dense Subgraph Count v.s. Overlap” Issue**
>
> ### **Context: Related Comments**
>
> - At least in the main paper, the theoretical work gives guarantees about the tractability of the sampling algorithms, but there is no **proof** that binding can alleviate modeling issues beyond Proposition 5.2, which is fairly vague - "binding produces higher or equal subgraph densities", but how much higher? E.g., **does binding solve the motivating issue in Theorem 3.3 that edge-independent models cannot generate many triangles with low overlap?**
>
> ### **Our Response**
>
> **<Clarification 3-1> Theoretically, “maximum binding” can achieve the upper bounds of expected subgraph densities.**
>
> - **In Property 4.4, we obtain the upper bounds** of expected subgraph densities in the graphs generated by EPGMs with given edge probabilities.
> - **In Section 5.1, we show that “maximal binding” can achieve the upper bound** obtained in Property 4.4. The formal statement is Lemma B.3 in Appendix B.3, which is also mentioned in Remark 4.5.
> - **In Remark 5.6** (Remark 5.5 in the initial submission), **we show that local binding** with $g(v) \equiv 1$ is maximal binding, and thus **also achieves the upper bound**.
> - Similarly, it is easy to see that **parallel binding** with $g(v) \equiv 1$ is also maximal binding, and thus **also achieves the upper bound**.
>
> **<Clarification 3-2>** At the same time, Property 4.7 tells us that overlap only depends on the marginal edge probabilities and thus binding does not affect overlap, which implies that **binding can achieve the upper bounds of expected subgraph densities while maintaining the same overlap, i.e., binding solves the motivating issue in Theorem 3.3.**
>
> - See also the empirical evidence in Table 4 in Section 6.6 and Table 19 in Appendix E.6 (it was Table 13 in the initial submission; some new tables were added during the revision).

---

> ### Author Response · Authors · 2024-11-20
> **Typos / Minor**
>
> ## Typos / Minor
>
> ### **Comments**
>
> - Lines 3-5 in Algorithm 2: Is this simply a for-loop from 1 to R? That might be slightly clearer if so.
>
> ### **Our Response**
>
> **<Clarification>** Thanks for the comments. In the current writing, it means a while-loop until “$P_{rem}$ becomes empty” which breaks if the condition is not satisfied within R iterations. As you suggested, it is equivalent to a for-loop from 1 to R with “$P_{rem}$ becomes empty” as an early-stop condition.
>
> **<Revision>** In the revised manuscript, we **rewrite it as a for-loop with an early-stop condition** as you suggested.
>
> ### **Comments**
>
> - Line 371 "EGPMs" -> "EPGMs".
>
> ### **Our Response**
>
> **<Revision>** Thanks. The typo has been corrected in the revised manuscript.

---

### Official Review · Reviewer_sYr6 · 2024-11-02

**Soundness:** 2
**Presentation:** 2
**Contribution:** 3
**Rating:** 6
**Confidence:** 3

**Summary:**

The paper introduces the concept of EPGM, which is wider than the edge-independent graph models. The authors propose a general binding algorithm, as well as local and parallel binding algorithms, to generate networks with high variability and clustering. They
also present closed-form results concerning triangles and discuss time complexities. The authors conduct simulation studies to evaluate their models. Overall, the idea of binding is interesting.

**Strengths:**

The authors propose a general binding algorithm, as well as local and parallel binding algorithms, to generate networks with high variability and clustering. They also present closed-form results concerning triangles and discuss time complexities.

**Weaknesses:**

1. Only provides results for triangles
2. Focus on a specific subclass of EPGM
3. Simulation results are not very convincing

**Questions:**

(a)  Similar effort can be found in the exchangeable network models, in view of relaxing conditional independence among edges. See for example

Harry Crane and Walter Dempsey. Edge exchangeable models for interaction networks.
Journal of the American Statistical Association, 113(523):1311–1326, 2018. PMID:
30467447.

Weichi Wu, Sofia Olhede, and Patrick Wolfe. Tractably modeling dependence in networks
beyond exchangeability. Bernoulli, 31(1):584 – 608, 2025

Please discuss more on this in literature.

(b) The organization of the paper can be improved. For example, Definition 3.1 and Theorem 3.3  seem to take up too much space without demonstrating any clear importance. By contrast, the key definition of binding is placed in the appendix.

(c) The authors focus a specific subclass ( good subsets ) of EPGM in section 4, however, why are the subsets considered are important? The paper can be improved if more motivation besides the feasibility are given.

(d)   The authors should compare and discuss more general models, such as the graphon model and the exponential random graph model  as mentioned in their Section 3.2 in addition to the four specific models in Section 5.4.  These models exhibit edge dependence, and is able to produce many triangles (especially the exponential random graph). It would be beneficial if the authors could provide a discussion on these models.

(e) Theorem 5.7 provides the probability result for 3-motifs only, and the proof uses an enumeration method  that could be difficult to extend to higher order motifs, which can be restrictive sometime in practice.  A way to alleviate this is to provide codes in package for motives 4,5,6 as discussed  in page 21. Also, please discuss how the closed-form varies with changes in \( p \), \( g \), and \( R \) for readers to better understand the results. Is it possible to consider several motives together?

(f) Besides triangle, other network motives such as transitivity, can be important. Could the binding methods extend to 'transitivity' and 'triangle' simultaneously?  Also please add some reference for the importance of triangle since the paper focuses on this quantity.

(g)The authors conduct fittings in their simulations, however, the discussion of fitting in the main text is quite limited. If possible, the authors could provide some discussion on the properties of fitting, such as consistency.

(h) The results in Table 1 is a little confusing. In the caption it writes ``The statistics are averaged over 100 random trails".  But the number of triangles are solved by  tractability results. The data set is given, so what does the `random trial mean'? For edge independent models, densities of triangles can be computed tractably, too. On the other hand, the results in Table 1 might not be very meaningful. First, the results show that a parameter of a model is close to the corresponding real world data. However, what really matter is that the proposed method can produce a graph with similar property in high probability, which can be evaluated using simulated mean squared of $($ground true $\Delta$-generated $\Delta$$)^2$. Second, the comparison is not meaningful. For instance, in the case of the ER model, the authors estimate the connection probability matrix using the ER graph, then optimize their model parameters (number of triangles) based on a loss function that relates to the triangles, and subsequently compare the number of triangles from their model with those from the ER graph. Then the results are surely better since the authors are optimizing the number of triangles (with respect to the observed number of triangles) in a large class of model that include the ER graph. On the other hand, ER model, or other independent edge model is not necessarily good for the real data set in addition to the discrepancy of the numbers of triangles. Therefore, improving in triangle numbers comparing with those possibly inappropriate model could be less significant in practice. The authors could consider comparing with other models that take into account triangles, such as exponential random graphs and others.

(i)  Please discuss in  detail the differences between LOCLBDG and PARABDG, as these two methods are distinct (as evidenced by the results in Figure 3). It would be beneficial to explain under which scenario one should choose LOCLBDG (or PARABDG)

---

> ### Author Response · Authors · 2024-11-20
>
> Dear Reviewer sYr6,
>
> Thanks for your detailed comments.
>
> Please find our responses to your individual comments below.
>
> We welcome any further comments and questions during the discussion phase.
>
> ### **Updated manuscript**
>
> We have updated the manuscript accordingly. The updated parts are $\textcolor{blue}{\text{highlighted in blue}}$. When we refer to contents in the manuscript, we **refer to the updated manuscript by default**, unless specified otherwise.
>
> - See the updated manuscript PDF at https://openreview.net/pdf?id=xljPZuprBA

---

> ### Author Response · Authors · 2024-11-20
> **<Response 1> On Exchangeable Network Models**
>
> ## **<Response 1> On Exchangeable Network Models**
>
> ### **Context: Related Comments (Question (a))**
>
> - Similar effort can be found in the **exchangeable network models**, in view of relaxing conditional independence among edges. See for example
>     - Harry Crane and Walter Dempsey. Edge exchangeable models for interaction networks. Journal of the American Statistical Association, 113(523):1311–1326, 2018. PMID: 30467447.
>     - Weichi Wu, Sofia Olhede, and Patrick Wolfe. Tractably modeling dependence in networks beyond exchangeability. Bernoulli, 31(1):584 – 608, 2025
> - Please discuss more on this in literature.
>
> ### **Our Response**
>
> Thanks for pointing out the related works.
>
> **<Background 1-1>** Exchangeable network models are a class of random graph models where isomorphic graphs are generated with equal probability.
>
> **<Revision 1-2> In the revised manuscript, we added additional discussions on exchangeable network models in Section 3.2 when we discuss other edge-dependent RGMs.**
>
> - **Newly added discussions:** There are also theoretical studies on edge-dependent RGMs, especially on exchangeable network models (Diaconis & Janson, 2007; Crane & Dempsey, 2018; Wu et al., 2025). However, these studies are **largely theoretical**, and existing graph generation algorithms for general exchangeable network models are **restricted to very small graphs due to scalability limitations**.
>
> **<Clarificaiton 1-3> Indeed, several special cases of exchangeable network models were already discussed in the initial submission**
>
> - In Section 3.2, we mentioned “graphon models”, which is a special case of exchangeable network models.
> - In Section 6.6, we evaluated “random geometric graphs”, which are also a special case of exchangeable network models.
> - Also, Erdos-Renyi and Erdos-Renyi with either local binding or parallel binding are also special cases of exchangeable network models due to the node equivalence.

---

> > ### Comment · Reviewer_sYr6 · 2024-11-27
> > **Discussion on Response 1**
> >
> > Thank you for the revision. I still have concerns as follows.
> >
> > First, the statement that graphon is a special case of exchangeble graph model is inadequate.  According to the Aldous-Hoover theorem, an infinite array is exchangeable if and only if it can be represented using a graphon formulation.
> >
> > Second, the statement that the generating method for graphon model are limited to very small size graph is inappropriate. For generating networks with a high density of triangles and other motifs, one can infer a suitable graphon based on the method of moment (see Theorem 1 in [1]), and generate the network efficiently according to the Aldous-Hoover theorem, i.e., once a graphon function is given, the remaining procedure is to sample i.i.d. uniformly distributed random variables. For a real network, the method of moment paper [1] proves that the empirical subgraph count can be used to estimate the moment, based on which one can construct the graphon with desired moment such as the desired number of triangle.    On the other hand, efficient estimation methods other than methods of moment for estimating graphon models, such as those presented in [2], are also available.
> >
> > Third, the previous literature I suggest are in the research line of non-exchangeble model which is certainly not graphon model. It would be beneficial for the authors to clarify whether their proposed model accommodates both exchangeable and non-exchangeable cases, and to elaborate on the advantages their model offers over the graphon framework.
> >
> > [1] Bickel, Peter J., Aiyou Chen, and Elizaveta Levina. "The method of moments and degree distributions for network models." (2011): 2280-2301.
> >
> > [2] Chan, Stanley, and Edoardo Airoldi. "A consistent histogram estimator for exchangeable graph models." International Conference on Machine Learning. PMLR, 2014.

---

> ### Author Response · Authors · 2024-11-20
> **<Response 2> On Paper Organization**
>
> ## **<Response 2> On Paper Organization**
>
> ### **Context: Related Comments (Question (b))**
>
> - The organization of the paper can be improved. For example, Definition 3.1 and Theorem 3.3 seem to take up too much space without demonstrating any clear importance.
> - By contrast, the key definition of binding is placed in the appendix.
>
> ### **Our Response**
>
> **<Revision 2-1>** In the revised manuscript, we have moved the formal definition of binding to the main text.
>
> - See Definition 5.1 in Section 5.1.
>
> **<Arguments 2-2>** We respectfully emphasize that both Definition 3.1 and Theorem 3.3 are indeed important.
>
> - Definition 3.1 formally defines “variability” so that we can numerically measure and compare the “variability” of random graph models.
> - Theorem 3.3 is the main motivation of this work, i.e., edge-independent models theoretically cannot produce high subgraph densities and high output variability.
> - Therefore, we want to keep those formal statements for the sake of formality and clarity.

---

> ### Author Response · Authors · 2024-11-20
> **<Response 3> On Subclass of EPGMs**
>
> ## **<Response 3> On Subclass of EPGMs**
>
> ### **Context: Related Comments (Question (c))**
>
> - The authors focus a specific subclass ( good subsets ) of EPGM in section 4, however, **why are the subsets considered important**? The paper can be improved if more motivation besides the feasibility are given.
>
> ### **Our Response**
>
> **<Clarification 3-1>** In Section 4, we discuss theoretical properties of EPGMs **in general**, and raise the question “How can we find good subsets?” It is **from Section 5** that we start to actually find and focus on “good subsets” of EPGMs.
>
> **<Clarification 3-2>** We respectfully emphasize that the whole Section 5 is about why the considered subset of EPGMs (binding-based EPGMs) are good: They are **realistic**, **flexible**, and **tractable**.
>
> - **Realistic: Compared to EIGMs, EPGMs with binding provably generate graphs with no-lower clustering. See Proposition 5.3.**
>     - **Proposition 5.3:** Binding produces higher or equal subgraph densities, compared to the corresponding EIGMs.
> - **Flexible: With binding, we can have the whole spectrum between minimal binding (EIGMs) and maximal binding (the theoretical upper bound).**
>     - **See the end of Section 5.1 (Lines 221-227).**
> - **Tractable: With binding, we can still derive closed-form formulae for subgraph densities. See Theorems 5.8 and B.5.**
> - **Real-world motivations: We have real-world motivations for binding, i.e., how binding mimics group interactions in real-world systems. Such contents can be found in Section 5.2 and Appendix D.4.**
>     - **Section 5.2:** See Lines 237-245.
>     - **Appendix D.4:** The whole section is about the real-world motivations of binding.

---

> ### Author Response · Authors · 2024-11-20
> **<Response 4> On Other Models**
>
> ## **<Response 4> On Other Models**
>
> ### **Context: Related Comments (Question (d))**
>
> - The authors should compare and discuss more general models, such as the graphon model and the exponential random graph model as mentioned in their Section 3.2 in addition to the four specific models in Section 5.4. These models exhibit edge dependence, and is able to produce many triangles (especially the exponential random graph). It would be beneficial if the authors could provide a discussion on these models.
>
> ### **Our Response**
>
> **<Argument 4-1> The studies on graphon models (or exchangeable network models in general) are mainly theoretical and the generation of graphon models is limited to very small graphs due to scalability issues.** Therefore, to the best of our knowledge, we cannot find practical graph generation algorithms of graphon models (or exchangeable network models in general) that are applicable to the datasets used in our experiments.
>
> - See also our **<Response 1>** above on exchangeable network models above.
>
> **<Additional Experiments 4-2> Exponential random graph models (ERGMs) failed to fit the datasets we use.**
>
> - We tried to fit ERGMs to the datasets we use w.r.t. two features: The number of edges and the number of triangles.
> - For each dataset, when given 5 hours for fitting, the fit ERGM model generates near-empty graphs in most cases (most generated graphs have fewer than 100 edges), likely because the used datasets are too large for ERGMs to converge to a meaningful distribution. See, e.g., the discussions in the following references.
>     - Snijders et al. "New specifications for exponential random graph models." Sociological methodology 36.1 (2006): 99-153.
>     - Robins. "Exponential random graph models for social networks." The Sage handbook of social network analysis (2011): 484-500.
>     - Yon et al. "Exponential random graph models for little networks." Social Networks 64 (2021): 225-238.
> - Notably, fitting ERGMs to large graphs is an active research topic itself.
>     - An. "Fitting ERGMs on big networks." Social science research 59 (2016): 107-119.
> - We used an online open-source implementation of ERGMs (https://github.com/jcatw/ergm). It would be highly appreciated if you know a better implementation of ERGMs that works better for large graphs. We are willing to evaluate it.

---

> ### Author Response · Authors · 2024-11-20
> **<Response 5> On High-Order Motifs**
>
> ## **<Response 5> On High-Order Motifs**
>
> ### **Context: Related Comments (Question (e)-1)**
>
> - Theorem 5.7 provides the probability result for 3-motifs only, and the proof uses an enumeration method that could be difficult to extend to higher order motifs, which can be restrictive sometime in practice. A way to alleviate this is to provide codes in package for motives 4,5,6 as discussed in page 21.
>
> ### **Our Response**
>
> Thanks for the advice.
>
> **<Revision 5-1>** **We implemented code for listing the possible cases we need to consider to calculate the probabilities for high-order motifs.**
>
> - Specifically, we list all the possible sequences of sampled nodes so that all the node pairs are determined.
>     - Examples for order 3: (1) {u, v, w}, (2) {u, v} → {u, v, w}, (3) {u, w} → {u, v, w}, etc., as in our proof.
> - Order 3: There are 16 possible cases in total. Among them, 4 cases involve edge dependency, as shown in our proof.
> - Order 4: There are 16205 possible cases in total. Among them, 5261 cases involve edge dependency.
> - Order 5: There are 4553161336 possible cases in total. Among them, 1642932856 cases involve edge dependency.
> - We provide both the C++ version and the Python version of the code.
>     - C++: https://anonymous.4open.science/r/epgm-7EBE/higher_order/possible_cases.cpp
>     - Python: https://anonymous.4open.science/r/epgm-7EBE/higher_order/possible_cases.ipynb
> - See all the possible cases for order 4 at https://anonymous.4open.science/r/epgm-7EBE/higher_order/possible_cases_4.txt
>
> **<Clarification 5-2>** In the initial submission, on page 21, the numbers of cases (the Bell numbers) were the number of partitions of node pairs. **Thanks to the comments, we re-examined the actual difficulty in considering higher-order motifs and realized that we have to consider many more possible cases for how the nodes are sampled.**
>
> - We leave efficient calculation/estimation for high-order motifs as a future direction.

---

> > ### Comment · Reviewer_sYr6 · 2024-11-27
> >
> > The authors point out that analyzing the 4-motif requires dealing with 5261 cases, which seems to be hard to do by hand in practice. Since the authors need to do the fitting in their simulations, further instructions on how to calculate them would be useful (e.g., is it possible to solve it automatically by computer?), as it seems to be quite nontrivial.

---

> ### Author Response · Authors · 2024-11-20
> **<Response 6> On How Motif Probabilities Vary w.r.t. Parameters**
>
> ## **<Response 6> On How Motif Probabilities Vary w.r.t. Parameters**
>
> ### **Context: Related Comments (Question (e)-2)**
>
> - Please discuss how the closed-form varies with changes in ( p ), ( g ), and ( R ) for readers to better understand the results.
>
> ### **Our Response**
>
> Thanks for the advice.
>
> **<Revision 6-1> In the revised manuscript, we added such discussions in the proof sketch of Theorem 5.8 (Theorem 5.7 in the initial submission).** See Lines 283-284.
>
> - Higher $p$ and $g$ values give higher clustering, e.g., the number of triangles.
> - The choice of $R$ is mainly for controlling the running time.

---

> ### Author Response · Authors · 2024-11-20
> **<Response 7> On Considering Multiple Motifs Together**
>
> ## **<Response 7> On Considering Multiple Motifs Together**
>
> ### **Context: Related Comments (Question (e)-3)**
>
> - Is it possible to consider several motives together?
>
> ### **Our Response**
>
> **<Clarification 7-1> Yes, it is possible. In Theorem 5.8 (Theorem 5.7 in the initial submission; also Theorem B.5), we calculate the probabilities for all the order-3 motifs.**
>
> - There are four different order-3 motifs: An empty graph, a single edge, a wedge, and a triangle.
>
> **<Clarification 7-2> In the initial submission, in Section 6.5 (Table 3), we conducted experiments where we successfully fit our models to match the numbers of triangles and wedges at the same time.**

---

> ### Author Response · Authors · 2024-11-20
> **<Response 8> On Transitivity**
>
> ## **<Response 8> On Transitivity**
>
> ### **Context: Related Comments (Question (f)-1)**
>
> - Besides triangle, other network motives such as transitivity, can be important. Could the binding methods extend to 'transitivity' and 'triangle' simultaneously?
>
> ### **Our Response**
>
> **<Clarification 8-1>** Transitivity usually means global clustering coefficient (GCC), which was **considered in our experiments**. See, e.g., Table 1.
>
> **<Clarification 8-2>** It is also **possible to fit** “transitivity” and “the number of triangles” at the same time. In the initial submission, in Section 6.5, we conducted **experiments where we successfully fit** our models to match the numbers of triangles and wedges at the same time, which is equivalent to matching the number of triangles and transitivity at the same time.
>
> - **Note:** Transitivity = Global clustering coefficient (GCC) = The ratio between the number of triangles and the number of wedges

---

> ### Author Response · Authors · 2024-11-20
> **<Response 9> On Reference for the Importance of Triangles**
>
> ## **<Response 9> On Reference for the Importance of Triangles**
>
> ### **Context: Related Comments (Question (f)-2)**
>
> - Also please add some reference for the importance of triangle since the paper focuses on this quantity.
>
> ### **Our Response**
>
> Thanks for the advice.
>
> **<Revision 9-1>** In the revised manuscript, we added discussions on “why triangles are important”: The direct target is to improve “clustering”, and the number of triangles is an important indicator of “clustering”.
>
> - See Section 6.1 (Line 405).
> - We added references:
>     - Tsourakakis et al. "Doulion: counting triangles in massive graphs with a coin." KDD’09
>     - Kolda et al. "Counting triangles in massive graphs with MapReduce." SIAM Journal on Scientific Computing 36.5 (2014): S48-S77.
>
> **<Clarification 9-2>** Another reason is that the theoretical limitations of EIGMs are also about the number of triangles (see Theorem 3.3). So improving the number of triangles is a direct remedy to the theoretical limitations.

---

> ### Author Response · Authors · 2024-11-20
> **<Response 10> On Properties of Fitting**
>
> ## **<Response 10> On Properties of Fitting**
>
> ### **Context: Related Comments (Question (g))**
>
> - The authors conduct fittings in their simulations, however, the discussion of fitting in the main text is quite limited. If possible, the authors could provide some discussion on the properties of fitting, such as consistency.
>
> ### **Our Response**
>
> Thanks for the advice.
>
> **<Clarification 10-1> Our fitting algorithms are deterministic and thus consistent.**
>
> - For Erdos-Renyi, the node-sampling probability is the same for all the nodes and we solve a single-variable equation, where no randomness is involved and the fitting results are always consistent.
> - For the other edge-probability models (Chung-Lu, SBM, and Kronecker), we use gradient descent. Since the initial parameters are fixed and we only fit a single graph, the process is deterministic with no algorithmic randomness involved.
>     - To empirically validate this, we fit the parameters in three independent trials and compared the fitting results.
>     - **Results:** The difference of node-sampling probabilities, averaged over all nodes, is consistently lower than $10^{-13}$, where the differences are likely from machine-level numerical errors (e.g., floating-point precision limitations).
>
> **<Additional Experiments 10-2> We also validated the recoverability (a.k.a., identifiability) of fitting, i.e., by observing random graphs generated with binding, we can use fitting to accurately obtain the parameters that are used to generate the observed graphs.**
>
> - **Results:** The error in node-sampling probabilities, averaged over all nodes, is consistently lower than $0.01$, and the relative error is consistently lower than 8%.
>     - Chung-Lu: 0.003787 (4.47%)
>     - SBM: 0.006257 (4.57%)
>     - Kronecker: 0.008945 (7.60%)

---

> > ### Comment · Reviewer_sYr6 · 2024-11-27
> >
> > It is commendable that the authors conducted additional simulations, which show that the estimated parameters are similar across different realizations of the random graph. However, the claim regarding "consistency" remains unjustified.
> >
> > In particular, the claim that “For Erdos-Renyi, the node-sampling probability is the same for all the nodes and we solve a single-variable equation, where no randomness is involved and the fitting results are always consistent.” is WRONG.  Erdos-Renyi is a RANDOM graph model. Every time you could generate a different graph with different estimate, while the parameters of interest $p,g,R$ stays the same. The consistency means with high probability that the estimate is close to the $p,g,R$. It is hard for me to evaluate the quality of the new approach, because when analyzing real data, the method requires an accurate estimate of graph parameters $p,g,r$, however, the paper does not justify well this estimation step.

---

> > > ### Author Response · Authors · 2024-11-28
> > > **Thank you & <Additional Response AR4> On Order-4 Motifs**
> > >
> > > Dear Reviewer sYr6,
> > >
> > > Thank you for your additional comments again. Below, we would like to provide responses regarding **order-4 motifs** and **fitting consistency**, which are particularly mentioned in your latest comments.
> > >
> > > ## **<Additional Response AR4> On Order-4 Motifs**
> > >
> > > **Reviewer’s Comments:**
> > >
> > > The authors point out that analyzing the 4-motif requires dealing with 5261 cases, which seems to be hard to do by hand in practice. Since the authors need to do the fitting in their simulations, further instructions on how to calculate them would be useful (e.g., **is it possible to solve it automatically by computer?**), as it seems to be quite nontrivial.
> > >
> > > ---
> > >
> > > **Our response:**
> > >
> > > Thank you for the constructive suggestions.
> > >
> > > ### **<Revision AR4-1> We have written additional code for automatically calculating 4-motif probabilities.**
> > >
> > > - See the Python code at https://anonymous.4open.science/r/epgm-7EBE/higher_order/compute_4_motifs.ipynb.
> > >     - Given (1) edge probabilities $p$, (2) node-sampling probabilities $g$, and (3) the maximum number of rounds $R$, the code is able to give the expected number of all possible 4-motifs in a graph generated with $p$, $g$ and $R$.
> > > - **High-level idea:** For each 4-subset of nodes and for each case, we calculate (1) the probability that the pairs are sampled in that case and (2) under that case, the probability of each motif.
> > >
> > > ### **<Additional Experiments AR4-2> We have used the additional code to calculate and fit the number of 4-cliques.**
> > >
> > > - To validate the correctness of our implementation, we test it on the Erdos-Renyi model.
> > > - We fit the number of 4-cliques in real-world datasets and examine whether the generated graphs.
> > > - See the Python code at https://anonymous.4open.science/r/epgm-7EBE/higher_order/compute_4_motifs_ER.ipynb.
> > > - **Results:** The number of 4-cliques in the simulation results is close to the ground truth in the original dataset, validating the correctness of our calculation and implementation.
> > >
> > > | dataset | ground-truth # 4-cliques | average # 4-cliques in simulation | simulation / ground truth |
> > > | --- | --- | --- | --- |
> > > | facebook | 30,004,668 | 30,253,085.2 | 1.008 |
> > > | hamsterster | 131,905 | 128,182.2 | 0.972 |
> > > | bio-CE-PG | 9,608,852 | 9,871,882.2 | 1.027 |
> > > | bio-SC-HT | 26,302,487 | 26,607,308.8 | 1.012 |
> > > | polblogs | 422,327 | 410,069.5 | 0.971 |
> > > | web-spam | 374,355 | 357,987.8 | 0.956 |
> > >
> > > ### **<Future direction AR4-3> We leave efficient calculation/estimation for high-order motifs as a future direction.**
> > >
> > > - Specifically, the current implementation for 4-motifs uses naive for loops and needs further algorithmic optimization for applications in general cases and large-scale graphs.

---

> > > > ### Author Response · Authors · 2024-11-28
> > > > **<Additional Response AR5> On Fitting Consistency**
> > > >
> > > > ## **<Additional Response AR5> On Fitting Consistency**
> > > >
> > > > **Reviewer’s Comments:**
> > > >
> > > > It is commendable that the authors conducted additional simulations, which show that the estimated parameters are similar across different realizations of the random graph. However, the claim regarding "consistency" remains unjustified.
> > > >
> > > > In particular, the claim that “For Erdos-Renyi, the node-sampling probability is the same for all the nodes and we solve a single-variable equation, where no randomness is involved and the fitting results are always consistent.” is WRONG. Erdos-Renyi is a RANDOM graph model. Every time you could generate a different graph with different estimate, while the parameters of interest $p,g,R$ stays the same.
> > > >
> > > > The consistency means with high probability that the estimate is close to the $p,g,R$. It is hard for me to evaluate the quality of the new approach, because when analyzing real data, the method requires an accurate estimate of graph parameters $p,g,r$, however, the paper does not justify well this estimation step.
> > > >
> > > > ---
> > > >
> > > > Thanks for your comments.
> > > >
> > > > ### **<Clarification AR5-1> Sorry for the confusion. In our <Response 10> above, we conducted two different types of experiments.**
> > > >
> > > > ***Our claim*** that “For Erdos-Renyi, the node-sampling probability is the same for all the nodes...” ***is about the type-1 experiments***, while we believe ***your comments are true for the type-2 experiments***.
> > > >
> > > > **[Type 1]** In <Clarification 10-1>, we conducted experiments on the consistency of **fitting results**. Specifically, for each real-world dataset, we ran our fitting algorithms multiple times **on the same original real-world graph** and validated that fitting results are consistent across different runs.
> > > >
> > > > - ***Type 1 is how we generate the graphs in our experiments.*** Specifically, we fit $p$, $g$, and $R$ to each real-world graph and generate multiple random graphs using the same $p$, $g$, and $R$.
> > > >
> > > > **[Type 2]** In <Additional Experiments 10-2>, we conducted experiments on the recoverability  (a.k.a., identifiability) of fitting results. Specifically, we
> > > >
> > > > - (1) generate random graphs using binding,
> > > > - (2) using fitting algorithms on the generated random graphs, and
> > > > - (3) compute the errors by comparing (i) the parameters fit on the generated random graphs and (ii) the actual parameters used to generate those random graphs.
> > > >
> > > > ---
> > > >
> > > > ### **<Additional Experiments AR5-2> For the type-2 experiments, we further added experiments on the Erdos-Renyi model.**
> > > >
> > > > - Thanks to your comments, we realize that we did not include the Erdos-Renyi model for the type-2 experiments in <Additional Experiments 10-2>, which might cause the confusion.
> > > > - We further conduct experiments on the recoverability  (a.k.a., identifiability) of fitting results on the Erdos-Renyi model.
> > > > - **Results:** Our previous conclusion still holds. That is, the error in node-sampling probabilities, averaged over all nodes, is consistently lower than $0.01$, and the relative error is consistently lower than 8%. Also, the errors on the Erdos-Renyi model are the lowest.
> > > >     - Erdos-Renyi: 0.000081 (0.20%)
> > > >     - (Below are the results already included in our <Response 10> above)
> > > >     - Chung-Lu: 0.003787 (4.47%)
> > > >     - SBM: 0.006257 (4.57%)
> > > >     - Kronecker: 0.008945 (7.60%)
> > > >
> > > > ---
> > > >
> > > > ### **<Clarification AR5-3> For the type-1 experiments, the fitting algorithms are indeed deterministic.**
> > > >
> > > > - The experiments in <Clarification 10-1> are on the ground-truth real-world datasets, not on the simulation results.
> > > > - The fitting algorithms on the Erdos-Renyi model (when fitting to a single real-world graph) indeed solve a single-variable equation on $g$.
> > > >     - See https://anonymous.4open.science/r/epgm-7EBE/fitting/ER_iid.wls and https://anonymous.4open.science/r/epgm-7EBE/fitting/ER_iter.wls.
> > > >
> > > > ---
> > > >
> > > > ### **<Clarification AR5-4> We believe that the results in our paper show that our models “accurately estimate” real-world data, well reproducing the graph properties.**
> > > >
> > > > - In our understanding, the results in our paper show that our algorithms can **accurately find** proper $g$, $p$, and $R$ for our model to **well reproduce the graph properties** of real-world graph data.
> > > > - **In Section 6.2**, we show that the proposed models well reproduce several clustering-related properties: **The number of triangles**, **average local clustering coefficient**, and **global clustering coefficient**.
> > > > - **In Section 6.3**, we show that the proposed models well reproduce **degree distributions** and **distance distributions**.
> > > > - **In Appendix E.3**, we show that the proposed models well reproduce six other graph properties: **modularity**, **core numbers**, **conductance**, **average vertex/edge betweenness**, and **natural connectivity**.
> > > > - It would be highly appreciated if the reviewer could suggest other specific ways to evaluate the quality of our models, and we would be willing to validate them additionally.

---

> > > > > ### Comment · Reviewer_sYr6 · 2024-11-28
> > > > >
> > > > > Thanks for the revision and I raise the point.

---

> > > > > > ### Author Response · Authors · 2024-11-28
> > > > > > **Thank you and we will include our fruitful discussions in the final manuscript**
> > > > > >
> > > > > > Dear Reviewer sYr6,
> > > > > >
> > > > > > Thanks for raising the score and for all the constructive comments.
> > > > > > We will make sure our fruitful discussions are included in the final manuscript.

---

> ### Author Response · Authors · 2024-11-20
> **<Response 11> On “Random Trails"**
>
> ## **<Response 11> On “Random Trails”**
>
> ### **Context: Related Comments (Question (h)-1)**
>
> - The results in Table 1 is a little confusing. In the caption it writes “The statistics are averaged over 100 random trails”. But the number of triangles are solved by tractability results. The data set is given, so what does the “random trial” mean? For edge independent models, densities of triangles can be computed tractably, too.
>
> ### **Our Response**
>
> **<Clarification 11-1> The results are the average results of simulations. The “random trials” are multiple runs of the generation algorithms.**
>
> **<Argument 11-2> Such simulation results are important and insightful.**
>
> - Indeed, we can compute the number of triangles for both the proposed models and the edge independent models, but the simultation results can **validate (1) the correctness of our fitting and generation algorithms and (2) the capability of the proposed models to capture the level of clustering in real-world graphs**.
> - Also, there are **other metrics** that are not easy to calculate, which requires simulation results.
>     - **Table 1:** Global clustering coefficients and local clustering coefficients
>     - **Section 6.3: Degrees and distances**
>     - **Appendix E.3: Modularity, conductance, core numbers, average vertex/edge betwenness centrality, natural connectivity**

---

> ### Author Response · Authors · 2024-11-20
> **<Response 12> On Performance Evaluation Metric**
>
> ## **<Response 12> On Performance Evaluation Metric**
>
> ### **Context: Related Comments (Question (h)-2)**
>
> - On the other hand, the results in Table 1 might not be very meaningful. First, the results show that a parameter of a model is close to the corresponding real world data. However, what really matter is that the proposed method can produce a graph with similar property in high probability, which can be **evaluated using simulated mean squared errors**.
>
> ### **Our Response**
>
> **<Clarification 12-1>** As mentioned above in our <Response 11>, the results in Table 1 are simulation results, which **validate (1) the correctness of our fitting and generation algorithms and (2) the capability of the proposed models to capture the level of clustering in real-world graphs**. Also, the results in Table 1 show that although **some other metrics** (global clustering coefficients and local clustering coefficients) are **not directly fit**, they are **still well reproduced**.
>
> **<Clarification 12-2> In the initial submission, we provided the standard deviations** of the simulation statistics. See Table 7 in Appendix E.2.
>
> **<Revision 12-3> In the revised manuscript, we further provided the simulated mean squared errors as suggested.** See Table 8 in Appendix E.2.
>
> - **Result summary: *In most cases, with binding, the simulated mean squared errors are lower than those for the edge independent models.***

---

> ### Author Response · Authors · 2024-11-20
> **<Response 13> On Comparison**
>
> ## **<Response 13> On Comparison**
>
> ### **Context: Related Comments (Question (h)-3)**
>
> **Reviewer comments:**
>
> - Second, the comparison is not meaningful. For instance, in the case of the ER model, the authors estimate the connection probability matrix using the ER graph, then optimize their model parameters (number of triangles) based on a loss function that relates to the triangles, and subsequently compare the number of triangles from their model with those from the ER graph. Then the results are surely better since the authors are optimizing the number of triangles (with respect to the observed number of triangles) in a large class of model that include the ER graph.
> - On the other hand, ER model, or other independent edge model is not necessarily good for the real data set in addition to the discrepancy of the numbers of triangles. Therefore, improving in triangle numbers comparing with those possibly inappropriate model could be less significant in practice.
> - The authors could consider comparing with other models that take into account triangles, such as exponential random graphs and others.
>
> ### **Our Response**
>
> **<Clarification 13-1>** As mentioned above in our <Response 11> and <Response 12>, the results **validate (1) the correctness of our fitting and generation algorithms and (2) the capability of the proposed models to capture the level of clustering in real-world graphs**. Also, the results in Table 1 show that although **some other metrics** (global clustering coefficients and local clustering coefficients) are **not directly fit**, they are **still well reproduced**.
>
> **<Clarification 13-2>** Our point is that binding can improve upon **various edge-probability models,** **even simple and weak ones like ER**.
>
> **<Additional Experiments 13-3> We tried exponential random graph models (ERGMs), and they generated near-empty graphs in most cases.** See our <Response 4> above.
>
> **<Clarification 13-4> As the reviewer also noticed, we compared with other models that are able to generate graphs with many triangles, e.g., BTER and random geometric graphs (RGGs), and discussed their problems.**
>
> - See Section 6.6 (Table 4) and Appendix E.6 (Table 19).

---

> ### Author Response · Authors · 2024-11-20
> **<Response 14> On Local Binding v.s. Parallel Binding**
>
> ## **<Response 14> On Local Binding v.s. Parallel Binding**
>
> ### **Context: Related Comments (Question (i))**
>
> - Please discuss in detail the differences between LOCLBDG and PARABDG, as these two methods are distinct (as evidenced by the results in Figure 3). It would be beneficial to explain under which scenario one should choose LOCLBDG (or PARABDG)
>
> ### **Our Response**
>
> **<Clarification 14-1> A clear superiority of parallel binding is speed, i.e., parallel binding is faster than local binding.**
>
> - **Empirically, parallel binding uses much less running time than local binding.**
> - See the running time in Table 2 in Section 6.4.
> - See also Table 17 in Appendix E.4.
>
> **<Clarification 14-2> Regarding the “quality” of generated graphs, as observed in Table 1, when both fitting the number of triangles, in most cases, local binding generates graphs with lower global clustering coefficient (GCC) and average local clustering coefficient (ALCC) than parallel binding.**
>
> - This is not necessarily good or bad, but depending on the specific graphs you want to fit.

---

> ### Author Response · Authors · 2024-11-27
> **Thank you for your additional comments & Our clarification and revision (part 1/3)**
>
> Dear Reviewer sYr6,
>
> Thank you for your additional comments. Below, we would like to provide responses regarding **the graphon model** and **(non-)exchangeable network models**, which are particularly mentioned in your latest comments.
>
> Also, we welcome further comments and discussions during the discussion phase.
>
> ---
>
> ### **Updated manuscript**
>
> We have updated the manuscript accordingly. The updated parts are $\textcolor{blue}{\text{highlighted in blue}}$.
> - See the updated manuscript PDF at https://openreview.net/pdf?id=xljPZuprBA
>
> ---
>
> ## **<Overall Revision> We have corrected and enriched the discussions on (non-)exchangeable network models**
>
> Thank you for your comments and suggestions. In the revised manuscript (https://openreview.net/pdf?id=xljPZuprBA), **we have corrected some statements and added more discussions regarding (non-)exchangeable network models**.
>
> In Section 3.2, when we introduce (non-)exchangeable network models, we now write:
>
> >Exchangeable network models (ENMs) (Lovasz & Szegedy, 2006; Diaconis & Janson, 2007) also involve edge dependency, where isomorphic graphs are generated with the same probability (i.e., all nodes are treated probabilistically in symmetry). However, ENMs cannot generate graphs with sparsity and power-law degrees, which are common patterns in real-world graphs (Crane & Dempsey, 2016). Recent efforts have introduced *asymmetry among nodes* to enhance expressiveness (Crane & Dempsey, 2018; Wu et al., 2025). In the same spirit but from a different perspective, we aim to improve expressiveness by introducing *dependence among edges* upon EIGMs. Since we build RGMs based on edge-probability models, the nodes are asymmetric (i.e., non-exchangeable), except for the Erdos-Renyi model with uniform edge probabilities.
>
> In Section 5.4, when we introduce parameter fitting for the Erdos-Renyi model, we have added related discussions:
>
> > As mentioned in Section 3.2, the ER model is the only case with node exchangeability, and the exchangeability is preserved with binding since the nodes are also treated symmetrically for binding.
>
> Please let us know if any other additional discussions and/or corrections are needed.
> ---
>
> ### **Below are some detailed responses related to your specific comments.**
>
> ## **<Additional Response AR1> On “Graphon Models” v.s. “Exchangeable Network Models”**
>
> **Reviewer’s comments:**
>
> The statement that graphon is a special case of exchangeable graph models is inadequate. According to the Aldous-Hoover theorem, an infinite array is exchangeable if and only if it can be represented using a graphon formulation.
>
> ---
>
> **<Revision AR1-1>** In the revised manuscript, we have added more comprehensive and rigorous discussions on (non-)exchangeable network models, where **we have avoided such imprecise expressions that may cause confusion**.
>
> ---
>
> **<Clarification AR1-2>** That said, some works still use the term “graphon” as “a special case” of exchangeable network models. For example, in [r1], the authors discuss “graphon” and “graphex” as two "specific cases" of exchangeable network models.
> - [r1] Li et al. "Recent advances on mechanisms of network generation: Community, exchangeability, and scale‐free properties." Wiley Interdisciplinary Reviews: Computational Statistics 16.2 (2024): e1651.

---

> > ### Author Response · Authors · 2024-11-27
> > **Thank you for your additional comments & Our clarification and revision (part 2/3)**
> >
> > ## **<Additional Response AR2> On Practical Graph Generation of Exchangeable Network Models**
> >
> > **Reviewer’s comments:**
> >
> > The statement that the generating methods for the graphon model are limited to very small-size graphs is inappropriate.
> >
> > - For generating networks with a high density of triangles and other motifs, one can infer a suitable graphon based on the method of moment (see Theorem 1 in [1]), and generate the network efficiently according to the Aldous-Hoover theorem, i.e., once a graphon function is given, the remaining procedure is to sample i.i.d. uniformly distributed random variables.
> > - For a real network, the method of moment paper [1] proves that the empirical subgraph count can be used to estimate the moment, based on which one can construct the graphon with a desired moment, such as the desired number of triangles.
> > - On the other hand, efficient estimation methods other than methods of moment for estimating graphon models, such as those presented in [2], are also available.
> > - [1] Bickel, Peter J., Aiyou Chen, and Elizaveta Levina. "The method of moments and degree distributions for network models." (2011): 2280-2301.
> > - [2] Chan, Stanley, and Edoardo Airoldi. "A consistent histogram estimator for exchangeable graph models." International Conference on Machine Learning. PMLR, 2014.
> >
> > ---
> >
> > Thanks for pointing out the related works.
> >
> > **<Additional Experiments AR2-1>** We tried the method in [2] for graph generation
> >
> > - We used the official MATLAB code by the authors of [2] at https://github.com/airoldilab/SAS to conduct inference of graphons and graph generation on the real-world graph datasets used in our experiments.
> > - **Results: In most cases, the graphs generated by the method in [2] have much lower clustering than the ground truth.**
> > - It would be highly appreciated if you know a better implementation of any exchangeable (or non-exchangeable) network model. We are willing to evaluate it.
> >
> > | dataset | normalized number of triangles | GCC | ALCC |
> > | --- | --- | --- | --- |
> > | facebook (ground truth) | 1.000 | 0.519 | 0.606 |
> > | facebook ([2]) | 0.240 | 0.129 | 0.038 |
> > | hamsterster (ground truth) | 1.000 | 0.229 | 0.540 |
> > | hamsterster ([2]) | 0.261 | 0.067 | 0.031 |
> > | web-spam (ground truth) | 1.000 | 0.145 | 0.286 |
> > | web-spam ([2]) | 0.470 | 0.080 | 0.028 |
> > | polblogs (ground truth) | 1.000 | 0.226 | 0.320 |
> > | polblogs ([2]) | 0.576 | 0.133 | 0.132 |
> > | bio-CE-PG (ground truth) | 1.000 | 0.321 | 0.447 |
> > | bio-CE-PG ([2]) | 0.636 | 0.199 | 0.201 |
> > | bio-SC-HT (ground truth) | 1.000 | 0.377 | 0.350 |
> > | bio-SC-HT ([2]) | 0.820 | 0.305 | 0.153 |
> >
> > ---
> >
> > **<Revision AR2-2>** In the revised manuscript, we have added more comprehensive and rigorous discussions on (non-)exchangeable network models, where **we have avoided such controversial arguments**.
> >
> > ---
> >
> > **<Clarification AR2-3>** After inferring a graphon, the “remaining procedure” might be inefficient for large-scale graphs.
> >
> > - In [r1] (see Section 3.3 of [r1]), the authors discussed the scalability issue in generating graphs after obtaining a graphon.
> > - **Quote:** An essential unresolved problem is how to sample graphs from a given graphon or graphex efficiently. The graph generation process … leads to a sampling algorithm with time complexity $O(N^2)$.
> > - **Quote:** Pareno (2022) further proposed a sampling algorithm with time complexity $O(N\ln N^2)$. It is highly desirable to design efficient algorithms to reduce the time complexity and apply them to large-scale networks.
> > - On the contrary, we have applied **our model to generate graphs with up to 64 million nodes.** See our <Response 3> to Reviewer qw53 (https://openreview.net/forum?id=xljPZuprBA&noteId=ecsT8DYQqM).
> > - [r1] Li et al. "Recent advances on mechanisms of network generation: Community, exchangeability, and scale‐free properties." Wiley Interdisciplinary Reviews: Computational Statistics 16.2 (2024): e1651.

---

> > > ### Author Response · Authors · 2024-11-27
> > > **Thank you for your additional comments & Our clarification and revision (part 3/3)**
> > >
> > > ## **<Additional Response AR3> On Non-Exchangeable Models**
> > >
> > > **Reviewer’s comments:**
> > >
> > > - The previous literature I suggest are in the research line of non-exchangeble model which is certainly not graphon model.
> > > - It would be beneficial for the authors to clarify whether their proposed model accommodates both exchangeable and non-exchangeable cases
> > > - and to elaborate on the advantages their model offers over the graphon framework.
> > >
> > > ---
> > >
> > > Thanks for the suggestions.
> > >
> > > ### **<Revision AR3-1>** Such non-exchangeable models are now more properly discussed.
> > >
> > > In Section 3.2, we now write (parts regarding non-exchangeable models highlighted):
> > >
> > > > Exchangeable network models (ENMs) (Lovasz & Szegedy, 2006; Diaconis & Janson, 2007) also involve edge dependency, where isomorphic graphs are generated with the same probability (i.e., all nodes are treated probabilistically in symmetry). **However, ENMs cannot generate graphs with sparsity and power-law degrees, which are common patterns in real-world graphs (Crane & Dempsey, 2016). Recent efforts have introduced *asymmetry among nodes* to enhance expressiveness (Crane & Dempsey, 2018; Wu et al., 2025).**
> > >
> > > ### **<Clarification & Revision AR3-2>** Indeed, the proposed model accommodates both exchangeable and non-exchangeable cases.
> > >
> > > Specifically, only the case with uniform edge probabilities, i.e., when built upon the Erdos-Renyi model, is exchangeable.
> > >
> > > In Section 3.2, we now write (related parts highlighted):
> > >
> > > > In the same spirit but from a different perspective, we aim to improve expressiveness by introducing *dependence among edges* upon EIGMs. **Since we build RGMs based on edge-probability models, the nodes are asymmetric (i.e., non-exchangeable), except for the Erdos-Renyi model with uniform edge probabilities.**
> > >
> > > In Section 5.4, when we introduce parameter fitting for the Erdos-Renyi model, we have added related discussions (related parts highlighted):
> > >
> > > > As mentioned in Section 3.2, **the ER model is the only case with node exchangeability, and the exchangeability is preserved with binding** since the nodes are also treated symmetrically for binding.

---

### Official Review · Reviewer_qw53 · 2024-11-02

**Soundness:** 3
**Presentation:** 2
**Contribution:** 2
**Rating:** 5
**Confidence:** 4

**Summary:**

The authors propose a generative random graph model that aims to balance realism of the generated graphs with computational efficiency of the generation process, seeking to increase clustering and variability in graphs generated by traditional models like Erdős-Rényi and stochastic block models. To achieve this, the paper uses an edge-dependent graph model (EPGM), which they use to create controlled dependencies among edges, and proposes a tractable algorithm for graph generation. The study includes a theoretical framework, practical algorithms, and experimental validation to demonstrate the generation of graphs.

**Strengths:**

- The paper builds on a solid theoretical foundation of edge-dependent random graph models, extending existing work to improve clustering and variability.
- The supplementary materials provide sufficient details for understanding and reproducing the experiments.
- The paper demonstrates a clustering and variability across by fitting different real-world datasets.
- The paper offers a _potentially_ useful tool for generating random graphs with enhanced clustering and variability by fitting model parameters to specific datasets.
- The authors provide a thorough reproducibility package, including all experimental workflows, parameters, and scripts, though this package could benefit from more comprehensive documentation.

**Weaknesses:**

- **Validation**: The term "desirable" graph properties---namely, realistic structures, patterns, and variability---is vague, highly domain-dependent, and lacks formalization in the paper, making it challenging to assess whether the proposed model achieves these goals. A formal definition and visual examples earlier in the text could maybe help clarify these aims.
- **Limited Exploration of Desirable Properties**: The paper does not fully explore the binding approach’s effects on other important network properties, such as modularity, conductance, and resilience, which are also highly desirable in certain domains.
- **Dependence on Parameter Fitting**: The model generates graphs after fitting model parameters to real-world networks, rather than directly from scratch. It is unclear whether this model could lend itself to straightforward manual parameterization to generate synthetic graphs independently, a feature offered by models like the Lancichinetti-Fortunato-Radicchi (LFR) benchmark graph generator. This suggest that the graph generator has a narrow practical use.
- **Scalability**: It is unclear if the model can efficiently generate realistic extremely large graphs, given that the experimental results focus on relatively small networks.
- **Unclear Tractability**: The explanation of the binding schemes' tractability could be more detailed and intuitive, especially as it seems that binding is computationally slower than edge-independent methods, which could hinder its scalability to larger graphs.
- **Comparison with Established Benchmarks**: The paper does not compare with the Lancichinetti–Fortunato–Radicchi (LFR) benchmark data generator, which also generates graphs with **power-law degree distributions, variability, and high clustering**. Given that the LFR model shares similar motivations and goals with the proposition, a comparison here would add necessary context. Additionally, faster and more efficient follow-up models to LFR align well with the goals of this study, particularly in terms of tractability.

**Questions:**

-/-

---

> ### Author Response · Authors · 2024-11-20
>
> Dear Reviewer qw53,
>
> Thanks for your detailed comments.
>
> Before **responding to your individual comments**, we would like first to **provide some overall clarification** to clear potential misunderstandings and make sure we are on the same page.
>
> We welcome any further comments and questions during the discussion phase.
>
> ### **Updated manuscript**
>
> We have updated the manuscript accordingly. The updated parts are $\textcolor{blue}{\text{highlighted in blue}}$. When we refer to contents in the manuscript, we **refer to the updated manuscript by default**, unless specified otherwise.
>
> - See the updated manuscript PDF at https://openreview.net/pdf?id=xljPZuprBA

---

> ### Author Response · Authors · 2024-11-20
> **<Overall Clarification OC1> Tractability ≠ Scalability**
>
> ## **<Overall Clarification OC1> Tractability ≠ Scalability**
>
> ### **Context: Related Comments**
>
> - The explanation of the binding schemes' **tractability** could be more detailed and intuitive, especially as it seems that binding is **computationally slower** than edge-independent methods.
> - **Faster and more efficient** follow-up models to LFR align well with the goals of this study, particularly in terms of **tractability**.
>
> ### **Our Clarification**
>
> **<Clarification OC1-1>** In this work, ***“tractability” is NOT something similar to “scalability”*** (e.g., the ability to perform on large-scale graphs).
>
> - We say a random graph model (RGM) is “tractable” if users ***can compute and control graph statistics*** of the graphs generated by the RGM, as also emphasized in the manuscript.
>     - **Note:** Below, the line numbers are according to the updated manuscript, but the contents were originally there in the initial submission.
>     - **The first sentence of the abstract (Lines 14-15)**: `Remain tractable to compute and control graph statistics`
>     - **The first paragraph of the introduction (Line 36)**: `RGMs should be tractable, i.e., we can compute and control graph statistics`
>     - **Discussions in Section 3.2 (Line 128-129)**: `The closed-form tractability results on subgraph densities derived by us are usually unavailable for existing RGMs with edge dependency`
>     - **The first paragraph in Section 5 (Line 185)**: `In a tractable (i.e., controllable graph statistics) way`
> - **Tractability is one of our main focuses** in this work, in addition to the realisticness and variability mentioned by the reviewer.
> - See also our **<Response 3>** below on “scalability” and “tractability”.
>
> **<Reivison OC1-2>** In the revised manuscript, we further clarify this point.
>
> - We added a footnote in the introduction. See Footnote 2.
> - **Footnote 2 (newly added):** In this work, *tractability* refers to the feasibility of deriving graph statistics, rather than the ability to handle large-scale graphs (which we refer to as *scalability*).

---

> ### Author Response · Authors · 2024-11-20
> **<Overall Clarification OC2a> Proposed Models Can Generate Graphs From Scratch Without Fitting Specific Graphs**
>
> ## **<Overall Clarification OC2a> Proposed Models Can Generate Graphs From Scratch Without Fitting Specific Graphs**
>
> ### **Context: Related Comments**
>
> - The paper offers a potentially useful tool for generating random graphs with enhanced clustering and variability by **fitting model parameters to specific datasets**.
> - The model generates graphs after **fitting model parameters to real-world networks**, rather than **directly from scratch**.
>
> ### **Our Clarification**
>
> **<Clarification OC2a-1> *The proposed models can be used to generate graphs “from scratch”.*** To do so, instead of finding the parameters by fitting them to existing graphs, one can just freely choose parameters (the underlying edge probabilities and the parameters for binding).
>
> - Indeed, the Chung-Lu model, the stochastic block model, and the stochastic Kronecker model have all been used to generate benchmark graphs “from scratch” without fitting specific graphs.
>     - **Chung-Lu:** Fasino, Dario, Arianna Tonetto, and Francesco Tudisco. "Generating large scale‐free networks with the Chung-Lu random graph model." Networks 78.2 (2021): 174-187.
>     - **SBM:** Tsitsulin et al. "Synthetic graph generation to benchmark graph learning." arXiv preprint arXiv:2204.01376 (2022).
>     - **Kronecker:** Murphy et al. "Introducing the graph 500." Cray Users Group (CUG) 19.45-74 (2010): 22.
> - Once edge probabilities are chosen, one can freely vary the parameters for binding (specifically, the node-sampling probabilities $g$; See Algorithms 2 and 3).
>
> **<Additional Experiments OC2a-2>** We conducted additional experiments, using the proposed models to generate graphs “from scratch”.
>
> - ***Key message: By choosing proper edge probabilities and varying node-sampling probabilities, we are able to generate a spectrum (series) of random graphs with different levels of clustering.***
>
> **<Revision OC2a-3>** In the revised manuscript, we put the **results** of these additional experiments, with **discussions** and **insights**.
>
> - In Section 6.1 (Lines 409-411), when we introduce “fitting”, we now mention that it is also possible to use EPGMs with binding to generate graphs “from scratch” with different levels of clustering by directly setting the parameters (without fitting specific graphs).
> - In Appendix E.7 (Tables 21-24), we include the detailed experimental settings and results.
> - In Appendix E.7, we also discuss the following insights:
>     - In line with the theoretical analysis, when we increase node-sampling probabilities, the clustering of generated graphs also increases.
>     - That is, we can use different node-sampling probabilities to generate graphs with different levels of clustering.
>     - There are also interesting observations on the correlation between node-sampling probabilities and some parameters in the underlying edge-probability models.
>         - For Chung-Lu, when node-sampling probabilities are positively correlated to the node degrees, the generated graphs have higher clustering.
>         - For the stochastic block model, when node-sampling probabilities are negatively correlated to the node degrees, the generated graphs have relatively lower clustering, while positive correlation and no correlation give similar results.
>         - For Kronecker, when node-sampling probabilities are positively correlated to the number of ones in the binary node labels, the generated graphs have higher clustering.
>
> **<Clarification OC2a-4> Our theoretical analysis on binding is general and not limited to fitting specific datasets.** It is only in the experiments that we fit our models to specific datasets, showing that our models can actually well reproduce the properties in real-world graphs.

---

> ### Author Response · Authors · 2024-11-20
> **<Overall Clarification OC2b> Fitting Specific Graphs and Generating Graphs From Scratch Are Both Useful**
>
> ## **<Overall Clarification OC2b> Fitting Specific Graphs and Generating Graphs From Scratch Are Both Useful**
>
> ### **Context: Related Comments**
>
> - The paper offers a potentially useful tool for generating random graphs with enhanced clustering and variability by **fitting model parameters to specific datasets**.
> - The model generates graphs after **fitting model parameters to real-world networks**, rather than **directly from scratch**.
>
> ### **Our Clarification**
>
> **<Background OC2b-1>** Random graph models (RGMs) have different usages, including “generating random graphs from scratch” and “fitting the model to specific graphs and generating similar graphs”.
>
> **<Clarification OC2b-2>** These two usages are complementary—each addresses distinct needs within the broader context of network analysis. Specifically, ***“fitting RGMs to specific graphs and generating similar graphs” has significant values with many applications.***
>
> - In many real-world applications, we want random graphs that are ***similar to specific real-world graphs at hand***. As typical examples, consider the applications mentioned in the first paragraph of the introduction (Lines 31-34).
> - **[Graph algorithm testing]** When we want to test graph algorithms on real-world graphs but lack sufficient data, we generate random graphs as “surrogate data.” For such “surrogate data” to be meaningful—i.e., to reasonably conclude that “if the algorithms perform well on these random graphs, they should also perform well on the original real-world graphs”—those random graphs should be ***similar to the specific real-world graphs we want to test***.
>     - **Ref:** Mishra et al. "An effective comparison of graph clustering algorithms via random graphs." International Journal of Computer Applications 22.1 (2011): 22-27.
> - **[Statistical testing]** When we want to study the statistical significance of some properties in real-world graphs, we generate random graphs and compare real-world graphs with random ones. For such a comparison to be meaningful—i.e., to confidently determine whether the observed properties can be caused by random chance—we should make these random graphs ***similar to the specific real-world graphs we want to test*** w.r.t. the graph metrics and properties other than the ones to be tested.
>     - **Ref:** Wandelt et al. "On the use of random graphs as null model of large connected networks." Chaos, Solitons & Fractals 119 (2019): 318-325.
> - **[Graph anonymization]** When we have real-world graphs containing sensitive information and want to anonymize them for public release, we generate random graphs that are ***similar to the specific real-world graphs we want to release*** and release those random graphs instead. Otherwise, the released graphs can be too different and even irrelevant, and thus less meaningful.
>     - **Ref:** Casas-Roma et al. "A survey of graph-modification techniques for privacy-preserving on networks." Artificial Intelligence Review 47 (2017): 341-366.
> - That said, in some cases, we need random graphs that catch some typical behaviors of real-world graphs at a high level, with no specific graphs as direct references. In such cases, “graph generation from scratch” is useful. A typical usage is to generate benchmark datasets, as mentioned by the reviewer.

---

> ### Author Response · Authors · 2024-11-20
> **<Response 1a> On “Desirable Properties”: Evaluation**
>
> ## **<Response 1a> On “Desirable Properties”: Evaluation**
>
> ### **Context: Related Comments**
>
> - The paper does not fully explore the binding approach’s effects on other important network properties, such as modularity, conductance, and resilience, which are also highly desirable in certain domains.
>
> ### **Our Response**
>
> **<Clarification 1a-1> *In the initial submission,*** we actually examined other network properties: ***degrees***, ***distances***, ***modularity***, and ***core numbers***.
>
> - **[Degrees and distances] In the initial submission, in Section 6.2 (Fig. 1),** we show that, binding does not negatively affect degree or distance distributions, and provides improvements sometimes.
> - **[Modularity and core numbers] In the initial submission, in Appendix E.3 (Tables 11 & 13), we show that, with binding, the generated graphs are more realistic w.r.t. modularity and core numbers.**
>     - It was Appendix E.3 (Tables 10 & 11) in the initial submission.
>     - **It is mentioned in the main text** at the end of Section 6.3 (see Lines 455-457 in the revised manuscript or Lines 456-458 in the initial submission).
>     - **Key results:** In most cases, ***the modularity and core numbers in the graphs generated with binding are closer to those values in the original real-world graphs***, compared to graphs generated without binding.
>
> **<Additional Experiments 1a-2>** We conducted additional experiments, further examining ***four other*** network properties: ***conductance***, ***average vertex/edge betweenness centrality***, and ***natural connectivity***. We show that, with binding, the generated graphs are **also more realistic w.r.t. those four additional network properties**.
>
> - **Key results:** For all four additional network properties, in most cases, ***their values in the graphs generated with binding are closer to those values in the original real-world graphs***, compared to graphs generated without binding.
> - Average vertex/edge betweenness centrality (Ref 1), and natural connectivity (Ref 2) are three different metrics used to measure the structural **robustness and resilience** of graphs.
>     - **[Ref 1]** Ellens and Kooij. “Graph measures and network robustness.” arXiv:1311.5064.
>     - **[Ref 2]** Chan et al. “Make it or break it: manipulating robustness in large networks.” SDM 2014.
>
> **<Revision 1a-3>** We put the results of these additional experiments in the revised manuscript.
>
> - **At the end of Section 6.3 (Lines 455-457)**, we mention the additional network properties (in addition to modularity and core number that were mentioned in the initial submission).
> - The detailed results have been added in **Appendix E.3. See Tables 11 to 16** (Tables 11 and 13 were in the initial submission and the other four tables are newly added)**.**

---

> ### Author Response · Authors · 2024-11-20
> **<Response 1b> On “Desirable Properties”: Definition**
>
> ## **<Response 1b> On “Desirable Properties”: Definition**
>
> ### **Context: Related Comments**
>
> - The term "desirable" graph properties---namely, realistic structures, patterns, and variability---is vague, highly **domain-dependent**, and lacks **formalization** in the paper, making it challenging to assess whether the proposed model achieves these goals. A **formal definition** and **visual examples** earlier in the text could maybe help clarify these aims.
>
> ### **Our Response**
>
> **<Clarification 1b-1> What are the “desirable properties” in this work?**
>
> - The three “desirable properties” of random graph models (RGMs) we focus in this work are (1) **realisticness**, (2) **variability**, and (3) **tractability**.
> - There were potential misunderstandings regarding “tractability”. See our <Overall Clarification OC1> above, where we clarified that “tractability ≠ scalability”.
>
> **<Revision 1b-2>** In the revised manuscript, we ***clarified the specific targets*** regarding those “desirable properties” at the **end of Section 3 (Lines 131-136)**.
>
> - **Realisticness:** Reproducing patterns observed in real-world graphs across different domains, e.g., high clustering, power-law degrees, and small diameters.
>     - See also, e.g., Lines 34-35: `Desirable RGMs should generate graphs with realistic patterns, such as high clustering, power-law degrees, and small diameters.`
> - **Variability:** Generating graphs with low overlap (Definition 3.1).
>     - See also, e.g., Lines 91-92: `They defined the concept of overlap to measure the variability of RGMs, where a high overlap value implies low variability.`
> - **Tractability:** Maintaining the feasibility of obtaining closed-form results of graph statistics.
>     - See also, e.g., Lines 127-129: `Specifically, the closed-form tractability results on subgraph densities derived by us are usually unavailable for existing RGMs with edge dependency.`
>     - See also our <Overall Clarification OC1> above.
>
> **<Argument 1b-3> *Those “desirable properties” are common across many domains.***
>
> - **Realisticness: We acknowledge that some patterns may not exist in *all* domains, but patterns are called patterns because they exist in real-world graphs *in many different domains*.**
>     - See, e.g., Chakrabarti and Faloutsos. "Graph mining: Laws, generators, and algorithms." ACM computing surveys (CSUR) 38.1 (2006): 2-es.
> - **Variability and tractability:** Those are mathematically desirable properties of random graph models, and we believe they are ***not domain-dependent at all.***

---

> ### Author Response · Authors · 2024-11-20
> **<Response 2> On Parameter Fitting**
>
> ## **<Response 2> On Parameter Fitting**
>
> ### **Context: Related Comments**
>
> - The model generates graphs after fitting model parameters to real-world networks, rather than **directly from scratch**. It is unclear whether this model could lend itself to straightforward manual parameterization to generate synthetic graphs independently, a feature offered by models like the Lancichinetti-Fortunato-Radicchi (LFR) benchmark graph generator. **This suggest that the graph generator has a narrow practical use.**
>
> ### **Our Response**
>
> ***Note: See our <Overall Clarification OC2> above for more details.***
>
> **<Clarification 2-1> *The proposed models can also be used to “generate graphs from scratch”.*** To do so, instead of finding the parameters by fitting them to existing graphs, one can just freely choose the parameters (the underlying edge probabilities and the binding parameters).
>
> **<Additional Experiments 2-2>** We conducted additional experiments, using the proposed models to generate graphs “from scratch”. ***By choosing proper edge probabilities and varying node-sampling probabilities, we are able to generate a spectrum (series) of random graphs with different levels of clustering.***
>
> **<Clarification 2-3>** Random graph models (RGMs) have different usages, e.g., “generating random graphs from scratch” and “fitting the model to specific graphs and generating similar graphs”. These two usages are complementary—each addresses distinct needs within the broader context of network analysis. ***Specifically, “fitting the model to specific graphs and generating similar graphs” has significant values with many applications.***

---

> ### Author Response · Authors · 2024-11-20
> **<Response 3> On “Scalability” and “Tractability”**
>
> ## **<Response 3> On “Scalability” and “Tractability”**
>
> ### **Context: Related Comments**
>
> - It is unclear if the model can efficiently generate realistic **extremely large graphs**, given that the experimental results focus on relatively small networks.
> - The explanation of the binding schemes' **tractability** could be more detailed and intuitive, especially as it seems that binding is computationally slower than edge-independent methods, which could hinder its **scalabiltiy to larger graphs**.
>
> ### **Our Response**
>
> **<Clarification 3-1>** In this work, ***“scalability” and “tractability” are different***. A random graph model (RGM) is “tractable” if users ***can compute and control graph statistics*** of the graphs generated by the RGM.
>
> - See our **<Overall Clarification OC1>** above for more details.
>
> **<Clarification 3-2>** We actually examined the scalability of our algorithms **in the initial submission.**
>
> - **See Appendix E.4 (Table 17; it was Table 12 in the initial submission). We showed the scalability of our algorithms while up-scaling a dataset up to 64 times (with 128k nodes) with 32GB RAM, as also mentioned in the end of Section 6.4 (Lines 467-469; it was Lines 469-471 in the initial submission).**
>
> **<Additional Experiments 3-3>** For parallel binding, we further **optimized our implementation** for even larger graphs and tested our algorithms on graphs with **up to 64 million nodes**.
>
> - **[Implementation Optimization 1] Instead of storing the edge probabilities for all $O(n^2)$ node pairs, we only store the equivalent classes (each class contains node pairs with the same edge probabilities).**
>     - For Erdos-Renyi (ER): A single value for all node pairs
>     - For Chung-Lu (CL): A single value for each pair of degrees
>     - For the stochastic block model (SB): A single value for each pair of blocks
>     - For Kronecker (KR): We compute prob using Kronecker product on the fly for each sampled node pairs
> - **[Implementation Optimization 2]** Instead of storing the generated edges in the RAM, we directly store them in the hard disk
>     - **Special technique for Erdos-Renyi:** In each round, we generate a clique (or nothing), so we only need to record the nodes of each generated clique instead of all the generated edges.
>     - **Similarly, for the stochastic block model:** In each round, within each block, we generate a clique or nothing. Between two blocks, we generate a bi-clique or nothing. Therefore, we only need to record the nodes for each block and node pairs for each block pair, instead of all the generated edges.
>
> | model | \|V\| | 1M | 2M | 4M | 8M | 16M | 32M | 64M |
> | --- | --- | --- | --- | --- | --- | --- | --- | --- |
> | ER + Parallel binding | avg. \|E\| | 4.03B | 16.11B | 64.4B | 257.7B | 1.0T | 4.1T | 16.5T |
> |  | time (sec.) | 5.942 | 12.449 | 28.174 | 60.975 | 121.889 | 262.736 | 490.985 |
> | CL + Parallel binding | avg. \|E\| | 8.0M | 16.1M | 32.2M | 64.4M | 128.8M | 257.6M | 515.1M |
> |  | time (sec.) | 102.150 | 220.177 | 423.836 | 815.883 | 1685.561 | 3135.217 | 6179.357 |
> | SB + Parallel binding | avg. \|E\| | 4.03B | 16.11B | 64.4B | 257.7B | 1.0T | 4.1T | 16.5T |
> |  | time (sec.) | 106.026 | 213.722 | 428.980 | 869.002 | 1798.333 | 3829.563 | 8638.938 |
> | KR + Parallel binding | avg. \|E\| | 81.6M | 210.5M | 543.0M | 1.4B | 3.6B | 9.3B | 24.1B |
> |  | time (sec.) | 105.062 | 219.351 | 439.110 | 875.381 | 1751.339 | 3504.719 | 7014.911 |
>
> - **Note:** Parallel binding is easily parallelizable. We can distribute the generation to multiple machines and finally merge the generated edges, which allows us to handle even larger graphs.
>
> **<Revision 3-4> We added these additional results in the revised manuscript.**
>
> - See Appendix E.4 (Table 18).

---

> ### Author Response · Authors · 2024-11-20
> **<Response 4> On Comparison with Established Benchmarks (LFR)**
>
> ## **<Response 4> On Comparison with Established Benchmarks (LFR)**
>
> ### **Context: Related Comments**
>
> - The paper does not compare with the **Lancichinetti–Fortunato–Radicchi (LFR)** benchmark data generator, which also generates graphs with power-law degree distributions, **variability**, and high clustering. Given that the LFR model shares similar motivations and goals with the proposition, a comparison here would add necessary context. Additionally, faster and more efficient follow-up models to LFR align well with the goals of this study, particularly in terms of **tractability**.
>
> ### **Our Response**
>
> Thanks for pointing out the related work.
>
> **<Additional Experiments 4-1> We evaluated the LFR model, and the LFR model failed to generate graphs with high clustering and high variability at the same time.**
>
> - **Why does the LFR model fail?** In our understanding, the LFR model essentially shifts edge probabilities in Chung-Lu to make the edge probabilities more biased to intra-community node pairs. Therefore, the LFR model is essentially an EIGM and inevitably has low variability if we want to use it to generate high clustering, as we discussed in Section 3.1 (see Lines 112-115).
>     - There exist methods that shift edge probabilities in order to improve upon existing EIGMs. Such methods are essentially still EIGMs, they inevitably have high overlap (i.e., low variability).
> - Due to the limitation of LFR in parameter fitting (see <Clarification 4-2> below), for each real-world graph used in our experiments, we do the following:
> - For the degree sequence, we use the ground-truth degrees in the graph.
> - For the community sizes, we use the Louvain method to detect communities in the graph and use the sizes of the detected communities.
> - **For the remaining hyperparameters (i.e., the mixing parameter $\mu$), we try different values. Specifically, from $0$ to $1$ with step size $0.1$.**
> - **For some cases with small $\mu$ values, the LFR fails to generate valid graphs.** Specifically, we encountered the error `Graph not realizable. The maximum internal degree is greater than the largest possible internal degree.`
>     - Such cases are recorded as “ERROR” in the table below.
> - Below are the results on the *hamsterster* and *facebook* datasets (as in Table 4 in Section 6.6). See Table 19 in Appendix E.6 for the full results.
>
> | dataset | hamsterster |  |  |  | facebook |  |  |  |
> | --- | --- | --- | --- | --- | --- | --- | --- | --- |
> | metric | # triangles | GCC | ALCC | overlap | # triangles | GCC | ALCC | overlap |
> | GT | 1.000 | 0.229 | 0.540 | N/A | 1.000 | 0.519 | 0.606 | N/A |
> | CL + Local Binding | 0.992 | 0.165 | 0.255 | 5.8% | 1.026 | 0.255 | 0.305 | 6.3% |
> | CL + Parallel Binding | 1.000 | 0.185 | 0.471 | 5.9% | 1.006 | 0.336 | 0.626 | 6.2% |
> | PA | 0.198 | 0.049 | 0.049 | 4.7% | 0.120 | 0.061 | 0.061 | 6.2% |
> | RGG ($d=1$) | 1.252 | 0.751 | 0.751 | 0.8% | 0.607 | 0.751 | 0.752 | 1.1% |
> | RGG ($d=2$) | 1.011 | 0.595 | 0.604 | 0.8% | 0.492 | 0.596 | 0.607 | 1.1% |
> | RGG ($d=3$) | 0.856 | 0.491 | 0.513 | 0.8% | 0.421 | 0.494 | 0.518 | 1.1% |
> | BTER | 0.991 | 0.290 | 0.558 | 53.8% | 0.880 | 0.525 | 0.605 | 68.0% |
> | LFR ($\mu=0.0$) | 1.140 | 0.262 | 0.546 | 43.5% | ERROR | ERROR | ERROR | ERROR |
> | LFR ($\mu=0.5$) | 0.296 | 0.068 | 0.081 | 13.4% | 0.161 | 0.084 | 0.120 | 17.0% |
> | LFR ($\mu=1.0$) | 0.197 | 0.045 | 0.047 | 7.0% | 0.105 | 0.055 | 0.059 | 6.7% |
>
> **<Clarification 4-2> It is known that it is challenging to analyze the LFR model theoretically (i.e., LFR has low tractability).**
>
> - **Ref:** Kaminski et al. "Artificial Benchmark for Community Detection (ABCD)—Fast random graph model with community structure." Network Science 9.2 (2021): 153-178.
> - As a speed-up version of LFR, although the ABCD model by Kaminski et al. is faster and easier to analyze theoretically, only asymptotic results are available, just as we discussed in in Section 3.2 (see Lines 129-130).
>     - `Usually, only asymptotic results, as the number of nodes approaches infinity, are available for such models.`
> - Consequently, there are no known standard parameter-fitting algorithms for LFR.
>
> **<Revision 4-3> We include the discussions and results on the LFR model in the revised manuscript.**
>
> - In Section 3.1, we now also cite the LFR model when we discuss methods that “shift edge probabilities”.
> - The results of the LFR model are now included in Section 6.6 (Table 4) and in Appendix E.6 (Table 19).
>
> **<Clarification 4-4>** In this work, ***“scalability” and “tractability” are different***. A random graph model (RGM) is “tractable” if users ***can compute and control graph statistics*** of the graphs generated by the RGM.
>
> - See our **<Overall Clarification OC1>** above for more details.
>
> **[Implementation details]**
>
> For the LFR model, we use the NetworkKit implementation. See https://networkit.github.io/dev-docs/cpp_api/classNetworKit_1_1LFRGenerator.html.

---

> ### Author Response · Authors · 2024-11-20
> **<Response 5> On Reproducibility (Code)**
>
> ## **<Response 5> On Reproducibility (Code)**
>
> ### **Context: Related Comments**
>
> - The authors provide a thorough reproducibility package, including all experimental workflows, parameters, and scripts, though this package could benefit from **more comprehensive documentation**.
>
> ### **Our Response**
>
> **<Revision 5-1> We added more documentation and instructions to the code.**
>
> - We added the description of each file in the README.
> - See the updated reproducibility package at https://anonymous.4open.science/r/epgm-7EBE

---

> ### Author Response · Authors · 2024-11-30
> **Follow-up regarding our latest responses**
>
> Dear Reviewer qw53,
>
> Thank you for raising your score and for providing constructive and insightful comments throughout the review process.
>
> We would like to follow up regarding our latest responses on the definition and quantification of “realisticness” to ensure we have adequately addressed your concerns. Specifically:
> - **Summary of <Additional Response AR1>:** In the revised manuscript, we have avoided using the term “realistic” and replaced it with clearer, more precise language.
> - **Summary of <Additional Response AR2>:** We have elaborated on what our method for quantifying “realisticness” is and why it is reasonable and appropriate.
>
> If our responses have resolved your concerns, we would greatly appreciate it if you could consider adjusting your evaluation. Should you have any further suggestions or comments, particularly on the definition or quantification of “realisticness”, we welcome your input and are fully committed to incorporating your valuable feedback.
>
> Thank you again for your time and thoughtful review.

---

### Author Response · Authors · 2024-12-03
**Summary of Discussions (Part 1/2: Strengths)**

Dear (S)ACs and Reviewers,

As the discussion period is ending, we would like to present a summary of our discussions with the reviewers for reference.
First, we thank all the reviewers for their invaluable time and constructive comments.

---

We are greatly encouraged that the reviewers found our work has the following strengths, aligning with our claims in the paper:

### **[Strength 1] An important and motivating problem of improving upon edge-independent graph models to develop better random graph models**
- [LPrH] `The central problem that is tackled, alleviating limitations of edge-independent / inhomogeneous ER graphs, is well-motivated in the text`
- [xVy1] `The topic of "realistic" random graph models is very important and this paper certainly advances the literature (even if by a small step) on this topic`

### **[Strength 2] An interesting idea of edge-dependent graph models (EPGMs) with solid theoretical contributions**
- [qw53] `The paper builds on a solid theoretical foundation of edge-dependent random graph models`
- [sYr6] `Overall, the idea of binding is interesting`
- [sYr6] `They also present closed-form results concerning triangles and discuss time complexities`
- [LPrH] `The proposed concept of binding can be applied to augment several different edge-independent random graph models`
- [LPrH] `There is theoretical work guaranteeing the tractability of the sampling algorithms`
- [xVy1] `An elegant, novel and flexible method to generate random graphs`
- [xVy1] `A good step in the right direction and the basic binding primitive is natural and analyzable`

### **[Strength 3] Empirical strengths of the proposed models and algorithms to generate better random graphs**
- [qw53] `The paper demonstrates a clustering and variability across by fitting different real-world datasets`
- [qw53] `The paper offers a potentially useful tool for generating random graphs with enhanced clustering and variability`
- [LPrH] `Experiments demonstrate the effective of binding at matching triangle on four well-known kinds of RGMs and several well-known graph datasets`
- [LPrH] `Empirical validation of the increased quality of samples when using binding, showing that upon fitting parameters of their model to match the input graph's triangle count, samples indeed match the triangle count closely, while not harming how well the node degree and node pair distance distributions are matched`
- [xVy1] `The authors also run some experiments demonstrating the verstality of this method in practice, showing ability to generate diverse and realistic degree distributions or obtain a large triangle coefficient. Another experiment shows that the running time is decently fast, especially for the parallelizable version`

### **[Strength 4] Clear and logical writing**
- [LPrH] `The organization of the paper is generally logical and clear`
- [xVy1] `The paper is well written in general. The mathematical statements are clear, intuitive, interesting, and believable`

### **[Strength 5] Good reproducibility with public code and details in the supplementary materials**
- [qw53] `The supplementary materials provide sufficient details for understanding and reproducing the experiments`
- [qw53] `The authors provide a thorough reproducibility package, including all experimental workflows, parameters, and scripts`

---

> ### Author Response · Authors · 2024-12-03
> **Summary of Discussions (Part 2/2: Fruits)**
>
> Moreover, through our fruitful discussions with the reviewers, our work has been further strengthened. Below, let us summarize the major "fruits" of the discussions.
>
> ## **Fruits Part 1: Additional Experiments**
>
> ### **[Fruit 1] Additional experiments on more random graph models** (Reviewers qw53 and sYr6)
> - On top of the baseline models in our initial submission (e.g., preferential attachment, random geometric graphs, and BTER), we further conducted experiments on three other models: (1) the Lancichinetti-Fortunato-Radicchi (LFR) benchmark graph generator, (2) exponential random graph models (ERGMs), (3) sort and smooth (SAS) for the graphon model.
> - **Result summary:** The proposed models also outperform those models.
>
> ### **[Fruit 2] Additional experiments on more graph metrics** (Reviewers qw53, sYr6, and xVy1)
> - On top of the graph metrics in our initial submission (e.g., clustering, degrees, distances, modularity, and core numbers), we further examined four other network properties: (1) conductance, (2) average vertex betweenness centrality, (3) average edge betweenness centrality, and (4) natural connectivity.
> - **Result summary:** With binding, the generated graphs are also more realistic w.r.t. those four additional network properties.
>
> ### **[Fruit 3] Additional experiments on scalability** (Reviewer qw53)
> - **Result summary:** On top of the scalability results in our initial submission, for parallel binding, we further optimized our implementation for even larger graphs and tested our algorithms on graphs with up to 64 million nodes.
>
> ### **[Fruit 4] Additional implementation and experiments on higher-order motifs** (Reviewer sYr6)
> - **Result summary:** We wrote additional code for calculating 4-motif probabilities, and used the additional code to calculate and fit the number of 4-cliques.
>
> ### **[Fruit 5] Additional experiments on fitting consistency** (Reviewer qw53)
> - **Result summary:** We showed that fitting results are consistent across different runs and that from random graphs generated using binding, we can accurately infer the parameters used to generate them.
>
> ### **[Fruit 6] Additional experiments on generating graphs “from scratch”** (Reviewer qw53)
> - **Result summary:** By choosing proper edge probabilities and varying node-sampling probabilities, we are able to generate a spectrum (series) of random graphs with different levels of clustering.
>
> ### **[Fruit 7] Additional experiments on downstream applications in graph learning** (Reviewer xVy1)
> - We explored the proposed models' potential for anonymizing graph topologies when the original structure cannot be shared due to privacy concerns, without harming the performance of downstream tasks.
> - **Result summary:** We found that for heterophilious graphs (i.e., graphs with low label homophily), even trained with randomized graph topology generated by the proposed models, the trained GNN models can still perform well on the original datasets.
>
> ---
>
> ## **Fruits Part 2: Additional Discussions and Clarifications**
>
> ### **[Fruit 8] Additional clarifications on the desirable properties of random graph models** (Reviewer qw53)
> - We clarified the specific targets regarding those desirable properties.
> - We replaced the term “realistic” with clearer, more precise language.
>
> ### **[Fruit 9] Additional clarifications on the differences between “tractability” and “scalability”** (Reviewer qw53)
> - We clarified the differences between “tractability” and “scalability”: `tractability refers to the feasibility of deriving graph statistics, rather - than the ability to handle large-scale graphs (which we refer to as scalability).`
>
> ### **[Fruit 10] Additional discussions on exchangeable network models** (Reviewer sYr6)
> - We discussed how exchangeable network models involve edge dependency.
> - We discussed the similarities and differences between the ideas in some non-exchangeable network models and those in the proposed models.

---

### Meta-Review · Area_Chair_m5u1 · 2024-12-20

**Metareview:**

Reviews on this paper were split. Reviewers felt that the problem being addressed by the paper (the inability of edge-independent graph models to capture realistic network features) is important. The solution of correlating edge selections together via `binding' is simple and elegant, and allows for a theoretically tractible model. The binding technique can be applied to any underlying edge-independent model.

However, there are concerns that the binding model may not be the best choice in terms of generating realistic graphs. Of course, binding by design can allow targeting any specified triangle density in the generated graph. But this is just one property that one may consider when looking at the realism of a graph model. The authors did add experiments during the rebuttal studying other important graph properties like conductance and vertex centrality measures. They show that the binding approach at least doesn't hurt here, but it doesn't seem to help much either.

Overall, this paper was at the borderline of the bar for acceptance.

**Additional Comments On Reviewer Discussion:**

The excessive length and confusing structure of the rebuttal significantly hampered efficient evaluation of the paper. A more concise and to-the-point rebuttal would be more effective in the future.

---

### Decision · Program_Chairs · 2025-01-22

Reject